**Method**

# AI-guided pipeline for protein–protein interaction drug discovery identifies a SARS-CoV-2 inhibitor

Philipp Trepte [1,2,17✉], Christopher Secker [1,3,17✉], Julien Olivet [4,5,6,7,8,18,19], Jeremy Blavier [4,18], Simona Kostova[1], Sibusiso B Maseko [4], Igor Minia[9], Eduardo Silva Ramos [1], Patricia Cassonnet[10], Sabrina Golusik[1], Martina Zenkner[1], Stephanie Beetz[1], Mara J Liebich[1], Nadine Scharek[1], Anja Schütz [11], Marcel Sperling [12], Michael Lisurek [13], Yang Wang[5,6,7], Kerstin Spirohn[5,6,7], Tong Hao[5,6,7], Michael A Calderwood [5,6,7], David E Hill [5,6,7], Markus Landthaler [9,14], Soon Gang Choi [5,6,7✉], Jean-Claude Twizere [4,5,15,16,19✉], Marc Vidal [5,6,19✉] & Erich E Wanker [1,19✉]

## Abstract

Protein–protein interactions (PPIs) offer great opportunities to expand the druggable proteome and therapeutically tackle various diseases, but remain challenging targets for drug discovery. Here, we provide a comprehensive pipeline that combines experimental and computational tools to identify and validate PPI targets and perform early-stage drug discovery. We have developed a machine learning approach that prioritizes interactions by analyzing quantitative data from binary PPI assays or AlphaFold-Multimer predictions. Using the quantitative assay LuTHy together with our machine learning algorithm, we identified high-confidence interactions among SARS-CoV-2 proteins for which we predicted three-dimensional structures using AlphaFold-Multimer. We employed VirtualFlow to target the contact interface of the NSP10-NSP16 SARS-CoV-2 methyltransferase complex by ultra-large virtual drug screening. Thereby, we identified a compound that binds to NSP10 and inhibits its interaction with NSP16, while also disrupting the methyltransferase activity of the complex, and SARS-CoV-2 replication. Overall, this pipeline will help to prioritize PPI targets to accelerate the discovery of early-stage drug candidates targeting protein complexes and pathways.

**Keywords** Protein–Protein Interactions; Machine Learning; AlphaFold; VirtualFlow; SARS-CoV-2
**Subject Categories** Methods & Resources; Pharmacology & Drug Discovery; Proteomics

## Introduction

Enzymes, ion channels, and receptors are among the most favored proteins for target-based drug discovery (Santos et al, 2017). However, the number of newly approved drugs per billion dollars invested per year has decreased in the last 60 years (Scannell et al, 2012; Ringel et al, 2020), and the currently approved small molecules target less than 700 proteins altogether or ~3% of the human protein-coding genome (Harding et al, 2018). Proteins are part of signaling pathways and multisubunit complexes (Vidal et al, 2011), thus their macromolecular interactions such as protein–DNA and protein–protein interactions (PPIs) are key targets to expand the druggable proteome (Makley and Gestwicki, 2013; Lu et al, 2020). Consequently, characterizing molecular complex interactions and defining contacts between constitutive protein subunits is essential to identify new classes of targets for drug discovery and development.

Affinity purification coupled to mass spectrometry (AP-MS) techniques are highly efficient in identifying the composition of protein complexes at proteome scale (Huttlin et al, 2021; Bludau and Aebersold, 2020), while binary PPI assays such as yeast two-

[1]Proteomics and Molecular Mechanisms of Neurodegenerative Diseases, Max Delbrück Center for Molecular Medicine in the Helmholtz Association, 13125 Berlin, Germany. [2]Brain Development and Disease, Institute of Molecular Biotechnology of the Austrian Academy of Sciences, 1030 Vienna, Austria. [3]Zuse Institute Berlin, Berlin, Germany. [4]Laboratory of Viral Interactomes, Interdisciplinary Cluster for Applied Genoproteomics (GIGA)-Molecular Biology of Diseases, University of Liège, 4000 Liège, Belgium. [5]Center for Cancer Systems Biology (CCSB), Dana-Farber Cancer Institute, Boston, MA 02215, USA. [6]Department of Genetics, Blavatnik Institute, Harvard Medical School, Boston, MA 02115, USA. [7]Department of Cancer Biology, Dana-Farber Cancer Institute, Boston, MA 02215, USA. [8]Structural Biology Unit, Laboratory of Virology and Chemotherapy, Rega Institute for Medical Research, Department of Microbiology, Immunology and Transplantation, Katholieke Universiteit Leuven, 3000 Leuven, Belgium. [9]RNA Biology and Posttranscriptional Regulation, Max Delbrück Center for Molecular Medicine in the Helmholtz Association, Berlin Institute for Medical Systems Biology, 13125 Berlin, Germany. [10]Département de Virologie, Unité de Génétique Moléculaire des Virus à ARN (GMVR), Institut Pasteur, Centre National de la Recherche Scientifique (CNRS), Université de Paris, Paris, France. [11]Protein Production & Characterization, Max Delbrück Center for Molecular Medicine in the Helmholtz Association, 13125 Berlin, Germany. [12]Multifunctional Colloids and Coating, Fraunhofer Institute for Applied Polymer Research (IAP), 14476 Potsdam-Golm, Germany. [13]Structural Chemistry and Computational Biophysics, Leibniz-Institut für Molekulare Pharmakologie (FMP), 13125 Berlin, Germany. [14]Institute of Biology, Humboldt-Universität zu Berlin, 13125 Berlin, Germany. [15]TERRA Teaching and Research Center, Gembloux Agro-Bio Tech, University of Liège, 5030 Gembloux, Belgium. [16]Laboratory of Algal Synthetic and Systems Biology, Division of Science and Math, New York University Abu Dhabi, Abu Dhabi, UAE. [17]These authors contributed equally as first authors: Philipp Trepte, Christopher Secker. [18]These authors contributed equally as second authors: Julien Olivet, Jeremy Blavier. [19]These authors contributed as senior authors: Julien Olivet, Jean-Claude Twizere, Marc Vidal, Erich E Wanker. ✉E-mail: philipp.trepte@imba.oeaw.ac.at; secker@zib.de; soongangchoi@gmail.com; jean-claude.twizere@uliege.be; marc_vidal@dfci.harvard.edu; erich.w@mdc-berlin.de

hybrid (Y2H) provide high-quality information about directly interacting, or "contacting", protein subunits (Luck et al, 2020). Structural biology technologies, and in particular cryo-electron microscopy (cryo-EM), capture near-atomic resolution pictures of complexes purified from native sources (Costa et al, 2017; Callaway, 2020). Also, they provide information on the precise assembly of subunits and the organization of their interaction interfaces. However, out of the ~7000 protein complexes that have been found in the human proteome (Drew et al, 2021), only ~4% of them currently have an experimentally resolved structure in the literature, which calls for complementary approaches to rapidly model subunit–subunit interactions.

Computational predictions are on the rise to help address this challenge. On the one hand, predictions of 3D protein structures based on artificial intelligence (AI) strategies such as those available in AlphaFold and RoseTTAFold (Jumper et al, 2021; Baek et al, 2021) can be exploited to model protein assemblies and interaction interfaces with much improved accuracies than previous computational tools (Evans et al, 2022; Gao et al, 2022). On the other hand, platforms like VirtualFlow can be used to screen billions of molecules in silico against a predicted target in a time- and cost-effective manner (Gorgulla et al, 2020).

Here, we combine experimental binary PPI mapping with in silico structure prediction and virtual screening for PPI-based drug discovery. We first used reference sets of PPIs and quantitative interaction data from seven binary PPI assays to establish an unbiased machine learning PPI scoring approach. We then applied this strategy to map and prioritize interactions between SARS-CoV-2 proteins and used AlphaFold to determine the corresponding 3D protein complex structures. Finally, we targeted the contact interface of the NSP10-NSP16 complex in an ultra-large virtual drug screening with VirtualFlow and identified a small-molecule PPI inhibitor that reduces the NSP16-linked methyltransferase activity and SARS-CoV-2 replication. Our findings show that combining high-quality quantitative binary interaction data, AI-based scoring systems, and computational modeling can help prioritize PPI targets for the development of novel therapeutics.

# Results

## Scoring binary interaction assays using fixed cutoffs results in variable recovery rates

We previously demonstrated that combining multiple complementary interaction assays and/or versions thereof significantly increases PPI recovery while maintaining high specificity (Venkatesan et al, 2009; Choi et al, 2019; Trepte et al, 2018). LuTHy, a bioluminescence-based technology (Trepte et al, 2018) combines two readouts in one. First, a bioluminescence resonance energy transfer (BRET)-based readout is used to quantify interactions in living cells (LuTHy-BRET; Fig. EV1A); then, cells are lysed and the luminescence is used to quantify interactions after protein co-precipitation (LuTHy-LuC; Fig. EV1A). Since LuTHy plasmids allow expression of each protein as N- or C-terminal fusions, and as donor (NanoLuc tag or NL) or acceptor (mCitrine tag or mCit) proteins, eight tagging configurations can be assessed for every protein pair of interest (Fig. EV1B). Thus, when all eight

configurations are tested, LuTHy-BRET and LuTHy-LuC assays generate a total of 16 data points for every tested X–Y pair.

To determine the accuracy of the LuTHy assay and compare it to other binary interaction assays (Choi et al, 2019; Yao et al, 2020), we tested an established positive reference set (PRS), hsPRS-v2, which contains 60 well-characterized human PPIs (Venkatesan et al, 2009; Choi et al, 2019). To control for specificity, a random reference set (RRS), hsRRS-v2, made of 78 pairs of human proteins not known to interact (Choi et al, 2019), was also tested (Source Data Fig. 1). To coherently score quantitative PPI data among different readouts and assays, we initially tried two different approaches: (i) we applied a receiver operating characteristic (ROC) analysis to determine cutoffs at maximal specificity (i.e., under conditions where none of the random protein pairs from hsRRS-v2 are scored positive in any of the tested configurations (Fig. EV1C), and (ii) we determined cutoffs based on the distribution of the data at the mean (Fig. EV1D) or median (Fig. EV1E) plus one standard deviation. For all assays, we observed highly variable recovery rates depending on the scoring approach used. For example, for the LuTHy-BRET readout, we recovered 20.0% of hsPRS-v2 interactions and 0.0% of hsRRS-v2 protein pairs at maximal specificity (Fig. EV1C), while we detected 36.7% and 1.3% at the mean plus one standard deviation (Fig. EV1D) and 45.0% and 1.0% at the median plus one standard deviation (Fig. EV1E), respectively. In contrast, for the LuTHy-LuC readout and MAPPIT assay, we obtained the highest recovery rates in the hsPRS-v2 set when applying the maximal specificity cutoff (Fig. EV1C), but lower recovery rates for the distribution-based scoring approaches (Fig. EV1D,E). These findings clearly demonstrate that different scoring approaches can result in highly variable recovery rates for different assays, highlighting the need for more robust approaches to coherently score quantitative PPI data and to obtain comparable results from various binary interaction assays across different labs.

## Establishing a machine learning algorithm to classify binary interactions

To provide a universal and unbiased approach to score and classify quantitative PPI data from various assays, we investigated the use of machine learning-based classifier algorithms. Therefore, we first evaluated a random forest (RF) and a support vector machine (SVM) learning algorithm, which are commonly used for binary classification tasks (Chang and Lin, 2011; Breiman, 2001). We tested both algorithms on the LuTHy assay data obtained from screening the hsPRS-v2 and hsRRS-v2 protein pairs. As training features, we selected for the LuTHy-BRET readout the cBRET ratios and acceptor fluorescence intensities (mCit), and for the LuTHy-LuC readout the cLuC ratios. Instead of training a single machine learning model on the complete reference sets, we trained 50 independent classifier models by assembling independent training sets through randomly sampling a constant amount of protein pairs, so that with a probability of 99.99% each protein pair from the hsPRS-v2 and hsRRS-v2 reference sets was used at least once for training. This strategy was used to prevent model overfitting, as not only a single trained model is used to predict the probability of an interaction to be true positive, but an ensemble of multiple model instances is trained and thus an average classification probability can be determined. Each of the resulting models was then applied to predict the classification

probabilities of the test protein pairs that were not used for training of the respective model. To evaluate training efficiency and the performance of the classifiers on unseen data, we plotted learning curves that show the accuracy, hinge loss and binary cross-entropy-loss when training with only 10% or up to 100% of the totally available data per ensemble and applying each model to the respective test data (Appendix Fig. S1A,B). For the RF algorithm, we observed a much better performance of the models on the training data compared to the test data (Appendix Fig. S1A,B), indicating strong overfitting and suggesting that the models do not generalize well and would perform poorly on unseen, new data. Even though we observed reasonable recovery rates for LuTHy-BRET and LuTHy-LuC protein pairs with >50%, >75%, and >95% interaction probability (Appendix Fig. S1C,D), we excluded the RF algorithm from further analysis due to its weak training performance. For the SVM algorithm, in contrast, we observed good training performance (Appendix Fig. S1A,B), but very low recovery rates (Appendix Fig. S1C,D). We hypothesized that the weak performance of the SVM classifier could be attributed to the fact that many protein pairs from the hsPRS-v2 do not score positive in all possible assays or tagging configurations (Fig. EV1A,B), and are thus technically mislabeled for machine learning training purposes. Notably, this is a common phenomenon of binary interaction assays, in which distant- or precipitation-based readouts might only be successful in certain configurations between distinct protein pairs of interest. For datasets that contain such a high degree of mislabeling, it has been previously shown that a multi-adaptive sampling approach can be used to iteratively update the labeling class of the training data and thereby improve the performance of SVM-based classifiers (Yang et al, 2017). Thus, we evaluated if such a multi-adaptive SVM (maSVM) learning algorithm could indeed improve the classification of quantitative PPI data (Fig. 1A). We first applied it to the LuTHy assay and used as before for the LuTHy-BRET the cBRET ratios and acceptor fluorescence (mCit) and for the LuTHy-LuC the cLuC ratio as training features (see methods for details) and assembled multiple independent training sets by random sampling. By applying multi-adaptive sampling, the label of each reference interaction in the training set was iteratively reclassified during training. Importantly, we optimized selected hyperparameters of the maSVM, namely the ensemble size 'e' (25, 50, or 100), the number of iterative reclassifications 'i' (1, 5, or 10) and the regularization parameter 'C' (0.01, 0.1, 1, or 10; Datasets EV1 and EV2). The training performance of the optimized hyperparameters was evaluated as before using learning curves and for 'e = 50' (Dataset EV3), 'i = 5' and 'C = 1' we obtained good training behavior for both assays and no signs of overfitting, i.e., continuous and steady reduction of the loss functions as well as no large gaps between the training and validation accuracies (Appendix Fig, S1E). Each of the resulting maSVM models was then applied to predict the classification probabilities of the protein pairs that were not used for training (Fig. 1B,C). Finally, the mean probabilities from all model predictions were calculated (Fig. 1D,E). We then performed ROC analyses to compare the sensitivity and specificity when scoring interactions using fixed cBRET or cLuC ratio cutoffs or when using the maSVM model predicted probabilities. Importantly, the maSVM-based scoring did not result in any loss of sensitivity or specificity for both LuTHy readouts, respectively (Fig. 1F,G). We also calculated the recovery rates for LuTHy-BRET and LuTHy-

LuC hsPRS-v2 and hsRRS-v2 protein pairs with >50%, >75%, and >95% interaction probabilities allowing us to distinguish for each assay, protein pairs to be "unlikely" (>50%), "likely" (>75%), or "very likely" (>95%) detected as true-positive interactions. Consequently, we observed that specificity increases with an increasing probability threshold while sensitivity decreases (Fig. 1H,I; Dataset EV4).

Next, we evaluated whether the maSVM algorithm would also be broadly applicable to score quantitative interaction data from various assays. Therefore, we applied it to published benchmarking data (Choi et al, 2019) from six different quantitative binary PPI assays: GPCA (Cassonnet et al, 2011), KISS (Lievens et al, 2014), MAPPIT (Eyckerman et al, 2001), NanoBiT (Dixon et al, 2015), N2H (Choi et al, 2019) and SIMPL (Yao et al, 2020) (Appendix Fig. S2). For each assay, we optimized the selected hyperparameters ('e', 'i', and 'C', Dataset EV1, Dataset EV2, Dataset EV3) to obtain good training behavior and performance of the classifiers (Appendix Fig. S1E), and predicted the interaction probabilities for all screened tagging configurations of hsPRS-v2 interactions and hsRRS-v2 protein pairs (Fig. 1J). Analyzing each screened configuration individually among assays, generally showed that recovered PRS PPIs showed high interaction probabilities not necessarily in all, but often in multiple configurations (e.g., BAD + BCL2L1, Appendix Fig. S3). However, certain hsPRS-v2 interactions only show high interaction probabilities for distinct configurations (e.g., SKP1 + BTRC, Appendix Fig. S3), which is most likely attributed to an increased distance between tags in distance-dependent readouts such as LuTHy-BRET, N2H, GPCA, or NanoBiT, or to tagging configuration-specific precipitation and expression efficiencies for co-precipitation-based assays such as LuTHy-LuC or SIMPL. Interestingly, the few interactions detected within the RRS showed only high interaction probabilities in one or two configurations. For example, the hsRRS-v2 interaction SLC6A1 + TM4SF4 was detected multiple times by different assays, but only in the N1-C2 and C1-C2 configurations, indicating that SLC6A1 might physiologically indeed bind to the C-terminus of TM4SF4 or that C-terminal tagging of TM4SF4 potentially increases its biophysical interaction propensity with SLC6A1 in overexpression systems (Appendix Fig. S4).

Next, we analyzed the recovery rates for each assay by classifying PPIs with an interaction probability >50%, >75% or >95% as positive (Fig. 1K; Dataset EV4). Comparing the maSVM scoring-based recovery rates of the most stringent probability group (>95%) to fixed cutoff-based approaches, we observed for almost all assays an improved (SIMPL, N2H, LuTHy, KISS, GPCA) or similar (NanoBiT) recovery rate of positive reference interactions without any (SIMPL, NanoBiT, MAPPIT, LuTHy, KISS) or only a minor (N2H, GPCA) increase in the recovery of random protein pairs (Fig. 1K). In addition, considering interactions with a probability >75% further increases recovery of positive reference interactions, however, with a partially substantial decrease in specificity (e.g., for LuTHy 55.0 vs. 66.7% hsPRS-v2 and 1.3 vs. 9.0% hsRRS-v2 recovery, respectively). Regarding the advantage of the maSVM-based scoring system over previous approaches, the MAPPIT assay represents an exception. Here, the maximal specificity cutoff leads to the highest recovery of hsPRS-v2 interactions (35% hsPRS-v2), indicating that considering the variance of the dataset in scoring interactions is not favorable in this case. However, considering interactions with a probability score >75% or even >50% for

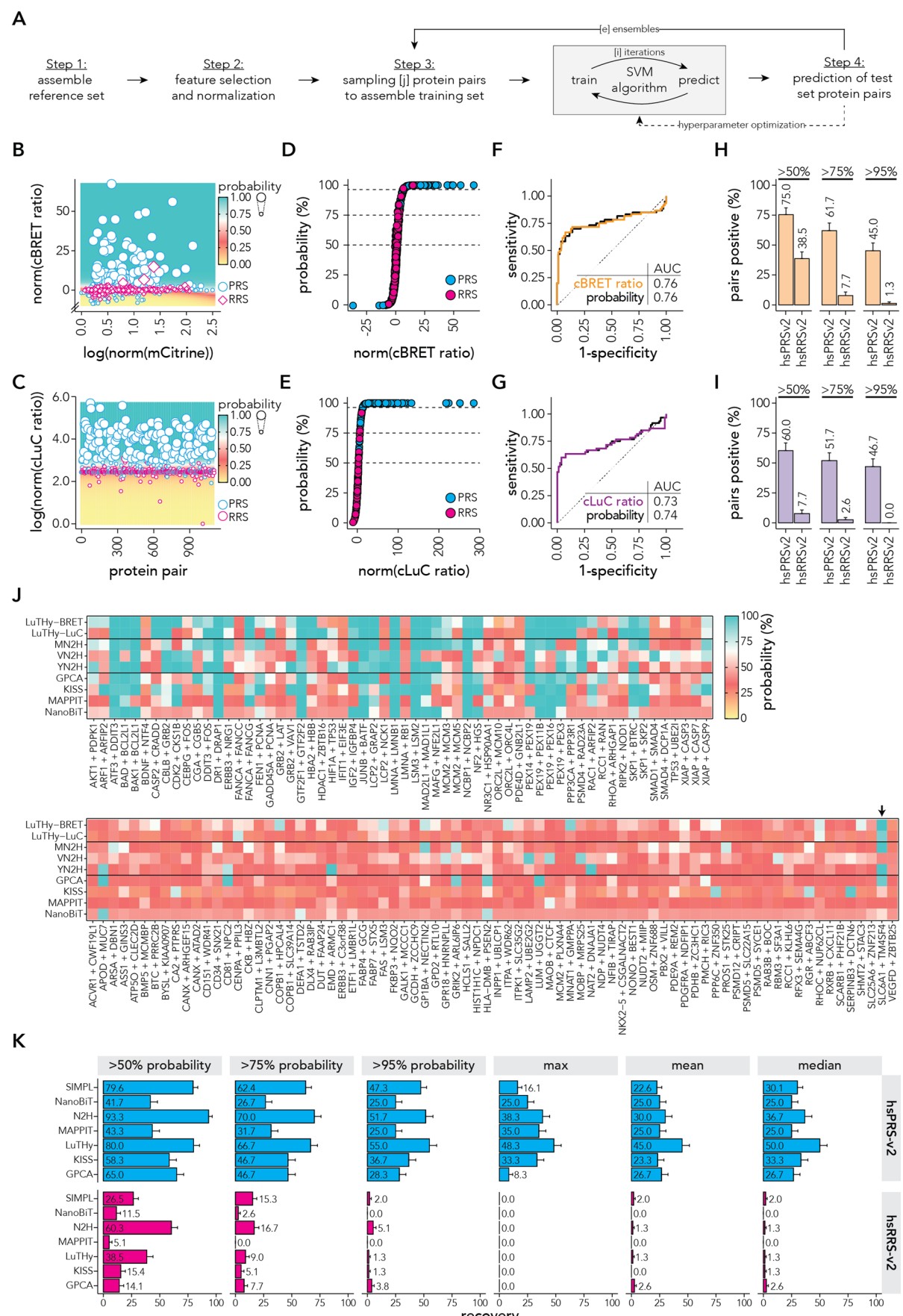

---

**Figure 1. Developing a maSVM algorithm to classify protein pairs from hsPRS-v2 and hsRRS-v2 using the LuTHy assay.**

(A) Schematic overview of the maSVM learning algorithm. Step 1: assembly of reference set; Step 2: feature selection and data normalization for training and test set; Step 3: assembly of 'e' training sets (ensembles) by unweighted sampling 'j' protein pairs from the reference set to train 'e' maSVM models, where the training classifier labels are reclassified in 'i' iterations; Step 4: prediction of test set protein pairs excluding training set pairs using the respective maSVM model. Scatter plot showing (B) log-transformed and normalized in-cell mCitrine expression (*x* axis) against normalized cBRET ratios (*y* axis limited to '>-10') or (C) the number of proteins pairs (*x* axis) against log-transformed and normalized cLuC ratios (*y* axis) for all hsPRS-v2 (blue) and hsRRS-v2 (magenta) protein pairs from all eight tagging configurations. Average classifier probabilities from the 50 maSVM models are displayed as the size of the data points and as a colored grid in the background. Scatter plot showing (D) normalized cBRET ratios (*x* axis) or (E) normalized cLuC ratios (*x* axis) against classifier probability (*y* axis) for all hsPRS-v2 (blue) and hsRRS-v2 (magenta) protein pairs from all eight tagging configurations. Receiver characteristic analysis comparing sensitivity and specificity between (F) cBRET ratios or (G) cLuC ratios and classifier probabilities. The calculated areas under the curve are displayed. Bar plots showing the fraction of hsPRS-v2 and hsRRS-v2 protein pairs that scored above classifier probabilities of 50%, 75%, or 95% with (H) LuTHy-BRET or (I) LuTHy-LuC. Only the highest classifier probability per tested tagging configuration is considered. (J) Heatmaps showing the highest classifier probabilities for the hsPRS-v2 (top) and hsRRS-v2 (bottom) protein pairs per tested tagging configuration. Due to different reference set interactions, heatmaps for SIMPL data from Yao et al (Yao et al, 2020) are shown in Appendix Fig. S3 and Appendix Fig. S4. (K) Bar plots showing the fraction of hsPRS-v2 and hsRRS-v2 protein pairs that scored above classifier probabilities of 50%, 75%, or 95% or above-fixed cutoffs at maximum specificity, mean or median plus one standard deviation for seven binary PPI assays. Only the highest classifier probability per tested tagging configuration is considered. All LuTHy experiments from this study were repeated twice with *n* = 2, biological replicates, each containing *n* = 3 technical replicates; all other from Choi et al (Choi et al, 2019) and Yao et al (Yao et al, 2020). Bars and error bars in this figure represent mean values and standard error of the proportion, respectively. Source data are available online for this figure.

---

MAPPIT as positive, improves recovery (31.7 or 43.3% hsPRS-v2, respectively) with no or only a minor loss in specificity (0.0 or 5.1% hsRRS-v2, respectively). Overall, this analysis suggests that the maSVM learning algorithm is universally applicable to reproducibly and robustly classify quantitative PPI results with improved sensitivity and specificity to traditional approaches, while adding additional information on interaction probabilities and improving comparability between assays.

## Benchmarking AlphaFold against established reference sets of protein pairs

With the emergence of highly accurate protein structure prediction algorithms, we asked how an AlphaFold-Multimer (AFM)-based PPI mapping together with the maSVM-based scoring approach would perform compared to binary PPI assays, when benchmarked against the hsPRS-v2 and hsRRS-v2 protein pairs (Fig. 2A). To this end, we used Google Colaboratory hosted ColabFold that provides accelerated protein complex prediction with the limitation that only protein complexes with less than 1400 amino acids could be predicted (Mirdita et al, 2022). This resulted in the downsizing of the reference sets to 51 positive (hsPRS-AF) and 67 random (hsRRS-AF) reference pairs for which we predicted five AFM complex models for each interaction (Source Data Fig. 2; Figs. EV2 and EV3).

To extract relevant features for training from the predicted complex structures, we used PDBePISA (Krissinel and Henrick, 2007) to obtain the interaction interface areas (iA) and the solvation-free energies (ΔG) for each AFM model that contained a measurable interface (521 out of 590, see methods for detail, Fig. 2B). Since it had been shown that the inter-chain predicted alignment error (inter-PAE) can be used to rank and assess the confidence of a predicted PPI (Mirdita et al, 2022), we also extracted the inter-PAEs from the AFM structures and filtered for amino acids with a predicted local distance difference test (pLDDT) >50 to exclude disordered regions (Tunyasuvunakool et al, 2021). Because PPIs are often driven by hot spot residues that are structurally conserved (Halperin et al, 2004), we used *k*-means clustering to group the respective interface regions into eight inter-PAE clusters. Thereby we identified the residues that are closest to each other and are thus most likely to mediate an interaction

(Fig. 2C,D). If a minimum of 10 amino acids at the inter-chain region had a pLDDT >50 and *k*-means clustering succeeded, we obtained the average PAE value from the inter-subunit amino acid cluster with the lowest average inter-PAEs from all eight clusters. We assumed that this cluster would best represent the dominant region forming the interaction interface. In case *k*-means clustering failed, we used the average inter-PAE of all amino acids with a pLLDT >50. However, if fewer than 10 amino acids at the inter-chain region had a pLDDT score >50, the PAE values were discarded and not used for scoring the respective complex structure (Fig. 2E).

To evaluate which of the obtained measures (i.e., PAE, iA, ΔG) would be the best training features to distinguish between positive and random reference pairs, we performed ROC analyses for each of the five AFM complex structures (Fig. EV2A). We found that all three measures are suitable to identify true-positive interactions, but that the average PAE of the inter-subunit clusters and the iA better distinguish between true- and false-positive interactions compared to the ΔG values (Fig. EV2A). Thus, we trained the maSVM algorithm on the PAE and iA values of the AFM complex structures obtained for the reference set interactions (Fig. 2F,G), performed hyperparameter optimization (Datasets EV1–EV3) and evaluated training performance as before (Appendix Fig. S1E). Using this AFM PPI mapping approach, we were able to identify 62.7% of the hsPRS-AF interactions and 1.5% of the hsRRS-AF pairs as true positives with an interaction probability >95% (Fig. 2H; Dataset EV4). Since AlphaFold's neural networks were trained on PDB structures, it is expected that it shows especially high recovery rates with 74.2% for hsPRS-v2 interactions with experimentally solved structures, including homologous structures (Dataset EV5; Fig. EV2B) (Meyer et al, 2018). Interactions without an experimentally solved structure were recovered with 45.0% at similar sensitivities to the results from the LuTHy assays on non-PDB structures (46.4%), but at noticeably higher sensitivity to the average recovery of not structurally resolved PPIs by the other analyzed binary interaction assays (16.8%, Fig. EV2B). However, it has to be noted that the LuTHy assay was the only assay that was tested in all eight tagging configurations, which can significantly improve recovery.

In summary, our analysis confirms that AFM is a powerful computational tool capable of distinguishing between well-

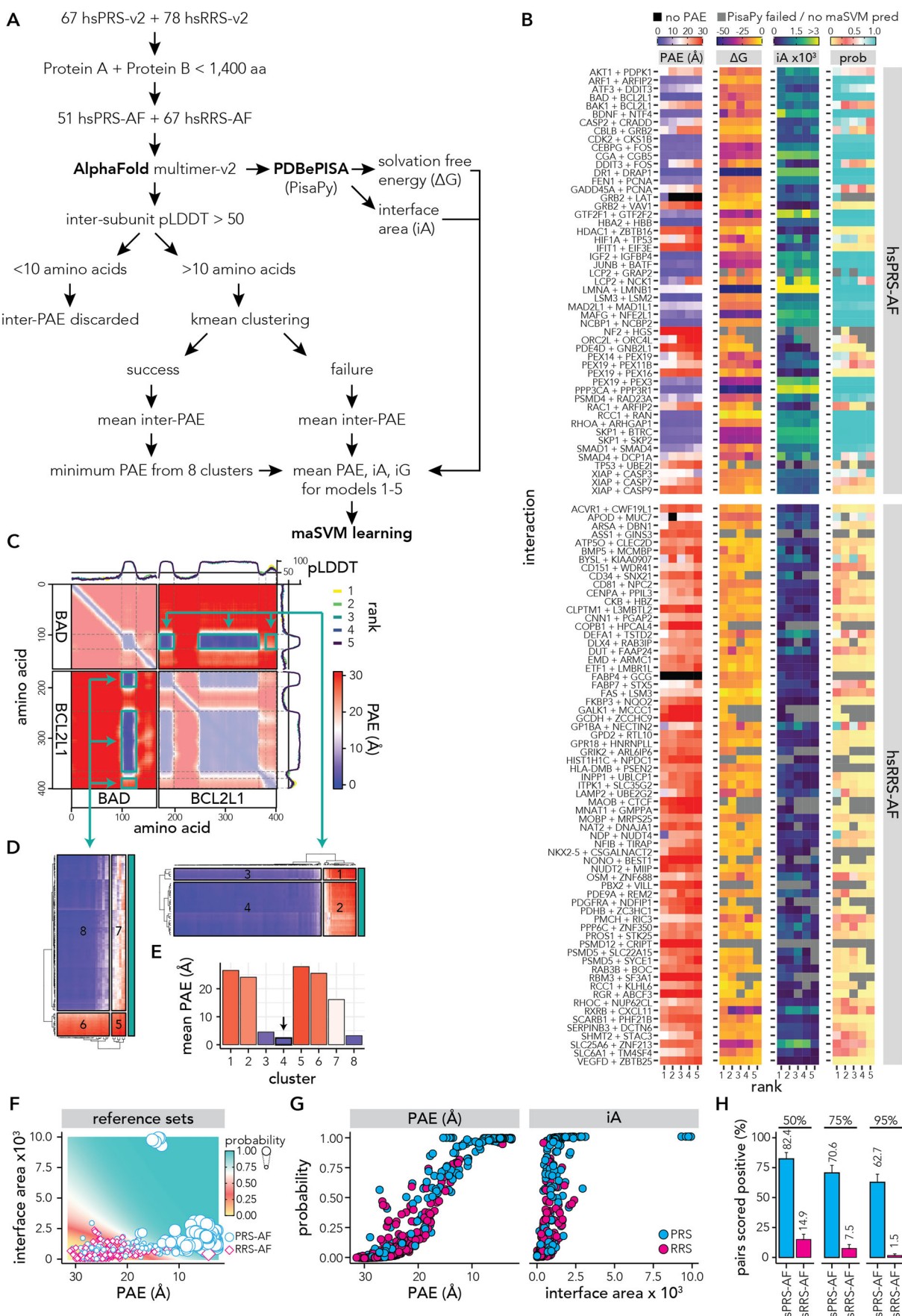

Figure 2. Benchmarking AFM using well-established positive and random reference sets.

**(A)** Schematic overview of AlphaFold-multimer (AFM) benchmarking. First, the hsPRS-v2 and hsRRS-v2 were filtered for protein pairs with less than 1400 amino acids combined, resulting in 51 positive (hsPRS-AF) and 67 random reference set pairs (hsRRS-AF). For these 118 protein pairs, five structural models were predicted each using ColabFold through the AFM algorithm (590 total structures). Following, PAE and pLDDT values were extracted from the AFM-predicted structures, and inter-subunit amino acids were filtered for pLDDT >50. If >10 inter-subunit amino acids remained, PAE values were $k$-means clustered. If clustering failed, the mean PAE of the unclustered amino acids was calculated, else the average PAE for each of the eight clusters were calculated and the cluster with the lowest average PAE was selected as the amino acid region with the minimal distance between the two proteins. In addition, PDBePISA was used to determine the solvation-free energy ($\Delta$G) and the area (iA) of the interface region for 521 of the 590 structures. For the remaining 69 structures PDBePISA could not identify an interface. Finally, a multi-adaptive maSVM learning algorithm was trained on the PAE and iA features of the hsPRS-AF and hsRRS-AF as outlined in Fig. 1A. **(B)** Heatmap of the PAEs, $\Delta$Gs, iAs, and predicted probabilities for protein pairs of the hsPRS-AF and hsRRS-AF. Shown are the minimum PAE values after $k$-means clustering. If <10 amino acids had a pLDDT >50, the PAE values were not used and are shown in black. Protein pairs where no interaction interface was detected by PDBePISA are shown in gray. **(C)** Representative example for the $k$-means clustering strategy of AFM reported PAE values. Heatmap shows the PAEs for the protein pair BAD + BCL2L1 (hsPRS-AF) rank 1 model. The intra-molecular PAEs are shown with 50% opacity. The predicted local distance difference test (pLDDT) for all five predicted models (rank 1–5) are shown as line graphs on top and on the right of the heatmap. Inter-molecular PAE regions with pLDDT >50 that were used for $k$-means clustering are highlighted with arrows. **(D)** Clustering results of regions highlighted in **(C)**. Cluster numbers are indicated. **(E)** Average PAE values for the eight clusters from **(C,D)**. The arrow indicates the cluster with the lowest average PAE value. **(F)** Scatter plot showing inter-PAE ($x$ axis) against interface area ($y$ axis) for all models of the hsPRS-AF (blue) and hsRRS-AF (magenta) protein pairs. Average classifier probability from the 100 maSVM models are displayed as the size of the data points and as a colored grid in the background. **(G)** Scatter plots showing PAE ($x$ axis, left panel) or interface area ($x$ axis, right panel) against classifier probability ($y$ axis) for all hsPRS-AF (blue) and hsRRS-AF (magenta) protein pairs. **(H)** Bar plots showing the fraction of hsPRS-AF and hsRRS-AF protein pairs that scored above classifier probabilities of 50%, 75% and 95%. Bars and error bars represent mean values and standard error of the proportion, respectively, with $n = 5$ structural models predicted. Note that this analysis includes interactions in the hsPRS-AF that have experimentally solved structures. A comparison between interactions with and without structural information can be found in Fig. EV2B. Source data are available online for this figure.

established positive PPIs and random protein pairs with similar accuracy to commonly used binary interaction assays. This also suggests that complex structures of experimentally determined interactions have a high chance to be successfully predicted by AFM.

## Classifying binary interactions within multiprotein complexes

To further generalize the overall applicability of the maSVM algorithm, we aimed to test its performance on different reference sets of protein pairs from multiprotein complexes. Therefore, we investigated proteins that are part of well-characterized complexes and screened their pairwise interactions using two binary interaction assays LuTHy and mN2H. To this end, we selected three human complexes based on the following criteria: (1) they consist of at least four subunits; (2) at least one 3D structure is available in PDB (Berman et al, 2000); and (3) at least 80% of cloned open-reading frames (ORFs) encoding the reported subunits are available in the human ORFeome 8.1 collection (Yang et al, 2011). This resulted in a list of 24 distinct protein complexes (Dataset EV6), among which three structurally diverse candidates with well-characterized biological functions were selected: (1) the LAMTOR complex, also termed "Ragulator" complex (Araujo et al, 2017), which regulates MAP kinases and mTOR activity and consists of seven subunits (LAMTOR1, LAMTOR2, LAMTOR3, LAMTOR4, LAMTOR5, RRAGA and RRAGC); (2) the BRISC complex, a large deubiquitinating machinery (Rabl et al, 2019) consisting of five proteins (ABRAXAS2, BABAM1, BABAM2, BRCC3 and SHMT2); and (3) the MIS12 complex that connects the kinetochore to microtubules (Petrovic et al, 2016), and is made of five subunits (CENPC1, DSN1, MIS12, NSL1, and PMF1) (Fig. 3A).

To map interactions between the subunits of the LAMTOR, BRISC and MIS12 complexes, out of 17 ORFs encoding the selected target proteins, 16 were sequence-verified and cloned into both LuTHy and N2H expression plasmids, whereas the ORF for LAMTOR5 was not available in the human ORFeome 8.1 collection. The resulting search space of 136 unique pairwise

combinations, corresponding to a total of 16 subunits for the three complexes, was systematically assessed with LuTHy and mN2H (Fig. 3B; Source Data Fig. 3). Since the different complexes are involved in distinct biological functions, we rationalized that true binary PPIs are mainly to be found between the respective subunits of a given complex (i.e., intra-complex pairs), but not between subunits belonging to different complexes (i.e., inter-complex pairs). Therefore, we considered all inter-complex pairs as random pairs, similar to protein pairs from a RRS (e.g., hsRRS-v2). In the data analysis, we observed that each individual LuTHy and mN2H fusion construct showed a broad distribution of interaction scores for intra- and inter-complex pairs, with a high variability between individual constructs (Appendix Fig. S5A–F). To compensate for different background signals between constructs in the downstream analysis, we therefore median-normalized outputs from all constructs and performed a robust scaler normalization (Pedregosa et al, 2011) for constructs with a higher interquartile range (IQR) than the IQR of the entire dataset (see "Methods" for details; Appendix Fig. S5A–F).

To classify interactions, we used the maSVM models trained on the hsPRS-v2 and hsRRS-v2 (Fig. 3C–E) to predict the interaction probabilities of the intra-complex interactions and inter-complex protein pairs (Fig. 3F–H). As previously, we calculated recovery rates for LuTHy-BRET, LuTHy-LuC, and mN2H for protein pairs with >50%, >75%, and >95% interaction probabilities. At >95% interaction probability, we recovered between 19 and 38% of the interactions within the BRISC, LAMTOR and MIS12 complexes by the three different assay versions, and between 0 and 2.4% of the inter-complex protein pairs (Fig. 3I–L). When also considering PPIs with an interaction probability >75% as positive, we recovered up to 61.9% of the multiprotein complex interactions; however, with a slightly increased detection of random inter-complex pairs for LuTHy-BRET (0.0 vs. 3.5%) and LuTHy-LuC (1.2 vs. 4.8%), and a more pronounced increase for the mN2H assay (2.4 vs. 12.9%). However, overall, the fraction of detected intra-complex interactions in the multiprotein complex set is similar to the fraction of recovered hsPRS-v2 interactions for the LuTHy and mN2H assays while maintaining a similar specificity.

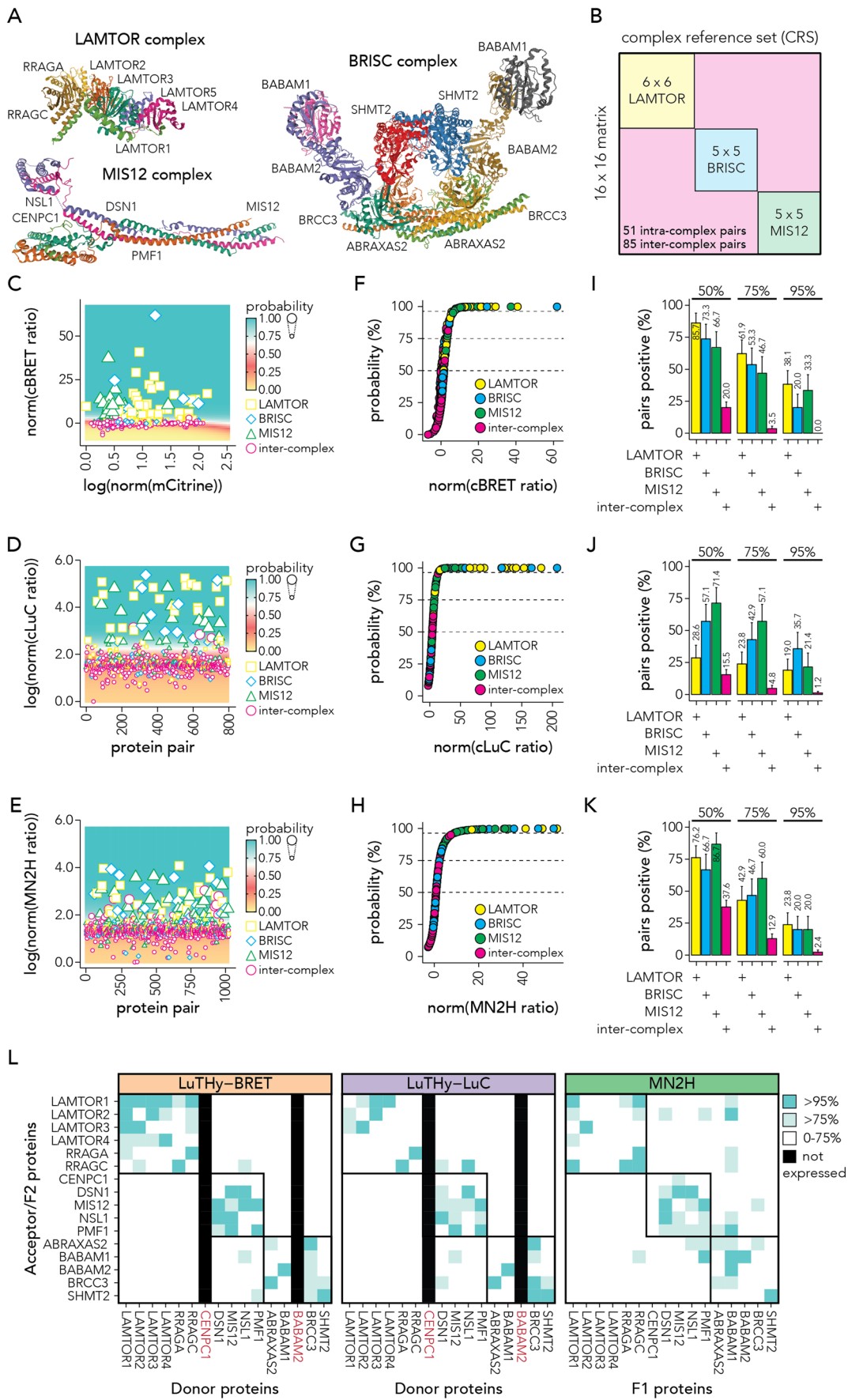

◄ **Figure 3. Validating the maSVM algorithm by mapping interactions within multiprotein complexes using the LuTHy and mN2H assays.**

(A) Structures of the protein complexes analyzed in this study: LAMTOR (PDB: 6EHR), BRISC (PDB: 6H3C) and MIS12 (PDB: 5LSK). (B) Binary interaction approach to systematically map PPIs within distinct complexes. Every protein subunit from each complex was screened against every other one (all-by-all, 16 × 16 matrix). (C–E) Scatter plot showing (C) log-transformed and normalized in-cell mCitrine expression (x axis) against normalized cBRET ratios (y axis), (D) number of protein pairs (x axis) against log-transformed and normalized cLuC ratios (y axis) or (E) the number of protein pairs (x axis) against the log-transformed and normalized mN2H ratios (y axis) for all protein pairs of the LAMTOR (yellow), BRISC (blue) and MIS12 (green) complexes and inter-complex (magenta) protein pairs from all eight tagging configurations. Average classifier probabilities from the 50 maSVM models for LuTHy-BRET (C) and LuTHy-LuC (D) or 100 maSVM models for mN2H (E) are displayed as the size of the data points and as a colored grid in the background. (F–H) Scatter plot showing on the x axis the normalized (F) cBRET ratios, (G) cLuC ratios, or (H) mN2H ratios against classifier probabilities (y axis) for protein pairs of the BRISC (blue), LAMTOR (yellow) and MIS12 (green) complexes and inter-complex (magenta) protein pairs from all eight tagging configurations. (I–K) Bar plots showing the fraction of protein pairs of the LAMTOR (yellow), BRISC (blue) and MIS12 (green) complexes and inter-complex protein pairs that scored above the classifier probabilities of 50%, 75% or 95% by (I) LuTHy-BRET, (J) LuTHy-LuC, and (K) mN2H. Only the highest classifier probability per tested tagging configuration is considered. (L) Tile plots showing the classifier probabilities for the Donor/F1 protein pairs (x axis) against the Acceptor/F2 protein pairs (y axis) for LuTHy-BRET (orange, left), LuTHy-LuC (purple, middle) and mN2H (green, right) for protein pairs above 75% or 95%. Only the highest classifier probability per tested tagging configuration is shown. LuTHy experiments were performed in HEK293 cells two times with n = 2 biological replicates, each containing n = 3 technical replicates. mN2H experiments were performed in HEK293T cells four times with n = 4 biological replicates and n = 1 technical replicate. Tiles of not expressed constructs are filled black and respective protein names are colored in red. Bars and error bars in this figure represent mean values and standard error of the proportion, respectively. Source data are available online for this figure.

Importantly, these results demonstrate that the maSVM PPI classifiers for LuTHy and mN2H, which were trained and benchmarked on the hsPRS-v2 and hsRRS-v2 reference sets, can also be applied to score and identify interactions within a completely independent dataset consisting of large multiprotein assemblies. This indicates that the maSVM-based models generated and provided here, including the ones for other binary PPI assays, are transferable and can be used to score and classify PPIs in diverse datasets. It is important to note, however, that to obtain reliable results, it is key to apply outlier-insensitive normalization, such as robust scaler normalization, prior to using the classifiers on new PPI data.

## Identifying high-confidence PPI targets for SARS-CoV-2

We next applied the maSVM-based scoring approach to identify and prioritize PPIs for drug discovery. Therefore, we experimentally assessed all possible pairwise combinations between SARS-CoV-2 proteins using the LuTHy assay (Fig. 4A). As described above, before classification, we median-normalized interaction scores for all constructs of the SARS-CoV-2 test set and performed a robust scaler normalization for constructs with an IQR higher than the IQR of the entire dataset (see "Methods"; Appendix Fig. S6A–D). We then used the LuTHy maSVM models trained on the hsPRS-v2 and hsRRS-v2 to predict the classification probabilities of the 350 SARS-CoV-2 protein pairs in the test set (2548 configurations, Fig. 4B-G). In total, 29, 68 and 168 protein pairs were classified by the algorithm to interact with >95%, >75% or >50% probability in the LuTHy-BRET assay (Fig. 4B–D; Dataset EV7), and 9, 34, and 76 in the LuTHy-LuC assay, respectively (Fig. 4E–G; Dataset EV7). Among the high-confidence PPIs (>95% interaction probability), we found the structurally resolved interactions between NSP8 and NSP12 (PDB: 6YYT, 7EIZ), NSP10 and NSP16 (PDB: 6WVN, 6W4H) (Rosas-Lemus et al, 2020), NSP10 and NSP14 (PDB: 7DIY, 7EIZ) (Lin et al, 2021; Yan et al, 2021), NSP3 and the nucleocapsid protein N (PDB: 7PKU) (Bessa et al, 2022; Jiang et al, 2021) and the homodimerization of NSP8 (PDB: 7EIZ) (Yan et al, 2021), ORF3a (PDB: 6XDC) (Kern et al, 2021), N (PDB: 6VYO), the membrane glycoprotein M (Savitt et al, 2021; Yuan et al, 2022), and the well-established homodimerization of the spike protein (S) (PDB: 6VYB, for example) (Walls et al, 2020). We also detected the NSP7 homodimerization, which was previously

described by two independent studies (Yin et al, 2020; Wilamowski et al, 2021). In addition, we confirmed the known interactions of NSP3 and N (Jiang et al, 2021), the homodimerization of the envelope protein E (Mandala et al, 2020; Li et al, 2021), and its interaction with the membrane glycoprotein M (Savitt et al, 2021; Yuan et al, 2022). Overall, 91 previously reported interactions obtained from the IMEx database (Orchard et al, 2014), of which 21 were recently found to interact by Y2H (Kim et al, 2022), were not among the high-confidence (>95% probability) interactions (Dataset EV8). High-confidence interactions detected with LuTHy that were not previously reported (Orchard et al, 2014; Kim et al, 2022; Perfetto et al, 2020; Toro et al, 2021) include the heterodimerization of the envelope protein E with NSP6 and ORF7a, as well as between M and ORF3a, NSP12 and NSP16, NSP15 and NSP16, NSP2 and NSP3, NSP4 and NSP14, NSP4 and ORF7b, NSP6 and ORF7a, ORF3b and ORF8, ORF3a and ORF7a, ORF3a and NSP9, ORF3b and NSP14, ORF6 and NSP12, and the NSP4 homodimer (Fig. EV3A; Dataset EV9). For validation, we selected 8 of the newly identified SARS-CoV-2 interacting pairs and performed mN2H assays to confirm their interactions. Similar to the LuTHy SARS-CoV-2 data, the obtained mN2H interaction data was classified using the maSVM models trained on the hsPRS-v2 and hsRRS-v2 (Fig. EV3B,C). Thereby, we were able to validate 5 out of the 8 interactions with >95% probability, strengthening the confidence in our results (Fig. EV3D,E).

## Predicting the structure of SARS-CoV-2 PPI complexes using AlphaFold-Multimer

We next aimed to predict the complex structures of the LuTHy-identified interactions and to systematically validate their interaction probabilities using our AFM-based PPI classifier. We first used AFM to obtain structures for 23 out of the 34 LuTHy-positive, high-confidence interactions (Dataset EV9) and then employed PDBePISA to determine the iA and ΔG values for each of the complexes (Fig. EV3F). Similar to before, we used k-means clustering to identify the region with the lowest average inter-subunit PAE, which we suggest is most likely participating in the interaction (Fig. EV3F). We then used the AFM classifier models that were trained on the hsPRS-AF and hsRRS-AF pairs, to predict the classification probabilities of the 23 AFM SARS-CoV-2 complex structures (Figs. 4H and EV3G). Thereby, we validated 15 of the 23

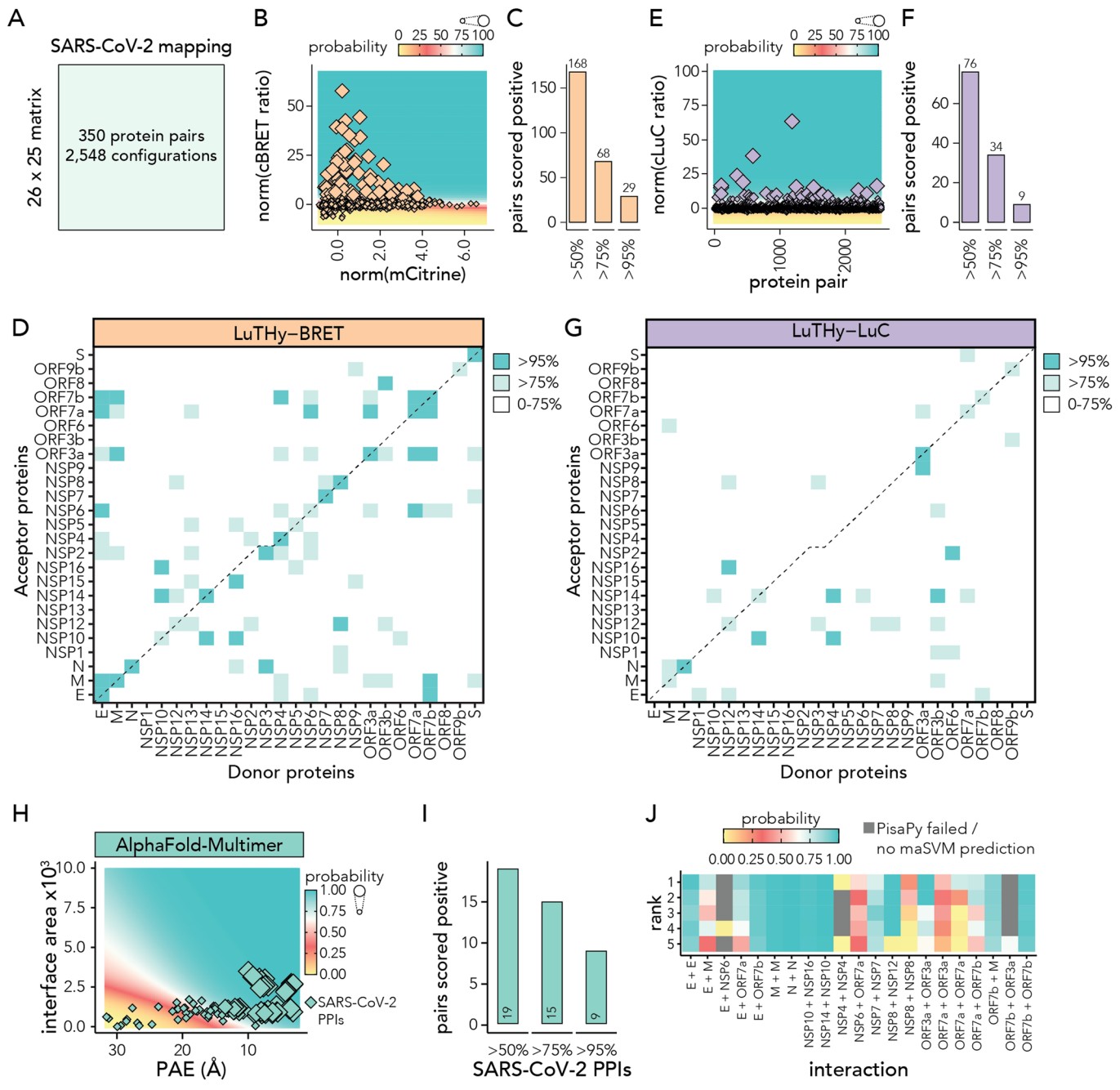

**Figure 4. Mapping binary interactions between SARS-CoV-2 proteins.**

(A) Search space between SARS-CoV-2 proteins tested by LuTHy. (B,E) Scatter plots showing (B) log-transformed and normalized in-cell mCitrine expression (x axis) against normalized cBRET ratios (y axis) or (E) number of protein pairs (x axis) against log-transformed and normalized cLuC ratios (y axis) for SARS-CoV-2 (orange) protein pairs from all eight tagging configurations. Average classifier probability from the 50 maSVM models is displayed as the size of the data points and as a colored grid in the background. (C, F) Bar plots showing the fraction of hsPRS-v2 and hsRRS-v2 protein pairs that scored above classifier probabilities of 50%, 75%, or 95% by (C) LuTHy-BRET or (F) LuTHy-LuC. Only the highest classifier probability per tested tagging configuration is considered. (D, G) Tile plots showing SARS-CoV-2 protein pairs with >95% and >75% classifier probability detected with (D) LuTHy-BRET and (G) LuTHy-LuC. Only the highest classifier probability per tested tagging configuration is shown. All LuTHy experiments were performed in HEK293 cells two times, with n = 2 biological replicates, each containing n = 3 technical replicates. (H) Scatter plot showing PAE (x axis) against interface area (y axis) for 23 SARS-CoV-2 protein pair structures predicted with AlphaFold-Multimer, with n = 5 models each. Average classifier probability from the 100 maSVM models trained on the hsPRS-AF and hsRRS-AF (Fig. 2F) is displayed as the size of the data points and as a colored grid in the background. (I) Bar plots showing the number of AlphaFold-Multimer predicted SARS-CoV-2 protein pair structures that scored above classifier probabilities of 50%, 75%, and 95%. (J) Heatmaps showing the classifier probabilities for the AFM-predicted SARS-CoV-2 protein pair structures. Source data are available online for this figure.

LuTHy-positive SARS-CoV-2 PPIs with a classification probability >75% and nine with a probability >95%. (Fig. 4I,J). We further wanted to investigate whether the binding free energy difference of the predicted complex structures correlated with the in-cell binding strength of the detected PPIs. Therefore, we performed LuTHy-BRET donor saturation experiments for 16 of the AFM-predicted SARS-CoV-2 structures and determined the $BRET_{50}$ value for the interacting proteins (Appendix Fig. S7). Interestingly, we observed a significant correlation between the $BRET_{50}$ and the $\Delta G$ values (Fig. EV3H), which is in line with our previously published results showing that the $BRET_{50}$ value is directly correlated to the dissociation constant ($K_D$) of the respective PPI resembling their binding affinity (Trepte et al, 2018). Furthermore, seven of the nine AFM-predicted complex structures with a probability >95% were also experimentally resolved (Fig. EV3I), such as for example, the heterodimerization between NSP10 and NSP16, which supports our classification approach for AFM-predicted structural models.

## Targeting the NSP10-NSP16 interaction interface by virtual screening

To directly apply our results to a relevant application, we next wanted to target one interaction interface by virtual screening. To maximize success rate and prioritize between the predicted structures we applied the following criteria: (i) inhibition of one complex member was previously shown to affect viral replication or function, and (ii) the 3D complex structure was both experimentally solved and AFM-predicted. Based on this rationale, we selected the interaction between NSP10 and the NSP16 RNA methyltransferase (MTase). Importantly, it was reported that inhibiting this interaction is able to completely abrogate the MTase activity of NSP16 (Chen et al, 2011), which is required to ensure normal viral replication (Daffis et al, 2010).

Overall, the five AFM-predicted complex structures showed very low predicted aligned errors (Fig. 5A) and a high overlap to the published 3D structure (Rosas-Lemus et al, 2020) (Fig. 5B). We used PDBePISA to determine the interaction hot spots (Clackson and Wells, 1995), i.e., the interface residues that contribute most to the binding, and identified lysine 93 of NSP10 and aspartate 106 of NSP16 having the lowest $\Delta G$ (Fig. 5C,D). We then performed site-directed mutagenesis and introduced charged changes at lysine 93 of NSP10 by substituting it with glutamic acid (Lys93Glu), and at aspartate 106 of NSP16 by substituting it with lysine (Asp106Lys). Both charged residue changes resulted in a strong reduction of the interaction between NSP10 and NSP16 as measured by LuTHy-BRET donor saturation assays (Fig. 5E). Importantly, we did not observe an effect of the point mutations on expression levels of NSP10-NL or mCit-NSP16, suggesting that the overall stabilities of the proteins were not affected (Fig. EV3J,K). This confirmed that Lys93 and Asp106 are critical hot spot residues in the NSP10-NSP16 interface, which is consistent with published results (Hamre and Jafri, 2022; Lugari et al, 2010) and make this contact site a promising target for the identification of PPI modulators.

Based on these results, we decided to target this specific region at the NSP10-NSP16 interface with small molecules using Virtual-Flow, a highly versatile open-source platform for ultra-high-throughput virtual compound screening (Gorgulla et al, 2020). We chose the NSP10 interface as the primary target site since the geometry at the critical hot spot residue site appeared to be a better candidate for small-molecule binding after visual inspection. We placed the virtual screening target area, i.e., the docking box, on NSP10 at the interaction interface with NSP16 comprising lysine 93 (Fig. 5F) and screened ~350 million compounds from the Enamine REAL library (Fig. 5G) using VirtualFlow and the docking program Quick Vina 2 (Alhossary et al, 2015). Among the top 100 virtual screening hits, we obtained comparable docking scores as previously described for similar groove-shaped target regions (Gorgulla et al, 2021), which suggested high-quality results (Fig. 5H). The top ~10 million (0.03%) hits were re-docked using VirtualFlow with AutoDock Vina (Trott and Olson, 2010) and Smina Vinardo (Quiroga and Villarreal, 2016), allowing 12 amino acid residues at the binding interface to be flexible. Finally, we selected compounds among the top 10,000 virtual hits that were re-docked with the two different approaches and subjected ~2000 molecules to chemical clustering and filtering (see "Methods" for details). A total of 20 representative molecules were selected, among which 15 were successfully synthesized and used for follow-up studies.

## Inhibiting the NSP10-NSP16 interaction reduces SARS-CoV-2 replication

To prioritize between the 15 selected compounds, we tested their abilities to inhibit the MTase activity of the NSP10-NSP16 complex in vitro (Bouvet et al, 2010; Decroly et al, 2011). We therefore incubated the purified NSP10-NSP16 complex with a Cap-0 RNA (m7G, N7-methyl guanosine) and monitored the methylation on the initiating nucleotide, which would generate a Cap-1 structure (Fig. 6A). Among the 15 selected compounds, three showed a significant reduction in the NSP10-NSP16 MTase activity compared to the DMSO control ($P < 0.05$), of which compound 459 had the strongest effect (Fig. 6B) with about 50% enzyme inhibition and was thus selected for further investigation.

To confirm the binding of the virtually docked compound 459 to the NSP10 protein (Fig. 6C,D), we next applied a microscale thermophoresis (MST) assay, which monitors the temperature-induced movement of fluorescently labeled molecules (Seidel et al, 2012) (Fig. 6E). To that end, we fluorescently labeled purified NSP10 protein and monitored its movement upon non-fluorescent compound addition. From the MST traces (Fig. 6F), we calculated the fraction of bound compound 459, which allowed us to determine a binding affinity of ~12.97 µM (Fig. 6G). To confirm that compound 459 could disrupt the NSP10-NSP16 interaction, we tested its effect in cells using the LuTHy-BRET assay. When incubating cells that express NL-NSP10 and mCit-NSP16 with compound 459, we observed a modest but significant concentration-dependent reduction in the BRET ratio with a half-maximal inhibitory concentration ($IC_{50}$) of 9.2 µM (Fig. 6H). This result indicated that the compound inhibits the binding of the two proteins in live cells. Since it was previously shown that normal MTase activity is required to ensure proper viral proliferation (Daffis et al, 2010), we evaluated the effect of compound 459 on SARS-CoV-2 replication using an infectious cDNA clone-derived reporter assay (Hou et al, 2020; Kim et al, 2022). We observed a concentration-dependent decrease of the luminescence signal in the SARS-CoV-2 replication assay, indicating an inhibition of viral replication with an $IC_{50}$ of 39.5 µM (Fig. 6I). Importantly, cell viability was not affected by treatment with compound 459 (Appendix Fig. S8).

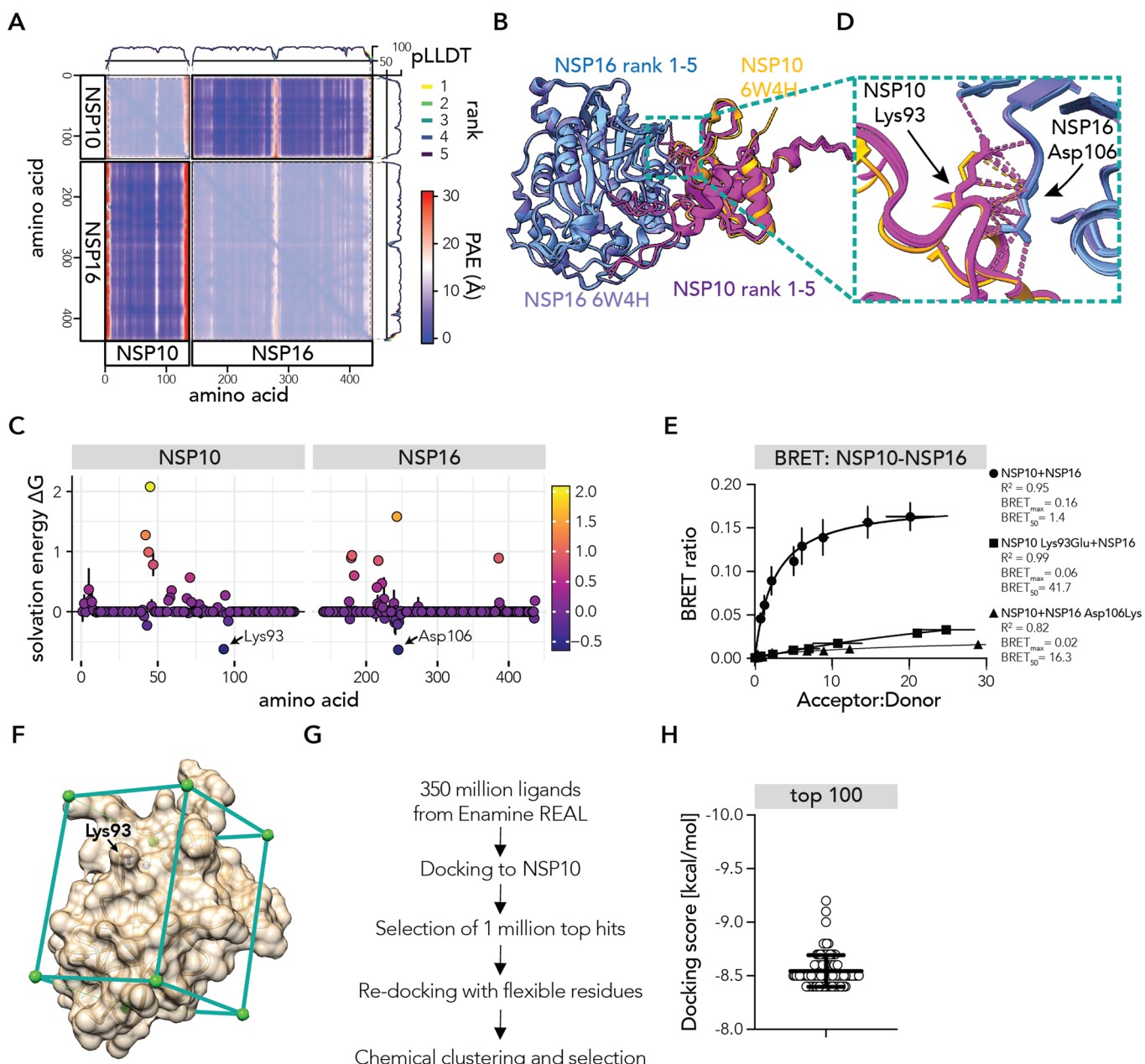

**Figure 5. Predicting the NSP10-NSP16 PPI complex with AFM to target the interaction interface by ultra-large virtual drug screening.**

(A) Heatmap showing the predicted alignment error (PAE) of the AlphaFold-Multimer predicted NSP10-NSP16 complex for the rank 1 model. The intra-molecular PAEs are shown with 50% opacity. The predicted local distance difference test (pLDDT) for all five predicted models (rank 1–5) are shown as line graphs on top and on the right of the heatmap. (B) The five models of the AlphaFold-Multimer predicted NSP10-NSP16 complex and the published crystal structure (PDB: 6W4H) are shown. Structures were overlaid using the "matchmaker" tool of ChimeraX. (C) Scatter plot showing for each amino acid (x axis) the solvation-free energy ($\Delta G$, y axis, fill color) upon formation of the interface, in kcal/mol, as determined by PDBePISA. Dots represent the mean $\Delta G$ for the five predicted models and error bars correspond to the standard deviation from $n = 5$ AFM-predicted structural models. The x axis indicates the amino acid positions of the whole complex structure starting from NSP10's N-terminus and ending with NSP16's C-terminus. Lysine 93 (Lys93) of NSP10 and aspartate 106 (Asp106) of NSP16, which showed the strongest solvation-free energy gain upon complex formation, are indicated, respectively. (D) Zoom-in into the NSP10-NSP16 complex showing the contacts of NSP10's Lys93 and NSP16's Asp106 as determined using ChimeraX, using the Contacts tool with the parameters, "VDW overlap ≥ -0.40 Å", "Limited by selection: with at least one selected" of NSP10 Lys93 and NSP16 Asp106; "Include intramodel"; "Display as pseudobonds". (E) LuTHy-BRET donor saturation assay, where constant amounts of NSP10-NL WT or Lys93Glu are co-expressed with increasing amounts of mCitrine-NSP16 WT or mCitrine-NSP16 Asp106Lys. Nonlinear regression was fitted through the data using the "One-Site – Total" equation of GraphPad Prism. Data points represent mean values from two $n = 2$ (NSP10 + NSP16, NSP10 Lys93Glu + NSP16) or $n = 4$ (NSP10 + NSP16 Asp106Lys) biological replicates each containing $n = 2$ technical duplicates. (F) Docking box on the NSP10 structure (PDB: 6W4H) used for the ultra-large virtual screen. (G) Schematic overview of the workflow of the virtual docking screen using VirtualFlow. (H) Docking scores of the top 100 molecules identified by virtual screening. The horizontal line indicates mean docking score and error bars the standard deviation, with the virtual screen performed once ($n = 1$). Source data are available online for this figure.

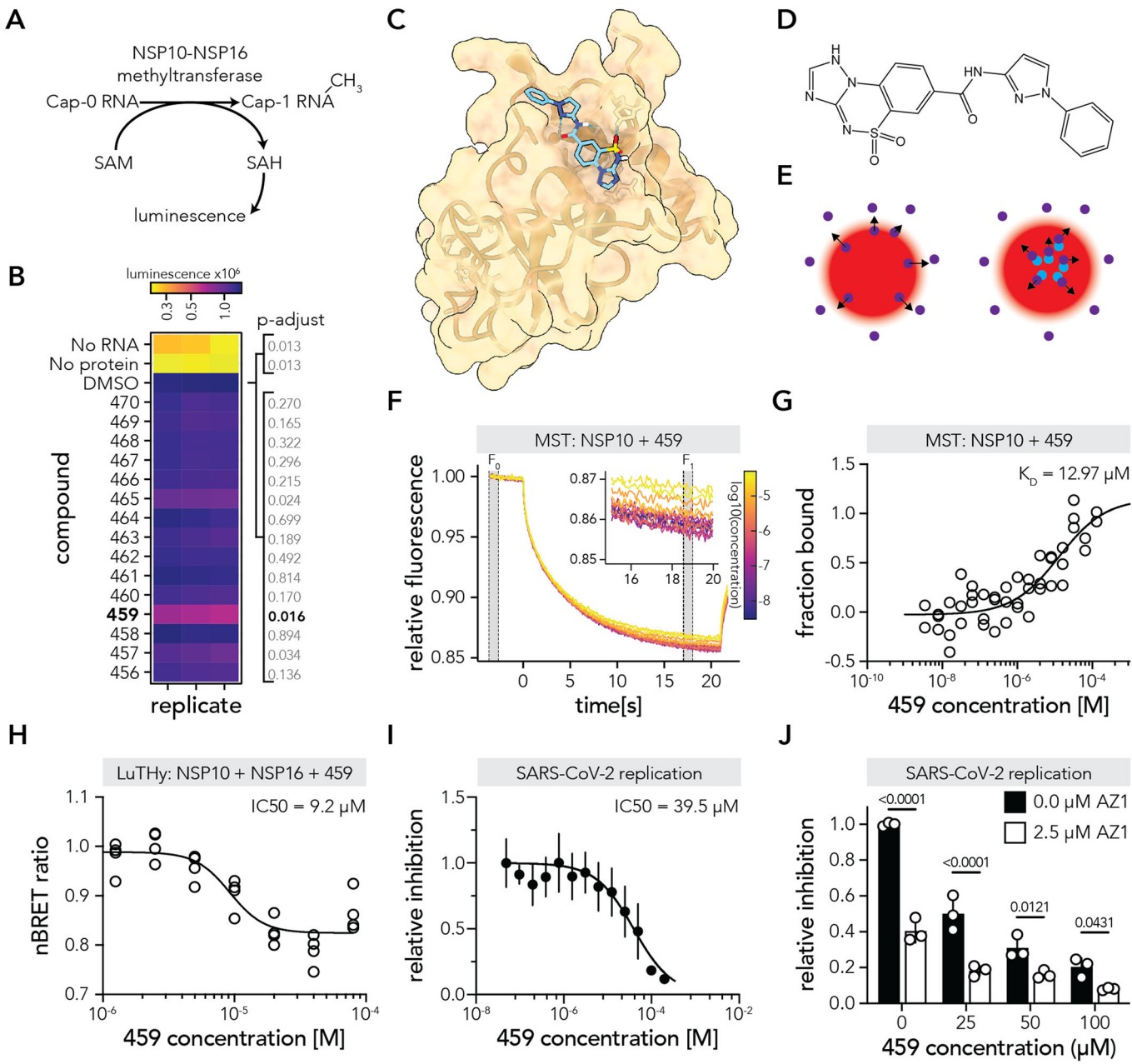

Finally, we investigated if NSP10-NSP16 inhibition by compound 459 would confer additive effects upon combination with AZ1, an enzymatic inhibitor of the human ubiquitin-specific peptidase 25 (USP25) (Wrigley et al, 2017). AZ1 was reported to impair SARS-CoV-2 replication, and USP25 was identified as an interactor of NSP16 (Kim et al, 2022). We assessed SARS-CoV-2 viral replication upon treatment with 2.5 µM AZ1 and increasing concentrations of compound 459, and observed an additive, concentration-dependent effect of the two molecules (Fig. 6J). While both molecules target NSP16-related functions, it is unknown whether compound 459 and AZ1 act in identical or distinct ways to block the viral replication. Our results indicate that 459 and AZ1 affect viral replication additively and that such combinatorial therapies could potentially improve the efficacy of treatments using small-molecule drugs.

## Discussion

### Support vector machine learning for PPI classifications

Targeting PPIs offers great opportunities to tackle various diseases, but it remains a great challenge to reliably identify and modulate protein complexes. To improve comparisons between binary PPI datasets generated in different experiments and laboratories and confidently prioritize potential targets for PPI drug discovery, we have utilized a maSVM learning algorithm (Chang and Lin, 2011; Yang et al, 2017) to coherently score interactions of quantitative PPI datasets. Traditional approaches involve (i) selecting a cutoff of maximal specificity, i.e., at which none of the random pairs used as negative controls are scored positive (Choi et al, 2019), (ii) ROC

**Figure 6.   Compound 459 inhibits the NSP10-NSP16 interaction and reduces SARS-CoV-2 replication.**

(A) Schematic overview of the NSP10-NSP16 methyltransferase (MTase) assay. (B) Heatmap showing the result of the MTase activity of the NSP10-NSP16 complex in the absence or presence of 100 μM of the top 15 compounds. Statistical significance was calculated with a kruskal-wallis test ($P$ value = 9.7e-5, chi-squared = 47.656, df = 17, the experiment was performed once with $n = 3$, technical replicates), followed by a post hoc Dunn test and adjusted p-values are shown. (C) Compound 459 docked onto the NSP10 structure (PDB: 6W4H). (D) Chemical structure of compound 459. (E) Assay principle of the microscale thermophoresis (MST) assay. The fluorescence intensity change of the labeled molecule (purple) after temperature change induced by an infrared laser (red) is measured. The binding of a non-fluorescent molecule (blue) can influence the movement of the labeled molecule. (F) Representative MST traces of labeled NSP10 and different concentrations of unlabeled compound 459. The bound fraction is calculated from the ratio between the fluorescence after heating ($F_1$) and before heating ($F_0$). (G) Scatter plot showing the 459 concentration ($x$ axis) against the fraction of 459 bound to NSP10 ($y$ axis). Nonlinear regression was fitted through the data using the "One-Site – Total" equation of GraphPad Prism (the experiment was repeated three times with $n = 3$, biological replicates). (H) Scatter plot showing the 459 concentration ($x$ axis) against the normalized BRET ratio (nBRET ratio) for the interaction between NSP10-NL and mCit-NSP16 measured in HEK293 cells. Nonlinear regression was fitted through the data using the "log(inhibitor) vs. response–Variable slope (four parameters)" equation of GraphPad Prism (the experiment was repeated four times with $n = 4$, biological replicates, each containing $n = 3$ technical replicates). (I) Scatter plot showing the 459 concentration ($x$ axis) against the relative luminescence measured from icSARS-CoV-2-nanoluciferase in HEK293-ACE2 cells. Nonlinear regression was fitted through the data using the "log(inhibitor) vs. normalized response" equation of GraphPad Prism (the experiment was repeated three times, with $n = 3$ for 0.1, 100, 200 μM; $n = 6$ for 0.2, 0.4 μM; all other $n = 9$; all biological replicates; error bars represent the standard deviation). (J) Barplot showing the relative luminescence measured from icSARS-CoV-2-nanoluciferase in HEK293-ACE2 cells upon incubation with 0, 25, 50, or 100 μM of compound 459 together with 2.5 μM AZ1 or without AZ1 (0.0 μM). Statistical significance was calculated in GraphPad Prism by a "two-way ANOVA", where each cell mean was compared to the other cell mean in that row using "Bonferroni's multiple comparisons test" (the experiment was repeated three times, with $n = 3$, biological replicates; error pars represent the standard deviation; source of variation: 57.91% 459 concentration, $P < 0.0001$; 28.33% AZ1 concentration, $P < 0.0001$; 11.40% 459/AZ1 interaction, $P < 0.0001$).

analyses with cutoffs at selected false-positive rates of ~5% (Yao et al, 2020), ~3% (Trepte et al, 2015), or ~1–2% (Trepte et al, 2018; Cassonnet et al, 2011), and (iii) distribution-based cutoffs (Taipale et al, 2012). With the maSVM approach, we show for the first time that a uniform and unbiased approach can be used for the automatic classification of large quantitative interaction datasets with high-confidence. Notably, such a classification strategy has been successfully used to predict kinase substrates from phospho-proteomics data (Yang et al, 2019; Kim et al, 2021), or to classify cell types from single-cell RNA-sequencing (Abdelaal et al, 2019). We show that the maSVM algorithm can provide probabilities for being a true interaction for every protein pair tested in a given assay. For example, we observed that in the LuTHy-BRET, mN2H and yN2H assays, one pair from the RRS hsRRS-v2 (SLC6A1 + TM4SF4) was classified as a true-positive interaction with >99% probability and in the LuTHy-LuC and KISS assays with 91.9% and 94.7% probability, respectively (Fig. 1J, black arrow). Due to its definition as a negative or random interaction, a traditional scoring approach might have classified this pair as strictly negative and thus classified similar or lower-scoring pairs also as negative. However, it is very likely that in these assays, the two proteins interact biophysically when overexpressed in HEK293 cells, making the pair a potential pseudo-interaction, i.e., a true biophysical interaction without in vivo biological relevance (Braun et al, 2009). The maSVM algorithm is able to deal with such exceptions and thus increases robustness in scoring PPIs even when inconsistencies among assays or in reference sets are present. We also show that the maSVM algorithm is universally applicable to classify binary PPIs from different quantitative datasets, including the interface analysis of AFM-predicted complex structures. However, it is required to fine-tune hyperparameters for each assay. Overall, the maSVM provides a framework to directly compare the results of various binary interaction assays and in silico predictions, which will lead to increased reproducibility and interpretability of results between experiments and methods.

## Considerations and limitations of the maSVM algorithm

A universal scoring approach should also be able to deal with the fact that not all positive interactions will score positive in each of the

tested configurations. In binary interaction assays, such as LuTHy, N2H and others, the protein pairs of interest are usually tested in different tagging configurations since e.g., distance-sensitive readouts such as LuTHy-BRET, or precipitation-based readouts such as LuTHy-LuC can be highly dependent on the configuration of the tags. Using multi-adaptive sampling, the negative results obtained in certain configurations from interactions in the PRS are automatically relabeled and thus do not negatively affect the overall performance of the classifier. Similarly, potential false-positive scores from single configurations are relabeled in the training process. However, it is obvious that the maSVM-based scoring approach is also influenced by assay artifacts and will hence "learn" those. If e.g., certain configurations always tend to produce larger scores than others, training of a configuration-specific classifier could be considered, which is then only used for the analysis of results for the selected assay configuration. Alternatively, datasets of different configurations could be normalized similarly to the construct normalization before training to obtain a similar dynamic range in the training and test datasets, respectively.

Importantly, the maSVM scoring approach is not limited in its application to PPI datasets from binary interaction assays. As mentioned above, it is a general approach for classification tasks and thus can be applied basically to any systems biology approach when classification is warranted. However, since SVMs are supervised learning algorithms that are trained with labeled training data to create a decision boundary (Cortes and Vapnik, 1995), a prerequisite is that labeled data of true positives and true negatives is available or can be generated (e.g., from tested positive and negative reference sets). One other PPI mapping technique, to which the maSVM algorithm presented here could also be applied, is e.g., cross-linking mass spectrometry (XL-MS) (Lenz et al, 2021; Giese et al, 2021), in which a SVM algorithm could be trained with known positive and negative interactors and then could be used for the classification of newly obtained data. Obviously, this is not limited to PPI XL-MS datasets, but can be also applied for protein-nucleic acids interactions from DNA/RNA XL-MS datasets (Sarnowski et al, 2022). In addition, to generate a robust classifier, the training set data should ideally reflect a similar variability as it is expected in the test dataset. This could e.g., be achieved by including a set of true-positive and true-negative reference interactions in every PPI screen. Alternatively, thorough data

normalization can be performed on the interaction scores before training the classifier.

Generally, the better the labeled training data resembles the test dataset, the better the prediction of true positives by the classifier. For PPI datasets this could mean e.g., that for screening a set of membrane proteins for interactions, ideally also a reference set of positive and negative interactions among membrane proteins is used for training. Similarly, if it is known that a certain assay yields e.g., lower or higher scores for interactions in a specific compartment, then it is favorable for the performance of the SVM classifier that the reference set also contains such subcompartment interactions.

## Identification and targeting of SARS-CoV-2 PPIs

In this study, we have applied the novel scoring approach to map and score binary PPIs in three established multiprotein complexes, and to identify PPI targets for drug discovery among SARS-CoV-2 proteins. Interestingly, due to lower mutation frequency and the high amino acid conservation of interaction interfaces (Guharoy and Chakrabarti, 2010; Gupta et al, 2020), PPI-targeting drugs could provide unique advantages over other types of drugs such as vaccines or antivirals circumventing mutations in the pathogens' genomes that can result in immune evasive properties. Of the 34 detected high-confidence (≥95% probability) SARS-CoV-2 PPIs we detected here, 19 are known interactions according to the IMEx database (Orchard et al, 2014), while 15 were not reported before (Dataset EV9). Further characterizing these previously undescribed interactions and understanding their biological functions could result in the identification of novel drug targets for SARS-CoV-2. In particular, our approach helped in prioritizing the NSP10-NSP16 interaction, where NSP10 serves as a cofactor for NSP16's MTase activity (Decroly et al, 2011), an enzyme crucial to single-stranded RNA viruses (Ramdhan and Li, 2022). This enzymatic complex has been a target of previous drug screening campaigns (Nencka et al, 2022) and peptides inhibiting the interaction could be successfully identified (Wang et al, 2015; Ke et al, 2012). Even though the experimental structure of the SARS-CoV-2 methyltransferase complex became available during the course of this study (Rosas-Lemus et al, 2020) and was already described for SARS-CoV-1 (Chen et al, 2011), we demonstrate that our pipeline of AI-guided experimental PPI mapping, structure prediction and experimental validation is able to define such protein complexes with high-confidence and to identify hot spots on their interaction interfaces. Furthermore, we demonstrate the use of this information in a subsequent virtual PPI inhibitor screening strategy with Virtual-Flow, an in silico method which allows to virtually screen billions of compounds and assess their different binding poses on the targeted surface area. VirtualFlow was already used to target 17 SARS-CoV-2 proteins, including the NSP10-NSP16 interface (Gorgulla et al, 2021). Here, we targeted the same interaction interface by virtual screening but further experimentally validated hit compounds for enzymatic inhibition of the NSP10-NSP16 protein complex and binding of hit compound 459 to the NSP10 target site. Interestingly, the predicted target binding site of compound 459 is similar to the target binding site of one of the top hits identified by Gorgulla et al (Gorgulla et al, 2021; https://vf4covid19.hms.harvard.edu/, screen ID: 25). It is also predicted to bind closely to lysine 93 of NSP10, however, the identified molecular scaffold is different from

compound 459 identified in this study. We also show that this compound inhibits the NSP10-NSP16 interaction and prevents SARS-CoV-2 replication with additive effects when combined with AZ1, a human USP25 inhibitor disrupting SARS-CoV-2 replication (Kim et al, 2022). Interestingly, the predicted target binding site of compound 459 on the NSP10 protein (Fig. 6C) is highly conserved among coronavirus groups (Lugari et al, 2010) and thus could potentially also inhibit the replication of other viruses belonging to the Coronaviridae family. However, despite its effects, compound 459 is an experimental compound with micromolar affinity that will require extensive optimization to improve its chemical scaffold and associated affinity and efficacy for further investigations.

Targeting viral–viral PPIs instead of virus–host PPIs can be advantageous. First, determining viral–viral compared to viral–host interactions is much simpler both experimentally and computationally. To map the SARS-CoV-2 interactome, 650 pairwise combinations of viral protein pairs were searched. In comparison, to identify virus–host PPIs, Kim et al searched 26 viral proteins against 17,472 human ORFs, which constituted the search space of 454,272 pairwise combinations. Second, Kim et al targeted one human-virus interaction, by inhibiting human USP25 with a small molecule. As USP25 is a human protein involved in protein degradation and cellular homeostasis, blocking its enzymatic activity could cause unintended effects in addition to its intended therapeutic benefits for blocking the viral replication. Therefore, inhibiting viral–viral PPIs could offer a safer and more specific strategy to selectively intervene with viral replication. Finally, viral–viral PPIs are often evolutionarily conserved, so that therapeutics targeting viral–viral interactions could potentially remain effective despite viral evolution.

A prerequisite to enable virtual drug and PPI inhibitor screenings as presented here is the availability of high-resolution protein and protein complex structures. Protein and protein complex structure predictions have exploded in the last few years with the development of AlphaFold and RoseTTAFold that allow structure predictions of entire proteomes (Jumper et al, 2021; Baek et al, 2021), as well as the prediction of protein complexes and PPIs (Humphreys et al, 2021; Burke et al, 2021; Evans et al, 2022; Gao et al, 2022; Bryant et al, 2022). Through ColabFold, which combines such algorithms with the fast homology search MMseqs2, the immense computing power needed was reduced and is now available within the Google Colaboratory, which makes protein structure prediction accessible to all (Mirdita et al, 2022). As current approaches have already predicted tens of thousands of interactions (Burke et al, 2021; Humphreys et al, 2021), it seems feasible to predict complex structures of the entire theoretical SARS-CoV-2 binary interactome, i.e., all 26 proteins against each other for a total of 650 pairwise combinations. While we were limited to structure prediction-based PPI mapping for complexes with up to 1400 amino acids, further improvements in the structure prediction algorithms regarding speed, memory usage and the maximal size of proteins and protein complexes are already available (Ahdritz et al, 2022). This will also enable virtual PPI mapping, including the approach presented here, for large protein complexes. Additional improvements in structure prediction of low complexity domains of proteins, membrane protein complexes, protein–DNA and –RNA complexes and the consideration of environmental factors will further improve and expand the scope of

such structure prediction-based PPI mappings. Overall, combining in silico and wet lab techniques for both the identification and validation of PPIs as well as for drug screening and validation of drug effects, should help to speed up the process of developing PPI-modulating therapeutics.

# Methods

## Reagents and tools

See Table 1 for a complete list of all reagents and resources.

**Table 1.  Reagents and tools.**

| Reagent/resource | Reference or source | Identifier or catalog number |
|---|---|---|
| **Experimental models** | | |
| HEK293 cells (*H. sapiens*) | ATCC | CRL-1573 |
| HEK293T cells (*H. sapiens*) | ATCC | CRL-3216 |
| **Recombinant DNA** | | |
| pDONR221 | ThermoFisher | 12536017 |
| pDONR223 | Rual et al, 2004 | – |
| pcDNA3.1(+) | ThermoFisher | V79020 |
| pDEST | Choi et al, 2019 | – |
| pcDNA3.1 LuTHy destination and control vectors | Addgene (Trepte et al, 2018) | 113442–113449 |
| pDEST-N2H destination and control vectors | Addgene (Choi et al, 2019) | 125547–125549; 125551–125552; 125559 |
| The CCSB Human ORFeome Collection (pDONR223) | Dana-Farber Cancer Institute | 8.1 |
| SARS-CoV-2 entry plasmids (pDONR223) | Addgene (Kim et al, 2020) | 149304–149312; 149314–149315; 149317; 149320–149321; 149323–149327; 152987–152988; 149322 |
| pDONR221 NSP10 WT | This study | CS683 |
| pDONR221 NSP10 Lys93Glu (K93E) | This study | CS916 |
| pDONR221 NSP16 WT | This study | CS688 |
| pDONR221 NSP16 Asp106Lys (D106K) | This study | CS1048 |
| pDONR221 ORF3A | This study | CS682 |
| **Oligonucleotides and sequence-based reagents** | | |
| Primer NSP16 Asp106Lys (D106K) FWD | 5'-CTTCGTGTCCaagGCCGACAGCA-3' | CS567 |
| Primer NSP16 Asp106Lys (D106K) REV | 5'-TCGTTCAGGTCGCTGTCC-3' | CS569 |
| RNA cap-0 oligo (MTase substrate) | 5'-(N7-MeGppp)ACAUUUGCUUCUGAC-3' | – |
| **Chemicals, enzymes, and other reagents** | | |
| Gateway™ LR Clonase™ II Enzyme mix | ThermoFisher | 11791100 |
| DMEM high glucose | ThermoFisher | 41965062 |
| Fetal bovine serum (FBS) | ThermoFisher | 10270106 |
| Linear polyethylenimine (PEI), MW 25000 | Polysciences | 23966 |
| Linear polyethylenimine (PEI), MW 40000 | Polysciences | 24765 |
| Cell culture microplate, 96-well white | Greiner | 655983 |
| High binding microplate, 384-well white, small volume | Greiner | 784074 |
| DPBS | ThermoFisher | 14190169 |
| Coelenterazine-h | pjk | 102182 |
| Nano-Glo | Promega | N1120 |
| Benzonase | Merck Millipore | 70664-3 |

**Table 1.** (continued)

| Reagent/resource | Reference or source | Identifier or catalog number |
|---|---|---|
| cOmplete protease inhibitor cocktail (EDTA-free) | Roche/Sigma-Aldrich | COEDTAF-RO |
| Sheep gamma globulin | Jackson ImmunoResearch | #013-000-002 |
| Rabbit anti-sheep gamma globulin | Jackson ImmunoResearch | #313-005-003 |
| **Software** | | |
| R | www.r-project.org | 4.2.1 |
| R package 'e1071' | CRAN.R-project.org/package=e1071 | 1.7-11 |
| R package 'binary PPI classifier' | This study (github.com/philipptrepte/binary-PPI-classifier) | 1.5.5.7 |
| R package 'AFM PISA classifier' | This study (github.com/philipptrepte/AFM-Pisa-classifier) | 1.0.0.0 |
| R studio | | 2022.07.0 |
| GraphPad Prism | graphpad.com | 7, 8, 9 |
| SerialCloner | serialbasics.free.fr/Serial_Cloner.html | 2-6-1 |
| AlphaFold-Multimer | github.com/google-deepmind/alphafold (Evans et al, 2022) | v2 |
| ColabFold | github.com/sokrypton/ColabFold (Mirdita et al, 2022) | 1.2.0 or 1.3.0 |
| PDBePISA | www.ebi.ac.uk/pdbe/pisa/ | 1.48 |
| PisaPy | github.com/hocinebib/PisaPy | latest |
| VirtualFlow for Virtual Screening | github.com/VirtualFlow/VFVS | vfvs-1 |
| **Databases** | | |
| IMEx | www.ebi.ac.uk/intact/imex (Orchard et al, 2014) | Last access: 2023-01-20 |
| PDB/RCSB | rcsb.org | |
| Interactome Insider | interactomeinsider.yulab.org (Meyer et al, 2018) | Last access: 2020-11-03 |
| VirtualFlow Ligand Library (Enamine REAL Database) | virtual-flow.org/real-library enamine.net/compound-collections/real-compounds | – |
| **Other** | | |
| Infinite Multimode readers | Tecan | M200/M1000/M1000 PRO/Spark |
| Freedom EVO platform | Tecan | 150/200 |
| Luminometers | Berthold | TriStar, Centro XS |

## Methods and protocols

### *ORF sequencing and plasmid generation*

For hsPRS-v2 and hsRRS-v2 proteins, the corresponding sequence-verified entry vectors published in Choi et al (Choi et al, 2019) (Table 1) were Gateway cloned into the different LuTHy destination plasmids. ORFs for subunits of the LAMTOR, BRISC and MIS12 complexes were taken from the CCSB human ORFeome 8.1, which is a sequence-confirmed clonal collection of human ORFs in a Gateway entry vector system (Yang et al, 2011). In total, 16 entry plasmids were picked from the collection, single clones were isolated, and ORFs were PCR-amplified and confirmed by bi-directional Sanger DNA sequencing. Entry clones were shuttled into LuTHy (Addgene #113446, #113447, #113448, #113449) and N2H (Addgene #125547, #125548, #125549, #125559) destination vectors using the Gateway Cloning Technology. SARS-CoV-2 ORF cDNA library was obtained from Kim et al (Kim et al, 2020) via Addgene. NSP10, NSP14, NSP16 and NSP10 mutant cDNA entry clones were generated by gene synthesis and subcloning into

pDONR221 (GeneArt, ThermoFisher Scientific). The NSP16 mutant entry plasmid was generated from the NSP16 wild-type (WT) plasmid by site-directed mutagenesis using the following primers: 5'-CTTCGTGTCCaagGCCGACAGCA and 5'-TCGTTCAGGTCGCTGTCC. All cDNA clones were sequence-verified and shuttled into LuTHy destination plasmids. All resulting vectors were analyzed by PCR-amplification of cloned ORFs and DNA gel electrophoresis (N2H plasmids), or restriction digestion and sequence validation (LuTHy plasmids). For the LuTHy assay, additional control plasmids (PA-NL, Addgene #113445; PA-mCit-NL, Addgene #113444; PA-mCit, Addgene #113443; NL, Addgene #113442) were used, as previously described (Trepte et al, 2018). For the mN2H assay, additional control plasmids (pDEST-N2H-F1-empty vector, Addgene #125551; pDEST-N2H-F2-empty vector, Addgene #125552) were used, as previously described (Choi et al, 2019). For validation of previously undescribed SARS-CoV-2 interacting pairs identified by LuTHy in this study, the corresponding ORFs of selected interactions were additionally cloned into pDEST-N2H-N1 or pDEST-N2H-N2 using the Gateway Cloning

Technology (no successful expression vector clone could be obtained for NSP3).

## LuTHy assay procedure

The LuTHy assay was performed as previously described (Trepte et al, 2018). In brief, HEK293 cells were reversely transfected in white 96-well microtiter plates (Greiner, #655983) at a density of $4.0\text{--}4.5 \times 10^4$ cells per well with plasmids encoding donor and acceptor proteins. After incubation for 48 h, mCitrine fluorescence was measured in intact cells (mCit$_{cell}$, Ex/Em: 500 nm/530 nm). For LuTHy-BRET assays, coelenterazine-h (pjk, #102182) was added to a final concentration of 5 µM (5 mM stock dissolved in methanol). Next, cells were incubated for an additional 15 min, and total luminescence as well as luminescences at short (370–480 nm) and long (520–570 nm) wavelengths were measured using the Infinite® microplate readers M200, M1000, or M1000 PRO (Tecan). After luminescence measurements, the luminescence-based co-precipitation (LuC) assay was performed. Cells were lysed in 50–100 µL HEPES-phospho-lysis buffer (50 mM HEPES, 150 mM NaCl, 10% glycerol, 1% NP-40, 0.5% deoxycholate, 20 mM NaF, 1.5 mM MgCl$_2$, 1 mM EDTA, 1 mM DTT, 1 U Benzonase, protease inhibitor cocktail (Roche, EDTA-free), 1 mM PMSF, 25 mM glycerol-2-phosphate, 1 mM sodium orthovanadate, 2 mM sodium pyrophosphate) for 30 min at 4 °C. Lysates (7.5 µL) were transferred into small volume 384-well microtiter plates (Greiner, #784074) and fluorescence (mCit$_{IN}$) was measured as previously described. To measure the total luminescence (NL$_{IN}$), 7.5 µL of 20 µM coelenterazine-h in PBS was added to each well, and the plates were incubated for 15 more minutes. For LuC, small volume 384-well microtiter plates (Greiner, #784074) were coated with sheep gamma globulin (Jackson ImmunoResearch, #013-000-002) in carbonate buffer (70 mM NaHCO$_3$, 30 mM Na$_2$CO$_3$, pH 9.6) for 3 h at room temperature, and blocked with 1% BSA in carbonate buffer before being incubated overnight at 4 °C with rabbit anti-sheep IgGs in carbonate buffer (Jackson ImmunoResearch, #313-005-003). In total, 15 µL of cell lysate was incubated for 3 h at 4 °C in the IgG-coated 384-well plates. Then, all wells were washed three times with lysis buffer and mCitrine fluorescence (mCit$_{OUT}$) was measured as described. Finally, 15 µL of PBS buffer containing 10 µM coelenterazine-h was added to each well and luminescence (NL$_{OUT}$) was measured after a 15 min incubation period. LuTHy experiments to screen hsPRS-v2/hsRRS-v2, LAMTOR, BRISC, MIS12, intra-complex, and SARS-CoV-2 protein pairs, were replicated twice in the laboratory accounting for the two biological replicates (different HEK293 freezings), with three technical replicates each that were arranged next to each other on the plate. HEK293 cells were regularly tested for mycoplasma contamination.

## LuTHy data analysis

Data analysis was performed as previously described (Trepte et al, 2018). In brief, the LuTHy-BRET and LuTHy-LuC ratios from BRET and co-precipitation measurements are calculated as follows:

$$BRET\ ratio = \frac{LWL}{SWL} - Cf \quad \text{with } Cf = \frac{LWL_{PA-NL}}{SWL_{PA-NL}} \quad (1)$$

with LWL and SWL being the detected luminescences at long (520–570 nm) and short (370–480 nm) wavelengths, respectively.

The correction factor (Cf) represents the donor bleed-through value from the PA-NL only construct. The corrected BRET (cBRET) ratio is calculated by subtracting the maximum BRET ratios of control 1 (NL/PA-mCit-Y), or of control 2 (NL-X/PA-mCit) from the BRET ratio of the studied interaction (NL-X/PA-mCit-Y).

For the LuC readout, the obtained luminescence precipitation ratio (PIR) of the control protein PA-NL (PIR$_{PA-NL}$) is used for data normalization, and is calculated as follows:

$$PIR_{PA-NL} = \frac{NL_{OUT}}{2 * NL_{IN}} \quad (2)$$

with NL$_{OUT}$ being the total luminescence measured after co-IP and NL$_{IN}$ the luminescence measured in the cell extracts, directly after lysis. Subsequently, LuC ratios are calculated for all interactions of interest, and normalized to the PIR$_{PA-NL}$ ratio:

$$LuC\ ratio = \frac{NL_{OUT}/2 * NL_{IN}}{PIR_{PA-NL}} \quad (3)$$

Finally, a corrected LuC (cLuC) ratio is calculated by subtracting either the LuC ratio of control 1 (NL/PA-mCit-Y), or of control 2 (NL-X/PA-mCit) from the LuC ratio of the studied interaction (NL-X/PA-mCit-Y). The calculated LuC ratios obtained for controls 1 and 2 are then compared to each other, and the highest value is used to correct the LuC ratio of the respective interaction. The described analysis was semi-automated, by using a Python script that copied the raw data from the Excel files generated by the Tecan plate readers into Excel templates that were manually controlled for missing values and outliers. Following, all Excel files were imported into R to calculate cBRET and cLuC ratios as described above.

## Mammalian cell-based version of the N2H assay (mN2H)

HEK293T cells were seeded at $6 \times 10^4$ cells per well in 96-well, flat-bottom, cell culture microplates (Greiner Bio-One, #655083), and cultured in Dulbecco's modified Eagle's medium (DMEM) supplemented with 10% fetal calf serum at 37 °C and 5% CO$_2$. Twenty-four hours later, cells were transfected with 100 ng of each N2H plasmid (pDEST-N2H-N1, -N2, -C1, or -C2) using linear polyethylenimine (PEI) to co-express proteins fused with complementary NanoLuc fragments, F1 and F2. The stock solution of PEI HCl (PEI MAX 40000; Polysciences Inc; Cat# 24765) was prepared according to the manufacturer's instructions. Briefly, 200 mg of PEI HCl powder were added to 170 mL of water, stirred until complete dissolution, and pH was adjusted to 7 with 1 M NaOH. Water was added to obtain a final concentration of 1 mg/mL, and the stock solution was filtered through a 0.22-µm membrane. The DNA/PEI ratio used for transfection was 1:3 (mass:mass). Twenty-four hours after transfection, the culture medium was removed and 50 µL of 100x diluted NanoLuc substrate (Furimazine, Promega Nano-Glo, N1120) was added to each well of a 96-well microplate containing the transfected cells. Plates were incubated for 3 min at room temperature. Luciferase enzymatic activity was measured using a TriStar or Centro XS luminometer (Berthold; 2 s integration time). Four technical replicates were generated for each protein pair when MN2H was used to map binary PPIs of the three protein complexes (Fig. 3).

### LuTHy-BRET donor saturation assays

LuTHy donor saturation assays were performed as previously described (Trepte et al, 2018). In brief, increasing acceptor expression plasmids were transfected to a constant amount of donor plasmids as described above (see "LuTHy procedure"). After 48 h, coelenterazine-h (pjk, #102182) was added to a final concentration of 5 µM. Cells were incubated for an additional 15 min and in-cell mCitrine and luminescence signals were quantified. Infinite® microplate readers M1000 or M1000Pro (Tecan) were used for the readouts with the following settings: fluorescence of mCitrine recorded at Ex 500 nm/Em 530 nm, luminescence measured using blue (370–480 nm) and green (520–570 nm) bandpass filters with 1000 ms (LuTHy-BRET). For data analysis, BRET ratios were calculated as described above. Acceptor-to-donor ratios were estimated by calculating the ratio of the fluorescence intensity of the acceptor to the total luminescence of the donor and normalization to the acceptor to donor signal intensities of the PA-mCit-NL tandem construct (Eq. (4)).

$$Acceptor : Donor\ ratio = \frac{mCit_{PPI}/NL_{PPI}}{mCit_{PA-mCit-NL}/NL_{PA-mCit-NL}} \quad (4)$$

### Processing publicly available data and selecting multiprotein complexes

Reference PDB structures and homologous structures for interactions in the hsPRS-v2 were obtained from interactome insider http://interactomeinsider.yulab.org/ (Meyer et al, 2018). Publicly available binary protein interaction datasets used in this study came from the original Choi et al (Choi et al, 2019) and Yao et al (Yao et al, 2020) publications.

Human protein complexes used in this study were selected based on the following criteria. First, human protein complexes should have at least one experimentally determined structure in PDB (Berman et al, 2000). Second, the complex should have at least four subunits. Third, at least 80% of entry clones for individual subunits of a complex should be present in the human ORFeome 8.1 collection (Yang et al, 2011). A total of 24 distinct complexes (Dataset EV5) with different PDB structures met those criteria, and three protein complexes with well-documented biological functions were selected from this list: LAMTOR, BRISC and MIS12. Published SARS-CoV-2 interactions were extracted from the IMEx database using the search term "coronavirus" (https://www.ebi.ac.uk/intact/imex) as of 2023-01-20 (Orchard et al, 2014).

### Multi-adaptive support vector machine learning algorithm

**Construct normalization:** The multi-adaptive supporting vector machine learning algorithm was adapted from Yang et al (Yang et al, 2019) and Kim et al (Kim et al, 2021). Standardization of datasets is a common requirement for many machine learning algorithms. We observed a strong construct-specific variance in the multiprotein complex reference set (Appendix Fig. S5) and the binary SARS-CoV-2 mapping (Appendix Fig. S6). We argued that constructs with a high variance are unlikely to form significantly more or less interactions than constructs with a low variance, but rather assumed that the observed variance is probably a technical artifact that could, for example, be explained by "sticky" proteins. To be able to apply a machine learning algorithm that universally applies to all constructs, we used a percentile-based scaling

approach (RobustScaler, https://scikit-learn.org/), that is not influenced by a small number of very large marginal outliers, i.e., not influenced by high scoring, e.g., true-positive interactions.

1. Calculate the $median_{construct}$ and $IQR_{construct}$ for constructs with at least 20 tested interactions. For constructs tested against less than 20 other constructs, we advise to not perform a construct normalization.
2. Calculate the global median and interquartile range (IQR) between the 25th to 75th quartile for the training and test sets combined: $median_{global}$ and $IQR_{global}$.
3. Calculate a correction factor for all constructs $Cf_{construct}$ as the $median_{construct} - median_{global}$.
4. Subtract from each interaction scores the construct-specific correction factor $Cf_{construct}$ and if the $IQR_{construct}$ is larger than the $IQR_{global}$ divide by the $IQR_{construct}/IQR_{global}$. This retains the original scale of the scores.
5. Recalculate the $IQR_{construct}$ for all constructs and if the $IQR_{construct}$ is larger than the $IQR_{global}$ for some constructs, repeat steps 3 and 4.

**Training feature normalization:**

1. Perform robust scaler or standard scaler normalization or other appropriate normalization steps for your training features. See Dataset EV2 for information and normalization procedures applied for the different assays.

**Model training:** Machine learning with adaptive sampling (AdaSampling) is a framework developed for both positive-unlabeled (PU) learning and learning with class label noise (LN) (Yang et al, 2019). For binary interaction assays, class LN can refer to protein pairs in the PRS that score negative or protein pairs in the RRS that score positive. Through adaptive sampling, the class mislabeling probability can be estimated, which progressively reduces the risk to select mislabeled instances for model training (Yang et al, 2019).

1. Assemble reference sets as described in the results section.
2. Ensure to perform construct normalization if appropriate as described above.
3. Select normalized training features. For the LuTHy-BRET assay, we selected the cBRET ratio and $mCit_{cell}$ and performed a robust scaler normalization. Training features and normalization parameters for all described assays can be found in Dataset EV2.
4. Assemble the positive and negative training sets (ensemble 'e') from the reference sets by unweighted sampling 'j' protein pairs (minimum 30) using a "seed" for reproducibility. Calculate the number of 'j' so that all interactions in the reference set will have been used for training at least once, which depends on the number of ensembles chosen (step 8) using the following formula: log(1–0.9999)/log(1 − j/total protein pairs).
5. Train a support vector machine learning algorithm using, for example, the "svm" function of the "e1071" package for R and perform hyperparameter optimization to find the optimal regularization parameter 'C' (Dataset EV1).
6. Use the resulting SVM model to reclassify the training set. Perform a hyperparameter optimization to find the optimal number of iterations (Dataset EV1).

7. Use the reclassified model to predict the classification probability of protein pairs in the test set.
8. Repeat the training and predictions (4–7) in a paired fashion. Perform a hyperparameter optimization to find the optimal number of ensembles how often to sample, train, reclassify, and predict.
9. Calculate the average classifier probabilities from all models.

**Evaluation of training performance:**

1. For each training set ensemble (e), sample, for example, 10%, 20%, 30%, 40%, 50%, 60%, 70%, 80% 90%, and 100% of the training set with size 'j' and train for each an maSVM algorithm as described above.
2. Use each model to predict the classifier probability on the entire test set (all hsPRS-v2 and hsRRS-v2 protein pairs without protein pairs used in the training set).
3. Calculate for each paired training and test set, the accuracy, hinge loss and binary cross-entropy loss for both using the following formulas:
   a. accuracy: actual labels / predicted labels
   b. hinge loss: 1 - actual labels * predicted labels with hinge loss set to 0 if calculation results in a negative value
   c. binary cross-entropy loss: -median(actual labels * log(predicted probs) + (1 - actual labels) * log(1 - predicted probs))
4. Plot the fraction of training data against the calculated values for the training and the test set and evaluate training performance and over- and underfitting.

The maSVM algorithm for scoring interactions and calculating recovery rates is available on GitHub: https://github.com/philipptrepte/binary-PPI-classifier. The LuTHy reference set data obtained in this work can also be accessed from the /data folder of the GitHub repository.

## *AlphaFold-Multimer protein complex prediction and classification*
**AlphaFold-Multimer:**

1. Predict protein complex structures by AlphaFold-Multimer (Evans et al, 2022) using, for example, ColabFold (Mirdita et al, 2022) with the following parameters: use_amber: 'no'; template_mode: 'none' (no pdb template information is used); msa_mode: 'MMseq2 (UniRef+Environmental)', pair_mode: 'unpaired+paired' (pair sequences from same species); model_type: 'auto' (AlphaFold2-multimer-v2); num_models: 5; num_recycles: 3; rank_by: 'auto'; stop_at_score: 100.
2. Predict for each protein complex 5 models.

**Extract pLLDT and PAE values from AFM predictions:**

1. Extract the resulting pLLDT and PAE scores extracted from .json files using, for example, the "fromJSON" function of the "rjson" library for R.
2. Filter the inter-subunit PAE values for each model on pLLDT scores >50.
3. If there are at least 10 amino acid inter-subunit PAE values remaining, perform *k*-means clustering row-wise and column-wise, which results in a total of 4 cluster, using, for example, the "*k*-means" function of the "stats" package for R. (Note that for 4

models of hsPRS-v2 PPIs and 6 models of hsRRS-v2 protein pairs the PAE clustering failed).
4. Calculate the average PAE for all complexes with successful *k*-means clustering in order to identify the interaction interface region with the closest average distance. In case *k*-means clustering fails, calculate the average PAE for all inter-chain amino acids with pLLDT scores >50.

**PDBePISA:**

1. Use PDBePISA (https://www.ebi.ac.uk/pdbe/pisa/) to determine the interaction interface and solvation-free energy of the AlphaFold-Multimer predicted structures (Krissinel and Henrick, 2007). The Python script PisaPy (https://github.com/hocinebib/PisaPy) can be used for batch analysis.
2. Process the "interfacetable.xml" files using, for example, the "read_xml" function of the "xml2 package" in R (Note that PDBePISA prediction failed for all five models of the hsRRS-v2 protein pair PSMD12 + CRIPT. For 13 hsPRS-v2 interactions and 56 hsRRS-v2 proteins at least one, but not all five models were predicted (see Fig. 2). Overall, 4.75±0.8 models were successfully predicted for all hsPRS-v2 interactions and 4.23±1.4 models for hsRRS-v2 protein pairs.

**Classification:**

1. Use the maSVM algorithm as described above without construct normalization. Standard scale the PAE and interface area values and use them as training features.
2. Perform a hyperparameter optimization for the ensemble size, number of reclassifications and the regularization parameter or use parameters determined for this study: e = 100 ensembles; i = 5 reclassifications; regularization parameter 'C' = 0.01 (Datasets EV1 and EV2).

For the structure predictions in this study, we have used AlphaFold2-multimer-v2 with a PDB training cutoff of 2018-04-30 (https://github.com/google-deepmind/alphafold/blob/main/docs/technical_note_v2.3.0.md) on ColabFold (version 1.2.0 or 1.3.0). The analysis pipeline for extracting PAE and pLLDT values from AFM predictions, *k*-means clustering and maSVM prediction is available on GitHub: https://github.com/philipptrepte/AFM-Pisa-classifier/. The PAE, pLLDT, interfaceArea and ΔG values for the AFM reference set data (hsPRS-AF, hsRRS-AF) can be accessed from the /data folder of the GitHub repository.

## *Ultra-large virtual screening with VirtualFlow*
Ultra-large virtual screening was performed using the VirtualFlow workflow engine (Gorgulla et al, 2020) on a Sun Grid Engine-(MaxCluster, Max Delbrück Center) or a SLURM-managed (JURECA supercomputer, Forschungszentrum Jülich) high-performance computing cluster (Krause and Thörnig, 2018). A subset (~350 M ligands) of the "ready-to-dock" Enamine REAL library was docked onto the experimentally validated NSP10 interaction interface. For primary ultra-large docking, Quick Vina 2 (Alhossary et al, 2015) was used with exhaustiveness set to 1. Ligands were ranked based on their predicted binding free energies in kcal/mol. Then, the top 10 M scoring ligands were re-docked using Smina Vinardo (Apr 2, 2016, based on Autodock Vina

version 1.1.2) and Autodock Vina (version 1.1.2) (Trott and Olson, 2010) with flexible residues at the target region (Val, Met, Phe, Ser, Cys, Cys, Arg, His, Tyr, Lys, Lys, His). AutoDockTools was used to generate the rigid and flexible structures in the PDBQT format. Exhaustiveness in the re-docking was also set to 1 and two iterations for each docking scenario were conducted. The size of the cuboid docking box for all scenarios and docking runs was set to $75.647 \times 16.822 \times 17.631$ Å. Ligands were ranked by the mean scores of the replica of the Smina Vinardo and Autodock Vina dockings, respectively. Finally, the ligands, which were present in the top 10 K of both docking scenarios were selected (~2 K ligands). The ligands were then chemically clustered to identify cluster representatives. Clustering was performed using the Cluster Molecules component embedded in the Pipeline Pilot Software (BIOVIA Pipeline Pilot, Release 2018, San Diego, Dassault Systèmes) using FCFP_4 Fingerprints and an average of 60 compounds per cluster in order to obtain 30 clusters. Molecules were filtered based on reactivity, toxicity and drug-likeness using Pipeline Pilot Software according to Horvath et al (Horvath et al, 2014) and solubility was predicted according to Cheng and Merz (Cheng and Merz, 2003). From each cluster, a representative molecule was selected based on reactivity, toxicity and drug-likeness properties as well as their predicted solubility. In total, 20 molecules were selected and ordered for synthesis at Enamine Ltd. (Kiev, Ukraine). Of the 20 compounds, 15 (#456 to #470) were successfully synthesized and delivered. Compounds were diluted to 10 or 50 mM stock solutions in DMSO, and stored at $-20\,°C$.

### Recombinant protein production

The NSP10 expression construct comprising amino acids 23–145 was cloned into a modified pET28a vector, resulting in the expression of an N-terminal $His_6$-tagged protein (MGSDKIHHHHHHNSTVLS... GCSCDQ). The protein was produced at $17\,°C$ using *E. coli* BL21-AI cells (ThermoFisher Scientific), induced with 0.5 mM isopropyl β-D-1-thiogalactopyranoside (IPTG) and 0.2% (v/v) L-arabinose. For purification, cells were resuspended in lysis buffer (50 mM sodium phosphate pH 7.8, 0.5 M NaCl, 5% glycerol) supplemented with 0.25% (w/v) 3-[(3-cholamidopropyl)-dimethylammonio]-1-propane- sulfo-nate (CHAPS), 1 mM phenylmethyl-sulfonyl fluoride (PMSF), 3000 U/mL lysozyme (Serva) and 7.5 U/mL RNase-free DNase I (AppliChem), lysed by multiple freeze-thaw cycles and the extract was cleared by 1 h centrifugation at $34.000 \times g$. The $His_6$-fusion protein was captured from the supernatant using metal affinity chromatography on a HisTrap™ FF Crude Column (Cytiva) equilibrated with 20 mM Tris-HCl pH 8.0 and 0.5 M NaCl. After several wash steps, the protein was eluted with 20 mM Tris-HCl pH 8.0, 0.5 M NaCl, and 250 mM imidazole, and further purified by a size-exclusion chromatography step on a 26/600 Superdex 75 prep grade column (Cytiva) equilibrated with 20 mM HEPES pH 7.5 and 0.2 M NaCl. The purified protein was concentrated to 8 mg/mL, flash-frozen with liquid nitrogen, and stored at $-80\,°C$ until further use.

The NSP10-NSP16 co-expression plasmid (cloned into pET-Duet-1, Novagen) was transformed into *E. coli* BL21-AI cells (ThermoFisher Scientific). The protein was produced at $17\,°C$ upon induction with 0.5 mM IPTG and 0.2% (v/v) L-arabinose. For purification, cells were resuspended in lysis buffer (50 mM Tris-HCl pH 7.6, 0.5 M NaCl, 5% glycerol) supplemented with 0.25% (w/v) CHAPS, 1 mM PMSF, 3000 U/mL lysozyme (Serva), 7.5 U/mL RNase-free DNase I (AppliChem), 1 mM $MgCl_2$, lysed by

multiple freeze-thaw cycles, and the extract was cleared by centrifugation. The $His_6$-tagged NSP16, complexed with co-expressed NSP10, was captured from the supernatant using metal affinity chromatography on a HisTrap™ FF Crude Column (Cytiva) equilibrated with 50 mM Tris-HCl pH 7.6, 0.5 M NaCl, and 5% glycerol. After several wash steps, the protein was eluted with the same buffer, including 250 mM imidazole, and further purified by a size-exclusion chromatography step on a 16/600 Superdex 75 prep grade column (Cytiva) equilibrated with PBS pH 7.4 and 0.3 M NaCl. The purified protein was supplemented with 5% (v/v) glycerol and 1 mM DTT.

The purified proteins were flash-frozen with liquid nitrogen and stored at $-80\,°C$ until further use. The molecular mass of all purified proteins was confirmed by intact mass analyses using an Agilent 1290 Infinity II UHPLC system coupled to an Agilent 6230B time-of-flight (TOF) instrument.

### Methyltransferase assay

Methyltransferase activity of the NSP10-NSP16 complex and inhibitor screening was performed using the MTase-Glo assay (Cat. No. V7601, Promega) (Hsiao et al, 2016) according to the manufacturer's instructions. In brief, 18.75 μM RNA cap-0 oligo (5'-(N7-MeGppp)ACAUUUGCUUCUGAC-3') or no RNA as control was incubated in the presence of 740 nM co-purified NSP10-NSP16 protein complexes or no protein as control in reaction buffer (20 mM Tris-HCl pH 8.0, 50 mM NaCl, 1 mM EDTA, 3 mM $MgCl_2$, 0.1 mg/ml BSA, 1 mM DTT) supplemented with 0.2 U/μL SUPERase RNAse inhibitor and 40 μM S-adenosylmethionine (SAM). The NSP10-NSP16 complex was incubated with 100 μM inhibitor compound or DMSO as a control for 10 min before adding the remaining components. Next, enzymatic reactions were kept for 2 h at $37\,°C$ in 8 μl total volume in 384-well plates (CLS3824, Corning). After incubation, 2 μl of 5× MTase-Glo Reagent (Promega) was added to wells, plates were shaken for 1 min at 1000 rpm and further incubated for 30 min at RT. Then, 10 μl of MTase-Glo Detection Solution (Promega) was added, plates were again shaken for 1 min at 1000 rpm and further incubated for 30 min at RT. Finally, firefly luminescence intensity (~565 nm) was quantified in a SpectraMax iD5 microplate reader (Molecular Devices).

### Microscale thermophoresis

For microscale thermophoresis (MST) assays, NSP10 protein was fluorescently labeled using *N*-Hydroxysuccinimide (*NHS*)-ester fluorophores according to the manufacturer's instructions (Protein Labeling Kit RED-NHS 2nd Generation, Nanotemper). Labeling was performed in the NSP16 protein storage buffer (20 mM HEPES pH 7.5 and 0.3 M NaCl). Prior to use in MST experiments, the labeled protein solution (2 μM) was centrifuged at $15.000 \times g$ for 10 min at $4\,°C$ to remove potential aggregated protein species. Compound/NSP10 experiments were performed in 20 mM HEPES, 0.3 M NaCl with 0.05% Tween. For the binding studies, 100 or 200 nM NSP10-RED (depending on labeling efficiency) were incubated with increasing amounts of compound for 15 min at RT. Measurements were performed in standard capillaries (Nano-temper) using a Monolith MST device (Nanotemper) with 80% LED and medium infrared (IR) laser power. Binding was analyzed from MST signals after 5 or 20 s compared to relative fluorescence before IR laser pulse.

### LuTHy-BRET compound assay

The LuTHy assay procedure was performed as described above using the in-cell BRET readout. After transfection and expression of the LuTHy constructs for 48 h, cells were treated with indicated concentrations of compound dissolved in DMSO infused into the cell culture media. Control wells were treated with DMSO only. After 3 h of compound incubation, the fluorescence of mCitrine was recorded at Ex 500 nm/Em 530 nm using the Infinite® microplate reader M1000 PRO (Tecan), and cell morphology as well as confluence (Appendix Fig. S8) were analyzed by automated imaging using a Spark multimode microplate reader (Tecan). Then, coelenterazine-h (pjk, #102182) was added to a final concentration of 5 μM (5 mM stock dissolved in methanol), cells were incubated for an additional 15 min and total luminescence intensity as well as luminescence intensities at short (370–480 nm) and long (520–570 nm) wavelengths were measured using the Infinite® microplate readers M1000 PRO (Tecan). BRET ratios were calculated as described above and normalized to solvent control wells (normalized BRET (nBRET) ratio).

### SARS-CoV-2 replication assay

HEK293-ACE2 ($3 \times 10^4$ cells per well) were plated in white 96-well plates. The cells were then infected with SARS-CoV-2 (Hou et al, 2020; Kim et al, 2022) (0.01 MOI) containing a nanoluciferase reporter and were simultaneously treated with the compounds 459 in 13-point twofold serial dilution 0–100 μM concentration. Cells were further cultured for another 24 h and luminescence was measured (Coutant et al, 2020). Cell viability was measured using the CellTiter-Glo Luminescent Cell Viability Assay kit (Promega,G7750). All experiments were performed in a BSL3 laboratory authorized by the Service Universitaire de Protection et d'Hygiène du Travail, Université de Liege, Belgium. The contained use of OGM/MGM or pathogens, including SARS-CoV2 in this BSL3 has been notified in the environmental permit PE/1/1 valid until January 22, 2034.

### Combination study of compound 459 with AZ1

HEK293-ACE2 were seeded in 96-well plates as described above. Cells were then infected with SARS-CoV-2-nanoluciferase (0.01 MOI) and treated with the indicated concentration of compound 459 with 2.5 μM AZ1 or DMSO (three replicates for each concentration). The following day, luminescence was measured using the Filtermax F5 microplate reader (Molecular Devices).

## Data availability

Newly generated DNA or RNA constructs, proteins and compounds are available upon request. The protein interactions from this publication have been submitted to the IMEx (http://www.imexconsortium.org) consortium through IntAct (Orchard et al, 2014) and have the assigned identifier IM-29926. All UniProt and RCSB-PDB accession codes, as well as the AlphaFold-Multimer predictions are provided in the Source Data. The datasets and computer code produced in this study are available in the following databases: (1) PPI datasets: LuTHy and mN2H binary interaction data (http://www.imexconsortium.org/) Identifier: IM-29926; (2) Virtual screening workflow code: GitHub (https://github.com/VirtualFlow); (3) Binary PPI classifier: GitHub (https://github.com/philipptrepte/

binary-PPI-classifier); (4) AFM PISA classifier: GitHub (https://github.com/philipptrepte/AFM-Pisa-classifier).

## Peer review information

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

## Acknowledgements

The authors would like to thank all members of the Wanker, Vidal, Twizere, and Landthaler laboratories for helpful discussions throughout this project. We would also like to thank Luke Lambourne for critically reading the manuscript and providing feedback on the machine learning and AlphaFold predictions. We also thank the team of the Protein Production & Characterization Technology Platform of the Max Delbrück Center (MDC), Berlin, Germany, and the GIGA Viral Vector Platform of the University of Liège, Liège, Belgium for excellent technical assistance. Additional gratitude to Sebastian Lührs (JURECA supercomputer, Forschungszentrum Jülich) and Martin Siegert (MaxCluster, MDC) for technical support on cluster usage. The authors would also like to thank ChemAxon for a free academic research license. This work was supported by the Helmholtz Association, iMed and Helmholtz-Israel Initiative on Personalized Medicine (Germany); the Federal Ministry of Education and Research and e:med Systems Medicine–IntegraMent 01GS0844 (Germany) all to EEW; as well as by the CHDI Foundation (USA), the German Cancer Consortium DKTK (Germany) and the Deutsche Krebshilfe, ENABLE (Germany) to EEW and PT. CS was in part funded by the Deutsche Forschungsgemeinschaft (DFG, German Research Foundation) under Germany's Excellence Strategy – The Berlin Mathematics Research Center MATH+ (EXC-2046/1, project ID: 390685689). IM and ML were in part funded by DFG LA 2941/17-1. This work was also supported by a Claudia Adams Barr Award to SGC, Fonds de la Recherche Scientifique (FRS-FNRS) Grants FC27371 to JO, FC38907 to JB, FC34947 to SBM and PER-40003579 to JCT and a Wallonia-Brussels International (WBI)-World Excellence Fellowship to JO. Additional support was provided by NIH grants P50HG004233, R01GM130885, R01CA266194 awarded to MV and U41HG001715 awarded to MV, DEH, and MAC. This work was also supported by the LabEx IBEID (grant 10-LABX-0062). JO thanks the Rega Institute for Medical Research for financial support. MV is a Chercheur Qualifié Honoraire, and JCT a Maître de Recherche of the Fonds de la Recherche Scientifique (FRS-FNRS, Wallonia-Brussels Federation, Belgium).

## Author contributions

**Philipp Trepte**: Conceptualization; Data curation; Software; Formal analysis; Supervision; Funding acquisition; Validation; Investigation; Visualization; Methodology; Writing—original draft; Project administration; Writing—review and editing. **Christopher Secker**: Conceptualization; Data curation; Software; Formal analysis; Supervision; Investigation; Methodology; Writing—original draft; Project administration; Writing—review and editing. **Julien Olivet**: Conceptualization; Data curation; Supervision; Funding acquisition; Investigation; Methodology; Writing—original draft; Project administration; Writing—review and editing. **Jeremy Blavier**: Investigation; Methodology; Writing—review and editing. **Simona Kostova**: Investigation; Methodology. **Sibusiso B Maseko**: Investigation; Methodology. **Igor Minia**: Investigation; Methodology. **Eduardo Silva Ramos**: Data curation. **Patricia Cassonnet**: Investigation. **Sabrina Golusik**: Investigation. **Martina Zenkner**: Investigation. **Stephanie Beetz**: Investigation. **Mara J Liebich**: Investigation. **Nadine Scharek**: Investigation. **Anja Schütz**: Resources; Investigation; Writing—review and editing. **Marcel Sperling**: Software. **Michael Lisurek**: Data curation. **Yang Wang**: Data curation. **Kerstin Spirohn**: Investigation. **Tong Hao**: Data curation; Investigation; Methodology. **Michael A Calderwood**: Supervision; Funding acquisition; Writing—review and editing. **David E Hill**: Supervision; Funding acquisition. **Markus Landthaler**: Resources; Supervision; Funding acquisition; Project administration. **Soon Gang Choi**: Conceptualization; Data curation; Funding acquisition; Investigation; Methodology; Writing—review and editing. **Jean-Claude Twizere**: Resources; Supervision; Funding acquisition; Project administration. **Marc Vidal**: Conceptualization; Resources; Supervision; Funding acquisition; Project administration; Writing—review and editing. **Erich E Wanker**: Conceptualization; Resources; Supervision; Funding acquisition; Project administration; Writing—review and editing.

## Funding

## Disclosure and competing interests statement

The authors declare no competing interests. MV is an editorial advisory board member. This has no bearing on the editorial consideration of this article for publication.

# Expanded View Figures

**Figure EV1.  (related to Fig. 1). Effect of different scoring approaches on recovery rates.**

(**A**) Schematic overview of the LuTHy-BRET and LuTHy-LuC assays. X: Protein X, Y: Protein Y, D: NanoLuc donor, A: mCitrine acceptor, AB: antibody. (**B**) With the LuTHy assay, each protein pair X–Y can be tested in eight possible configurations (N- vs. C-terminal fusion for each protein), and proteins can be swapped from one tag to the other resulting in 16 quantitative scores for each protein pair, i.e., eight for LuTHy-BRET and eight for LuTHy-LuC. (**C**) Line plots showing the fraction of protein pairs that scored positive (*y* axis) dependent on the quantitative interaction scores (*x* axis) for 10 binary PPI assay versions. For each tested protein pair, the tagging configuration with the highest interaction score is used. For LuTHy all eight tagging configurations were tested, whereas for MN2H, VN2H, YN2H, GPCA, NanoBi four and for KISS, MAPPIT and SIMPL two tagging configurations were tested. Recovery rates at maximum specificity, i.e., where none of the protein pairs in the RRS scored positive (0%), are indicated. Note that in Choi et al (Choi et al, 2019) recovery rates at maximum specificity were calculated by using distinct cutoffs for each tagging configuration. (**D**) Line plots showing the fraction of protein pairs that scored positive (*y* axis) dependent on the distribution of interaction scores, i.e., the mean of all interaction scores + n*(sd) (*x* axis) for 10 binary PPI assay versions. Recovery rates at mean + 1 standard deviation are indicated. (**E**) Line plots showing the fraction of protein pairs that scored positive (*y* axis) dependent on the distribution of interaction scores, i.e., the median of all interaction scores + n*(sd) (*x* axis) for 10 binary PPI assays. Recovery rates at median + 1 standard deviation are indicated. LuTHy experiments from this study were repeated twice with $n = 2$, biological replicates, each containing $n = 3$ technical replicates; SIMPL from Yao et al (Yao et al, 2020); all other from Choi et al (Choi et al, 2019). Note that the SIMPL assay was benchmarked by Yao et al (Yao et al, 2020) against 88 positive proteins pairs derived from the hsPRS-v1 (Venkatesan et al, 2009) and as a random reference set against "88 protein pairs with baits and preys selected from the PRS but used in combinations determined computationally to have low probability of interaction" (Yao et al, 2020).

▶

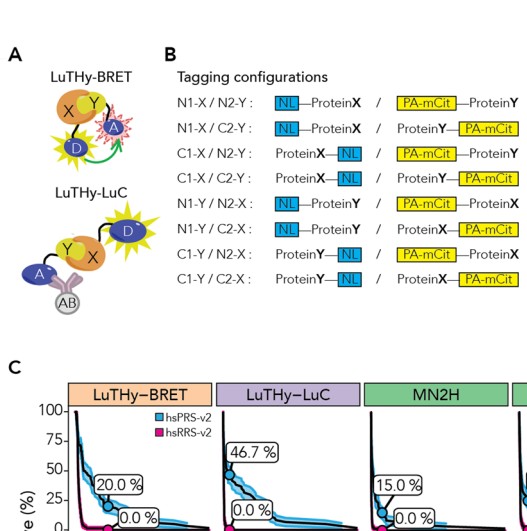

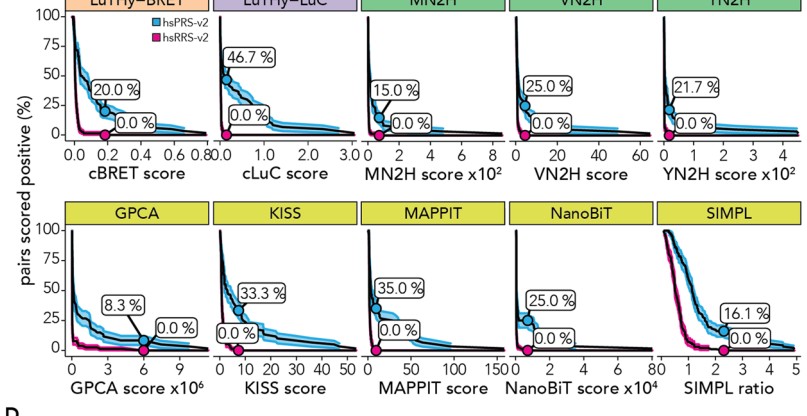

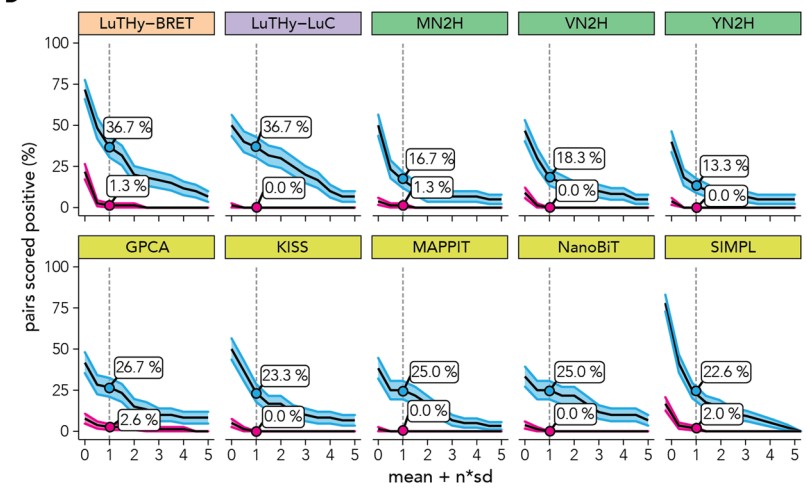

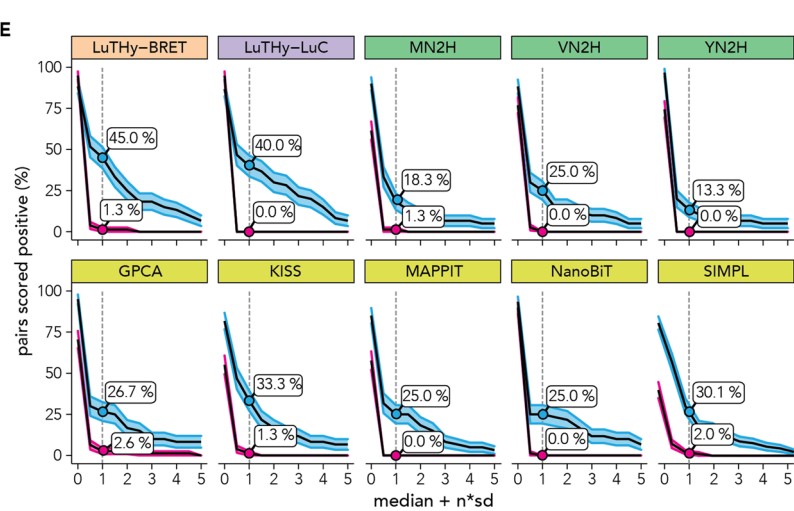

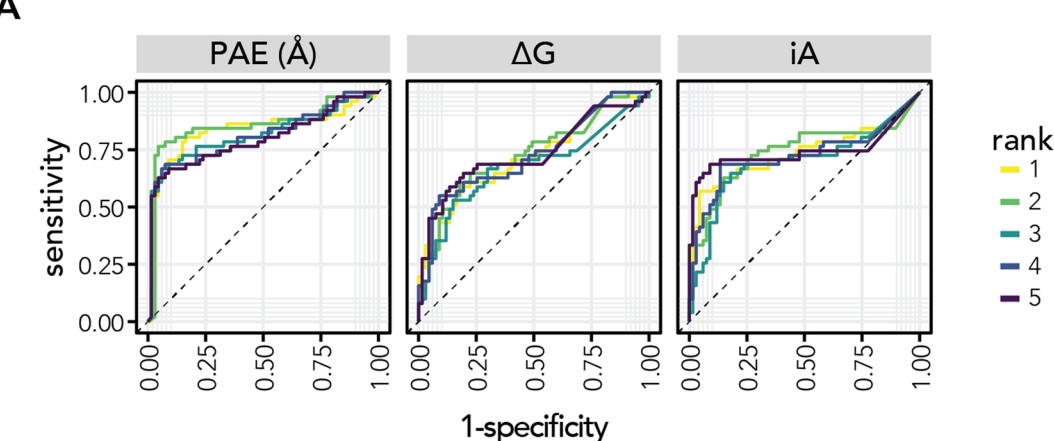

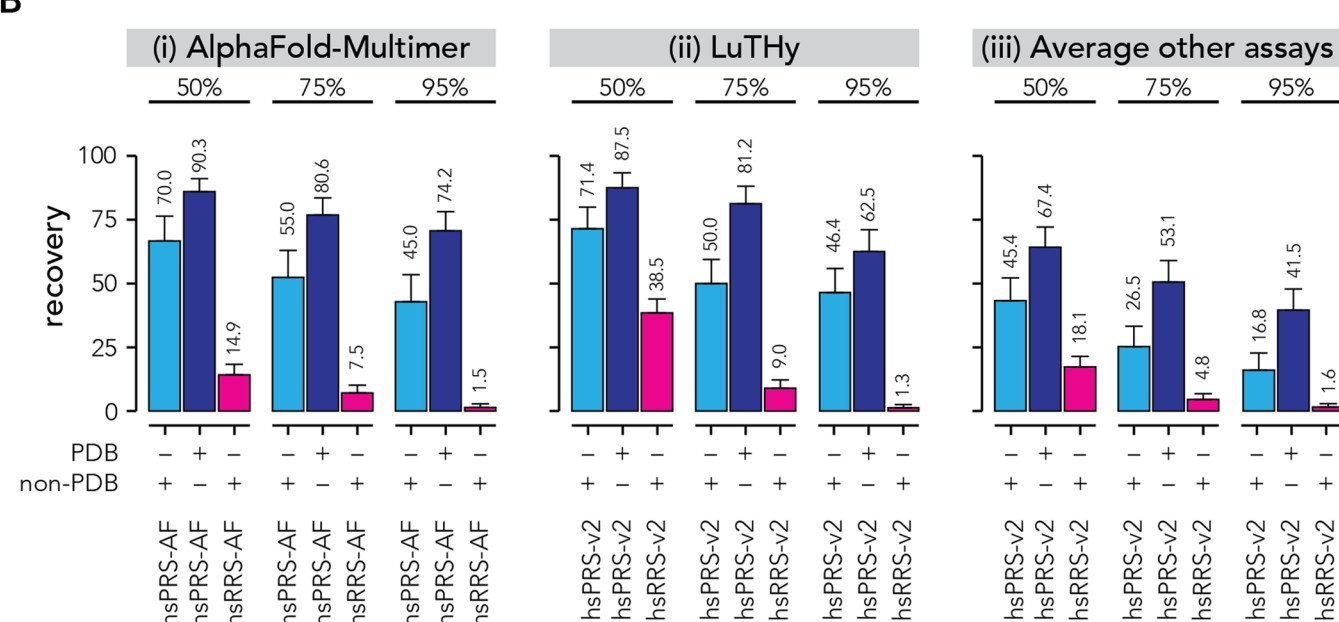

**Figure EV2.** (related to Fig. 2). **Training a maSVM algorithm to classify AFM-predicted structures.**

(A) Receiver characteristic analysis comparing sensitivity and specificity between the five AFM-predicted structural models for PAE, ΔG and iA of the hsPRS-AF and hsRRS-AF. (B) Bar plots showing the fraction of hsPRS-AF and hsRRS-AF interactions with structures deposited in PDB that scored above classifier probabilities of 50%, 75% and 95% by AlphaFold-Multimer (i) by LuTHy (ii) or the mean recovery of N2H (MN2H, VN2H, YN2H), GPCA, KISS, MAPPIT and NanoBiT (iii). Data for the SIMPL assay was excluded for this analysis due to the different composition of the reference sets. LuTHy experiments from this study were repeated two times with $n = 2$, biological replicates, each containing $n = 3$ technical replicates; AFM was used to predict $n = 5$ structural models; all other from Choi et al (Choi et al, 2019). Bars and error bars in this figure represent mean values and standard error of the proportion, respectively.

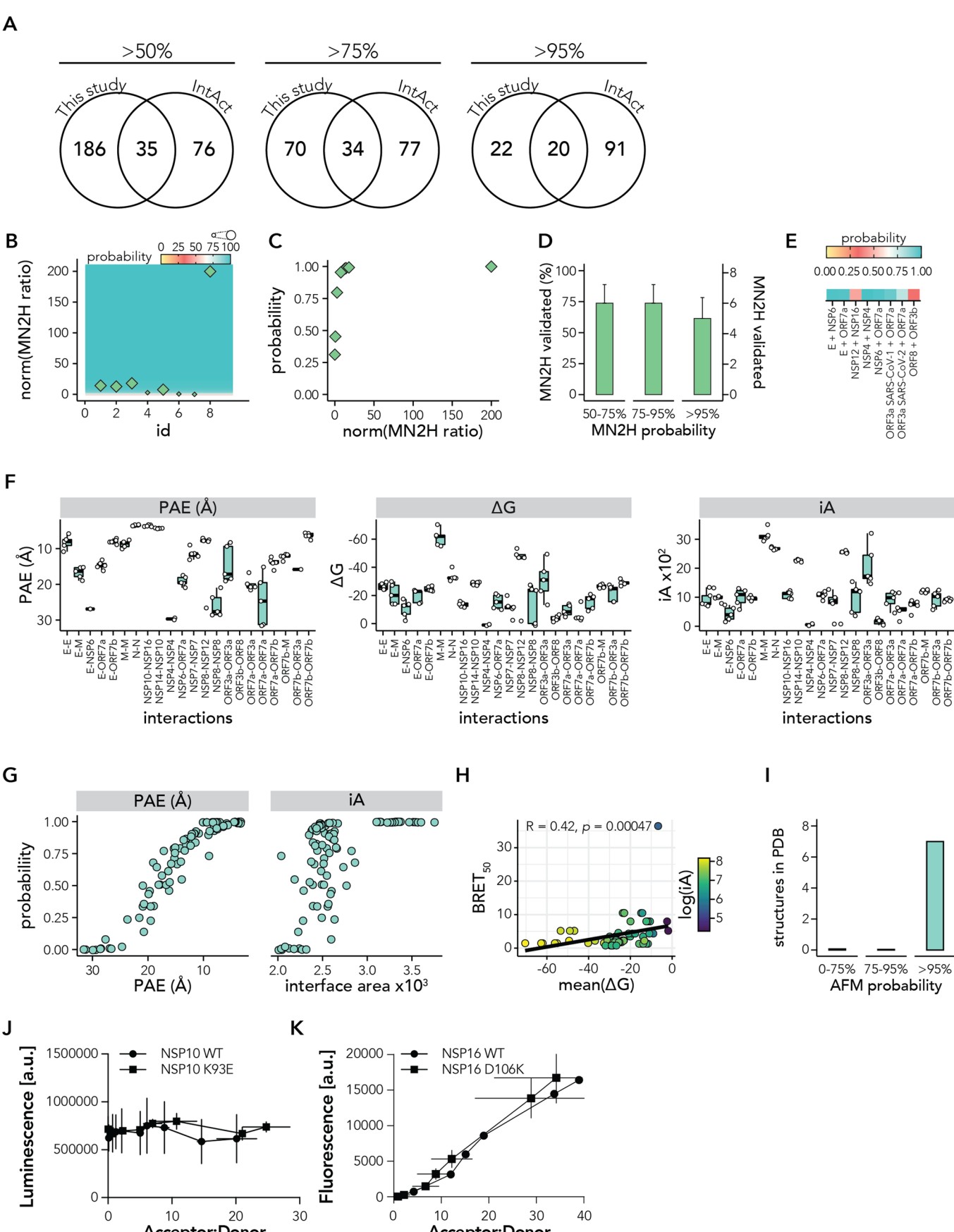

◀  **Figure EV3.   (related to Figs. 4 and 5). Validating SARS-CoV-2 protein interactions using the mN2H assay and predicting SARS-CoV-2 protein complexes structures using AlphaFold-Multimer.**

(A) Venn diagrams showing the overlap between interactions recovered by LuTHy at >50%, >75% and >95% probabilities and interactions deposited in the IntAct database (Orchard et al, 2014). (B) Scatter plot showing normalized mN2H ratios (y axis) of each of the eight SARS-CoV-2 interactions newly identified with LuTHy (x axis). Average classifier probabilities obtained from the hsPRS-v2/hsRRS-v2 mN2H models are displayed as the size of the data points and as a colored grid in the background. (C) Scatter plot showing normalized mN2H ratios (x axis) against classifier probabilities (y axis) for the newly identified SARS-CoV-2 interactions selected for validation. (D) Bar plots showing the fraction (left y axis) and number (right y axis) of newly identified SARS-CoV-2 interactions selected for validation that scored above classifier probabilities of 50%, 75% or 95% with mN2H. Bars and error bars represent mean values and standard error of the proportion, respectively, with n = 3 biological replicates. (E) Heatmaps showing the mN2H classifier probabilities for the newly identified SARS-CoV-2 interactions selected for validation. (F) Boxplots showing predicted alignment error (PAE), solvation-free energy (ΔG) and interface area (iA) from AlphaFold-Multimer (AFM) predicted SARS-CoV-2-AF structures. Boxplots display the median, lower and upper hinges of the 25th and 75th percentiles and lower and upper whiskers extending from the hinges with 1.5× the interquartile range. Each dot represents one predicted structural model. (G) Scatter plot showing PAE (x axis) against interface area (y axis) for all SARS-CoV-2-AF (orange) protein pairs. Average classifier probability predicted by the 100 maSVM models trained by the hsPRS-AF and hsRRS-AF set (see Fig. 2F), is displayed as the size of the data points. Each point in the colored grid in the background displays the average classifier probabilities from the 100 maSVM models. (H) Scatter plot showing the ΔG (x axis) for all five AFM-predicted structural SARS-CoV-2-AF models against the LuTHy-BRET determined binding strengths (BRET$_{50}$, see Appendix Fig. S7A,B). The respective log-transformed interface areas are indicated by the fill color of the data points. A linear regression fit through the data is shown and the Spearman correlation coefficient (R) and P value are indicated. (I) Barplot showing the fraction of AFM-predicted structures with 0–75%, 75–95% and >95% classification probability that have an experimentally reported structure deposited to the PDB (Berman et al, 2000) database. (J,K) Luminescence (J) and fluorescence (K) values from LuTHy-BRET donor saturation experiments, where constant amounts of NSP10-NL WT or K93E (Lys93Glu) are co-expressed with increasing amounts of mCitrine-NSP16 WT or D106K (Asp106Lys). Experiments with NSP10-NL WT and K93E were repeated two times, with n = 2, biological replicates, and each with n = 2 technical replicates; experiments with mCit-NSP16 WT and D106K were repeated four times, with n = 4, biological replicates, and each with two technical replicates, n = 2. Bars and error bars represent the mean and standard deviation, respectively.

