## [Peer Review File · Molecular Systems Biology]

AI-guided pipeline for protein-protein interaction drug discovery identifies a SARS-CoV-2 inhibitor

Philipp Trepte, Christopher Secker, Julien Olivet, Jeremy Blavier, Simona Kostova, Sibusiso Maseko, Igor Minia, Eduardo Silva Ramos, Patricia Cassonnet, Sabrina Golusik, Martina Zenkner, Stephanie Beetz, Mara Liebich, Nadine Scharek, Anja Schuetz, Marcel Sperling, Michael Lisurek, Yang Wang, Kerstin Spirohn, Tong Hao, Michael Calderwood, David Hill, Markus Landthaler, Soon Gang Choi, Jean-claude Twizere, Marc Vidal, and Erich Wanker

Corresponding author(s): Erich Wanker (erich.w@mdc-berlin.de), Christopher Secker (christopher.secker@mdc-berlin.de), Jean-claude Twizere (jean-claude.twizere@uliege.be), Philipp Trepte (philipp.trepte@imba.oeaw.ac.at), Soon Gang Choi (soongangchoi@gmail.com)

Review Timeline:

Submission Date:	9th Feb 23
Editorial Decision:	27th Mar 23
Revision Received:	5th Nov 23
Editorial Decision:	9th Jan 24
Revision Received:	22nd Jan 24
Accepted:	23rd Jan 24

Editor: Maria Polychronidou

Transaction Report:

27th Mar 2023

Manuscript Number: MSB-2023-11595

Title: AI-guided pipeline for protein-protein interaction drug discovery identifies an SARS-CoV-2 inhibitor

Dear Dr. Wanker,

Thank you again for submitting your work to Molecular Systems Biology. We have now heard back from the three reviewers who agreed to evaluate your study. As you will see below, the reviewers acknowledge that the presented approach seems relevant. They do however raise a series of concerns, which we would ask you to address in a revision.

Without repeating all the points listed below, some of the more fundamental issues are the following:

- Further comparisons of the presented method to existing approaches should be included.
- The methodology and the machine learning algorithm should be described in better detail. Importantly, the code should be made available.
- The limitations and advantages of the presented method should be discussed more extensively.
- Additional validations should be performed to better support the main conclusions.

All issues raised by the referees would need to be satisfactorily addressed. Please let me know in case you would like to discuss in further detail any of the issues raised, I would be happy to schedule a call.

On a more editorial level, we would ask you to address the following points:

- Please provide a .doc version of the manuscript text (including legends for the main figures) and individual production quality figure files for the main Figures (one file per figure).
- We have replaced Supplementary Information by the Expanded View (EV format). In this case, all additional figures and Tables can be included in a PDF called Appendix. Appendix figures and Tables should be labeled and called out as: "Appendix Figure S1, Appendix Figure S2... Appendix Table S1..." etc. Each legend should be below the corresponding Figure/Table in the Appendix. Please include a Table of Contents in the beginning of the Appendix. For detailed instructions regarding expanded view please refer to our Author Guidelines: .
- Supplementary Tables 1-5 should be provided as Datasets EV1-EV5. Please provide one file per dataset. In each file, a description of the dataset should be provided in a separate tab.
- Please provide a "standfirst text" summarizing the study in one or two sentences (approximately 250 characters), three to four "bullet points" highlighting the main findings and a "synopsis image" (550px width and max 400px height, jpeg format) to highlight the paper on our homepage.
- Please include 5 keywords.
- All Materials and Methods need to be described in the main text. We would ask you to use 'Structured Methods', our new Materials and Methods format, which is mandatory for Methods (and Articles with a strong methodological focus). According to this format, the Material and Methods section should include a Reagents and Tools Table (listing key reagents, experimental models, software and relevant equipment and including their sources and relevant identifiers) followed by a Methods and Protocols section in which we encourage the authors to describe their methods using a step-by-step protocol format with bullet points, to facilitate the adoption of the methodologies across labs. More information on how to adhere to this format as well as downloadable templates (.doc or .xls) for the Reagents and Tools Table can be found in our author guidelines: . An example of a Method paper with Structured Methods can be found here: .
- Please include a "Disclosure & Competing Interests Statement".
- Please note that our editorial policy does not allow "Data not Shown".
- Please include a "Data availability" section describing how the data, code etc. have been made available. This section needs to be formatted according to the example below:
The datasets and computer code produced in this study are available in the following databases:

- Chip-Seq data: Gene Expression Omnibus GSE46748 (<https://www.ncbi.nlm.nih.gov/geo/query/acc.cgi?acc=GSE46748>)
- Modeling computer scripts: GitHub (<https://github.com/SysBioChalmers/GECKO/releases/tag/v1.0>)
- [data type]: [full name of the resource] [accession number/identifier] ([doi or URL or identifiers.org/DATABASE:ACCESSION])

- For data quantification: please specify the name of the statistical test used to generate error bars and P values, the number (n) of independent experiments (specify technical or biological replicates) underlying each data point and the test used to calculate p-values in each figure legend. The figure legends should contain a basic description of n, P and the test applied. Graphs must include a description of the bars and the error bars (s.d., s.e.m.).

- When you resubmit your manuscript, please download our CHECKLIST (<https://bit.ly/EMBOPressAuthorChecklist>) and include the completed form in your submission.

Please note that the Author Checklist will be published alongside the paper as part of the transparent process (<https://www.embopress.org/page/journal/17444292/authorguide#transparentprocess>).

If you feel you can satisfactorily deal with these points and those listed by the referees, you may wish to submit a revised version of your manuscript. Please attach a covering letter giving details of the way in which you have handled each of the points raised by the referees. A revised manuscript will be once again subject to review and you probably understand that we can give you no guarantee at this stage that the eventual outcome will be favorable.

Kind regards,

Maria

Maria Polychronidou, PhD
Senior Editor
Molecular Systems Biology

We realize that it is difficult to revise to a specific deadline. In the interest of protecting the conceptual advance provided by the work, we recommend a revision within 3 months (25th Jun 2023). Please discuss the revision progress ahead of this time with the editor if you require more time to complete the revisions.

IMPORTANT: When you send your revision, we will require the following items:

1. the manuscript text in LaTeX, RTF or MS Word format
2. a letter with a detailed description of the changes made in response to the referees. Please specify clearly the exact places in the text (pages and paragraphs) where each change has been made in response to each specific comment given
3. three to four 'bullet points' highlighting the main findings of your study
4. a short 'blurb' text summarizing in two sentences the study (max. 250 characters)
5. a 'thumbnail image' (550px width and max 400px height, Illustrator, PowerPoint or jpeg format), which can be used as 'visual title' for the synopsis section of your paper.
6. Please include an author contributions statement after the Acknowledgements section (see <https://www.embopress.org/page/journal/17444292/authorguide>)
7. Please complete the CHECKLIST available at (<https://bit.ly/EMBOPressAuthorChecklist>). Please note that the Author Checklist will be published alongside the paper as part of the transparent process (<https://www.embopress.org/page/journal/17444292/authorguide#transparentprocess>).
8. When assembling figures, please refer to our figure preparation guideline in order to ensure proper formatting and readability in print as well as on screen: <https://bit.ly/EMBOPressFigurePreparationGuideline>
See also figure legend guidelines: <https://www.embopress.org/page/journal/17444292/authorguide#figureformat>
9. Please note that corresponding authors are required to supply an ORCID ID for their name upon submission of a revised manuscript (EMBO Press signed a joint statement to encourage ORCID adoption). (<https://www.embopress.org/page/journal/17444292/authorguide#editorialprocess>)
Currently, our records indicate that the ORCID for your account is 0000-0001-8072-1630.

Link Not Available

The system will prompt you to fill in your funding and payment information. This will allow Wiley to send you a quote for the article processing charge (APC) in case of acceptance. This quote takes into account any reduction or fee waivers that you may be eligible for. Authors do not need to pay any fees before their manuscript is accepted and transferred to the publisher.

EMBO Press participates in many Publish and Read agreements that allow authors to publish Open Access with reduced/no publication charges. Check your eligibility: <https://authorservices.wiley.com/author-resources/Journal-Authors/open-access/affiliation-policies-payments/index.html>

*** PLEASE NOTE *** As part of the EMBO Press transparent editorial process initiative (see our Editorial at <https://dx.doi.org/10.1038/msb.2010.72>), Molecular Systems Biology publishes online a Review Process File with each accepted manuscripts. This file will be published in conjunction with your paper and will include the anonymous referee reports, your point-by-point response and all pertinent correspondence relating to the manuscript. If you do NOT want this File to be published, please inform the editorial office at msb@embo.org within 14 days upon receipt of the present letter.

Reviewer #1:

The manuscript entitled "AI-guided pipeline for protein-protein interaction drug discovery identifies a SARS-CoV-2 inhibitor" presents a novel method to assess protein-protein interactions (PPIs) using the LuTHy assay and compares its accuracy with other binary interaction assays. Additionally, the authors establish a machine learning algorithm to classify binary interactions and benchmark the performance of AlphaFold-Multimer against established reference sets of protein pairs. The study is well-designed, and the manuscript is overall very well-written. However, there are several weaknesses that need to be addressed prior to publication:

- The manuscript would benefit from a more in-depth discussion of the limitations of the LuTHy assay, such as potential false positives/negatives, and how these limitations may impact the accuracy of the machine learning algorithm.
- The manuscript lacks a thorough comparison of the results obtained with the LuTHy assay and other binary interaction assays, such as GPCA, MAPPIT, NanoBiT, N2H, and SIMPL. A more detailed comparison would help to better understand the advantages and disadvantages of the LuTHy assay and its potential applicability to different protein pairs.
- In the machine learning algorithm development, the authors should describe more clearly the process of selecting and validating the best-performing model. Additionally, the authors should consider including other machine learning algorithms to ensure that the support vector machine (SVM) approach is indeed the best choice for this task.
- The authors could discuss the potential applications of their machine learning algorithm to other types of protein interactions, such as protein-nucleic acid or protein-ligand interactions. This would help to highlight the broader applicability of the approach and its potential impact on the field of molecular systems biology.
- The authors should provide more information on the generalizability of the maSVM algorithm to other complex PPI datasets. It would be beneficial to know if the algorithm can be applied to other types of multiprotein complexes and under what conditions the algorithm performs best.
- In the analysis of AlphaFold-Multimer's performance, the authors should discuss how the limitations of the current version of AlphaFold might impact the results, and whether these limitations might be addressed in future versions of the algorithm.
- The classification of binary interactions within multiprotein complexes is a complex task, and the authors should provide more information on how the maSVM algorithm was adapted for this specific application. The choice of training and test sets, as well as the evaluation of performance, could be discussed in more detail.

Reviewer #2:

In this interesting and complex manuscript, Trepte et al. describe a broad arch from benchmarking PPI detection to the development of a machine learning model integrating heterogeneous experimental PPI results and structural data, to virtual drug screening of a selected interaction and experimental testing of one of the top candidates. This is a very impressive work building on many different experimental and computational methods including several recent advances like AlphaFoldMultimer and VirtualFlow. The results are exciting, and the described pipeline appears applicable to other PPIs.

This applicability though relies on the fact that all resources the pipeline builds on be made available to the community. It is currently labelled that code "will be made available upon request". Good practice is to make code available directly along with the publication (see e.g. <https://doi.org/10.1038/s41597-021-00981-0>). This will be essential for others to use this pipeline and build on this work.

Perhaps due to the complexity of the work, it is at times hard to find relevant details such as features and training of the machine learning model. Perhaps the graphical overview figure could be annotated to indicate which figures and segments deal with the respective part of the work, or have a table-of-contents like structure in supplementary, or other ways to structure the manuscript.

The 8-way tagging for interaction detection is interesting and will provide a good benchmark for the community. The low level of overlap between interactions detected - by what ultimately still is one assay - appears to be somewhat concerning though. Are there possible other explanations? Differences between biophysical and biologically relevant interactions are touched upon in the discussion for false positives, yet looking at Fig. 1J there also appear to be a lot of false negatives. What are the authors' thoughts on implications of this low overlap for interpretation of other studies on PPI detection?

- in Fig. 1J it would be great to add a line with the maSVM prediction to allow the reader to visually assess its predictive power
- the example from the discussion (SLC6A1+TM4SF4) is not easily spotted from Fig. 1J, it may be helpful to highlight it or place a marker

The design and training procedures for the SVM need to be described in detail in the main text, or at the very least the reader should be pointed to the relevant section in the Methods.

- for reproducibility, it would be good to include the division into train, validation and test datasets
- what measures are taken to reduce the risk of overfitting?
- please include training/testing curves and a confusion matrix
- Including these details is crucial both for assessing the soundness of the work as well as for reproducibility.

Lastly, could the authors explain the rationale behind the k-means clustering of the PAE values of interface residues? Which scientific or technical properties are assessed via clustering of the alignment error?

minor comments

- figure EV7G, it is misleading to provide results for 6 (six) complexes in percent, just provide the actual counts
- p14 last line, reference to EV7H,I should be to EV7I,J

Reviewer #3:

Summary

In the manuscript "AI-guided pipeline for protein-protein interaction drug discovery identifies a SARS-CoV-2 inhibitor" Trepte et al developed a scoring algorithm for binary protein protein interaction based on a multi adaptive support vector machine (maSVM) learning algorithm.

Initially, the paper demonstrates the use of fixed cutoffs at a maximum specificity can result in highly variable recovery rates of protein-protein interactions (PPIs), so they developed the multi-adaptive support vector machine (maSVM) learning algorithm to classify PPIs, which showed lower variability of recovered interactions. To benchmark the maSVM algorithm to study interactions between subunits of protein complexes the authors selected 3 multi-subunit complexes, including LAMTOR, MIS12, and BRISC, and demonstrated that maSVM allows to successfully identify subunit interactions.

After training the maSVM on the two initial datasets, the authors applied the maSVM to identify high-confidence PPIs between SARS-CoV-2 proteins. From this dataset they selected an interaction between NSP10-NSP16 which has already been discovered previously and has also been structurally characterized to predict compounds targeting the NSP10-NSP16 interaction interface by virtual drug screening. Finally, one compound from the virtual drug screen (compound 459) was validated to moderately inhibit the NSP10-NSP16 interaction and reduce SARS-CoV-2 replication.

As my expertise is more on the experimental aspect of identifying protein-protein interactions and their biological validation, but less on the computational aspect, my comments focus more on the approach chosen, the conclusions from the data and the biological impact rather than on the computational approach chosen.

General remarks

In general, the manuscript is mostly focusing on a new algorithm for scoring binary protein-proteins interactions and showcasing it in multiple examples to demonstrate the recovery of known interactors. Therefore, I would categorize the nature of advance of the manuscript as technical, specifically on the scoring algorithm. Amongst these examples, the interactions between SARS-COV-2 proteins is one of the applications where the authors could confirm a number of known interactions between SARS-COV-2 proteins and a few novel interactions.

While the authors apply their algorithm to SARS-COV-2 PPIs, I don't think that the title "AI-guided pipeline for protein-protein interaction drug discovery identifies a SARS-CoV-2 inhibitor" is appropriate for this study. The main focus is the scoring algorithm for PPIs, but I do not see how the presented approach is specifically geared towards identifying interactions for drug discovery. If the authors want to keep this title, they should show how the high-scoring interactions from their binary PPI data can be further prioritized for drug discovery, which is however not shown in the manuscript. The combination with the structure prediction using AFM is interesting, but it would be more convincing if the authors would have chosen an interaction that was

either not previously discovered or did not have a previous experimental structure available. Therefore, my suggestion would be to change the title to better represent the strength of the study.

However, I do believe that the study is of interest most likely for researchers using binary PPI mapping approaches to obtain insights in the scoring of their datasets and predicting protein interaction interfaces, but I doubt that it would be of interest for researchers interested in prioritizing PPIs for drug discovery or virologists with interest in SARS-COV-2.

Major comments

The data for the first paragraphs is almost exclusively shown in the supplemental figures, which is confusing. Especially when developing a new scoring algorithm for the PPI data, I would want to see the performance comparison of the new and the old scoring algorithm in the main figure. Also, while there is a performance comparison of the maSVM to the fixed cutoff shown for the LuTHy assay for PPI mapping, the performance comparison for the other approaches SIMPL, NanoBiT, N2H etc is missing and should be added and critically discussed to provide a full characterization of the new algorithm for the reader. Especially since the authors claim that the maSVM algorithm is universal and unbiased, but it is not clear right now how universal it is, as the traditional scoring could outperform maSVM for some of the PPI mapping approaches.

How did the authors decide on splitting the data into 20 training sets? Would the performance characteristics comparing fixed cutoff vs maSVM look differently if the dataset would be split in a smaller number of training sets eg. 10 training sets?

In general, the recovery of the known protein interactions seem low with either of the scoring approaches. Why is this the case and how could this be improved? The authors should add a critical discussion to the manuscript addressing this point.

Why is the probability for the maSVM probability depicted using a 4-color color code, which is highly confusing and not intuitive to read? White in the chosen color code means a probability somewhere between 50 and 75%, but I would actually intuitively think white is the lowest probability. And why is a yellow to red gradient used for lower probabilities and then suddenly the highest probability is turquoise. This is particularly confusing when looking at the heatmaps, which are really hard to understand with this color code. The authors should change the color code to less and more similar colors for the gradient. My sense is that the authors chose this color code only to make the data look better using this color code with high contrasts.

I am struggling with the section "Benchmarking AlphaFold against established reference sets of protein pairs". It does not seem in line with the other sections and comes out of nowhere and after this section the manuscript changes back to benchmarking the maSVM algorithm for discovering interactions in known protein complexes. So, I think it would make it easier for the reader to leave this paragraph out. Especially, since all the figures referring to this section are anyways in the supplementary materials, it means to me that (1) either this section is not relevant for the overall message of the paper or/and (2) putting these figures in the main text would just interrupt the flow of the manuscript. Here, I think both are true. Furthermore, the text in this section and the associated figures are very hard to follow for a reader that is not familiar with structure/interface prediction (the flowchart in figure EV3A does not help the understanding). Finally, I am wondering if it scientifically makes sense to compare the performance of an experimental PPI mapping approach combined with a new scoring algorithm with a structure prediction approach. But given that I am not an expert in structural prediction, I might misunderstand the message/impact of this particular section.

It is interesting that in the intra-complex PPI mapping section LuTHy-LuC shows worse performance than LuTHy-BRET and N2H, which is in contrast to the initial benchmarking using a selection of known binary interaction, for which LuTHy-LuC performed similar than the other approaches. Is there a scientific explanation for this? It would be also interesting to know if there is a complementarity in the different approaches (for both benchmarking dataset)? Are the high-scoring interactions from the different approaches similar or are these different subsets? Also, how does the fixed cutoff approach perform on the intra-complex PPI scoring in comparison to the maSVM. For a rigorous benchmarking of their new algorithm it would be important to include this comparison.

In order to demonstrate that the new maSVM algorithm for scoring binary PPIs can identify novel PPIs in a dataset on which it has not been trained, it would be more convincing and meaningful to validate an interaction not reported before, instead of the known NSP10-NSP16 interaction. The study detected 26 high-confidence SARS-Cov-2 PPIs with the maSVM algorithm (18 known & 8 un-known PPIs). The authors should experimentally validate one of the previously un-known PPIs using an orthogonal experimental approach.

Also the structure prediction followed by drug screening approach would be more convincing if it was validated on an interaction for which there was no previous structure available because it feels like the PPI mapping and the AFM prediction was not needed to do the drug screening and it could have been performed on the previously experimentally determined structure, if I understand correctly.

Is there an explanation why LuTHy-LuC also underperforms in the SARS-CoV-2 dataset compared to LuTHy-BRET? Why are so many previously reported interactions missed in this dataset?

The authors trained the maSVM on two different datasets to generate models using the hsPRS-v2/hsRRSv2 and the multiprotein complex protein pairs and then applied these to the SARS-COV-2 dataset. How would the results differ if the

models were only trained on one of the datasets? In the future if researchers would like to use the maSVM algorithm, does it need to be retrained for each dataset?

There are various previous studies that have investigated SARS-CoV-2 protein interactions with host proteins and targeting some proteins using small molecules (PMID: 32353859, PMID: 32408336). How does this study compare to those? Can we learn complimentary information from this study that has not been provided previously?

Besides Lysine 93 of NSP10, there is an amino acid of NSP16 showing very low ΔG comparable to Lys93 in Figure 4C. Does this amino acid of NSP16 also affect the NSP10-NSP16 interaction? Would it be possible to run a docking screen against this amino acid? Or is this residue not localized at the interaction interface?

For testing compound 459 in cells, was cell survival taken into consideration for the effect on virus replication? The authors should provide data on cell survival with increasing concentrations of compound 459 treatment.

Minor comments

Figure 1 F/G: choosing blue and pink for the figure here is misleading, since I associated the colors immediately with PRS and RRS.

Figure 1J: Indicating the interactions for which structures or homologous structures are available in bold is not easy to understand, also adding the AFM results in this heatmap does not make sense, as the AFM is only described in a later paragraph of the manuscript. I would advise to remove the information about structures in this figure.

Figure 2F: It says, "The number of protein pairs with classifier probabilities of >50%, >75% and >95% are indicated." in page 40, but nothing indicated in Figure 2F.

Figure EV3:

(B) If this data is represented in the manuscript, use a simpler color code for the heatmaps. Why not use a two color gradient?

(C) The boxes and arrows should be colored differently, as the green and yellow chosen is the same color as the rank 1 and 2 model from the predictions, which makes a complicated figure even more confusing.

(D) Unclear for a reader that is not familiar where these 8 clusters come from. Can this be visualized better with examples?

"Figure EV5F" in Page 9 is supposed to be "Figure EV3F".

Only because the inhibition effect of compound 459 on SARS-CoV-2 replication is enhanced upon combination with AZ1 in Figure 5J, it does not seem justified to say compound 459 and AZ1 work on distinct mechanisms as stated on Page 16.

The authors should critically discuss the value of an inhibitor of the binding interface between two viral proteins vs a drug targeting a virus-host interaction.

In the discussion the authors state that the NSP10-NSP16 interface has already been used for drug screening with Virtualflow in a previous study. Are the hits similar or different?

Dear Dr. Wanker,

Thank you again for submitting your work to Molecular Systems Biology. We have now heard back from the three reviewers who agreed to evaluate your study. As you will see below, the reviewers acknowledge that the presented approach seems relevant. They do however raise a series of concerns, which we would ask you to address in a revision.

Without repeating all the points listed below, some of the more fundamental issues are the following:

- Further comparisons of the presented method to existing approaches should be included.

We have now added additional comparisons of the presented machine learning algorithm to state-of-the-art approaches. A side-by-side comparison of the traditional way of scoring binary PPI datasets to the classifier approach presented here can e.g. now be found in Figure 1K and in the manuscript on page 9 lines 234-253.

- The methodology and the machine learning algorithm should be described in better detail. Importantly, the code should be made available.

We agree that the machine learning-based PPI scoring approach needed to be described in more detail and the code should be made accessible and reasonably documented so that everyone is able to apply this to their own PPI datasets. We have added more text thoroughly describing the novel scoring approach (e.g. page 6, lines 155-168) and have added more detail and a step-by-step procedure description in the Materials and Methods section (page 34, line 854 to page 39, line 979). Additionally, the code was made available alongside with thorough documentation and well-described readme information:

<https://github.com/philipptrepte/binary-PPI-classifier> and

<https://github.com/philipptrepte/AFM-Pisa-classifier>

- The limitations and advantages of the presented method should be discussed more extensively.

As requested, we have added additional information on the advantages and limitations of our approach, for example on page 8, line 219 to page 9, line 233 and on page 20, line 548 to page 21, line 562. We have now discussed this topic more extensively and have stated in the discussion the prerequisites that need to be fulfilled to obtain reliable results with this approach.

- Additional validations should be performed to better support the main conclusions.

We have now performed additional experiments for validating the results and to better support the main conclusions. Therefore, we have used an orthogonal assay to validate the newly discovered (not previously described) interactions among SARS-CoV-2 proteins. From the 15 novel PPIs discovered by LuTHy, we tested eight using the mN2H assay, of which we could

validate five. This data is now included in Fig. EV3B-E. The results are described in the manuscript on page 15, lines 402-407.

All issues raised by the referees would need to be satisfactorily addressed. Please let me know in case you would like to discuss in further detail any of the issues raised, I would be happy to schedule a call.

On a more editorial level, we would ask you to address the following points:

- Please provide a .doc version of the manuscript text (including legends for the main figures) and individual production quality figure files for the main Figures (one file per figure).

- We have replaced Supplementary Information by the Expanded View (EV format). In this case, all additional figures and Tables can be included in a PDF called Appendix. Appendix figures and Tables should be labeled and called out as: "Appendix Figure S1, Appendix Figure S2... Appendix Table S1..." etc. Each legend should be below the corresponding Figure/Table in the Appendix. Please include a Table of Contents in the beginning of the Appendix. For detailed instructions regarding expanded view please refer to our Author Guidelines: <http://msb.embopress.org/authorguide#expandedview>.

We have now only three Expanded View Figures, and placed additional figures in the appendix. We also relabeled Supplementary Tables as Appendix Tables.

- Supplementary Tables 1-5 should be provided as Datasets EV1-EV5. Please provide one file per dataset. In each file, a description of the dataset should be provided in a separate tab.

We provide Supplementary Tables now as Appendix Tables in addition to one Appendix Dataset and Source Data for Figures 1 through 5.

- Please provide a "standfirst text" summarizing the study in one or two sentences (approximately 250 characters), three to four "bullet points" highlighting the main findings and a "synopsis image" (550px width and max 400px height, jpeg format) to highlight the paper on our homepage.

We now also included a "standfirst text"

- Please include 5 keywords.

We included 5 keywords on page 2, line 65: protein-protein interactions, machine learning, AlphaFold, VirtualFlow, SARS-CoV-2

- All Materials and Methods need to be described in the main text. We would ask you to use 'Structured Methods', our new Materials and Methods format, which is mandatory for Methods (and

Articles with a strong methodological focus). According to this format, the Material and Methods section should include a Reagents and Tools Table (listing key reagents, experimental models, software and relevant equipment and including their sources and relevant identifiers) followed by a Methods and Protocols section in which we encourage the authors to describe their methods using a step-by-step protocol format with bullet points, to facilitate the adoption of the methodologies across labs. More information on how to adhere to this format as well as downloadable templates (.doc or .xls) for the Reagents and Tools Table can be found in our author guidelines: <https://www.embopress.org/page/journal/17444292/authorguide#textformat>. An example of a Method paper with Structured Methods can be found here: <https://www.embopress.org/doi/10.15252/msb.20178071>.

We now include a ‘Reagents and Tools’ table, and for the new methods presented here, we rewrote the Materials and Methods section using ‘Structured Methods’.

- Please include a "Disclosure & Competing Interests Statement".

This is included in the manuscript.

- Please note that our editorial policy does not allow "Data not Shown".

We had previously not included the cell morphology and confluency data of the BRET compound assay. We have now added this data (Appendix Figure S8) and referenced it in the text accordingly. Thus, all data is included and no “Data not Shown” statement is in the manuscript.

- Please include a "Data availability" section describing how the data, code etc. have been made available. This section needs to be formatted according to the example below:

The datasets and computer code produced in this study are available in the following databases:

- Chip-Seq data: Gene Expression Omnibus GSE46748

(<https://www.ncbi.nlm.nih.gov/geo/query/acc.cgi?acc=GSE46748>)

- Modeling computer scripts: GitHub (<https://github.com/SysBioChalmers/GECKO/releases/tag/v1.0>)

- [data type]: [full name of the resource] [accession number/identifier] ([doi or URL or identifiers.org/DATABASE:ACCESSION])

This is now included.

- For data quantification: please specify the name of the statistical test used to generate error bars and P values, the number (n) of independent experiments (specify technical or biological replicates) underlying each data point and the test used to calculate p-values in each figure legend. The figure legends should contain a basic description of n, P and the test applied. Graphs must include a description of the bars and the error bars (s.d., s.e.m.).

Reviewer #1:

The manuscript entitled "AI-guided pipeline for protein-protein interaction drug discovery identifies a SARS-CoV-2 inhibitor" presents a novel method to assess protein-protein interactions (PPIs) using the LuTHy assay and compares its accuracy with other binary interaction assays. Additionally, the authors establish a machine learning algorithm to classify binary interactions and benchmark the performance of AlphaFold-Multimer against established reference sets of protein pairs.

The study is well-designed, and the manuscript is overall very well-written. However, there are several weaknesses that need to be addressed prior to publication:

We would like to thank the reviewer for the overall positive feedback and their productive and constructive suggestions to improve the manuscript prior to publication.

- The manuscript would benefit from a more in-depth discussion of the limitations of the LuTHy assay, such as potential false positives/negatives, and how these limitations may impact the accuracy of the machine learning algorithm.

We have added additional text in the discussion regarding the limitations of the LuTHy assay and its potential to detect false positive and negative interactions (page 6, lines 179-181; page 8, lines 223-228; and page 20, line 550 to page 21, line 564). As suggested, we have also discussed how this can impact the accuracy of the machine learning algorithm (page 21, lines 559-565). However, one of the advantages of the scoring approach presented here is that it can more robustly handle false-positives e.g. in a random (negative) reference set (page 21, lines 555-559). While in a scoring approach that uses fixed cutoffs at maximal specificity, such false-positive interactions would affect the overall scoring. However, the multi-adaptive sampling SVM is able to strongly reduce their contribution to the classifier. Also, in comparison to distribution based cutoffs, which are purely based on data distribution without using the existing class labels of the PRS and RRS sets, the maSVM algorithm draws "informed" decision boundaries to distinguish between positive and negative protein interaction pairs.

- The manuscript lacks a thorough comparison of the results obtained with the LuTHy assay and other binary interaction assays, such as GPCA, MAPPIT, NanoBiT, N2H, and SIMPL. A more detailed comparison would help to better understand the advantages and disadvantages of the LuTHy assay and its potential applicability to different protein pairs.

In the initial version of the manuscript, we already included a comparison of the results obtained with LuTHy and other binary interaction assays. We have now added an additional side-by-side comparison for LuTHy and all other assays with regard to fixed and distribution-based cutoffs and the maSVM-based scoring approach (see updated Figure 1K and Appendix Figures S3 and S4; page 9, lines 236-255). What can be observed is that most interactions that are detected by any of the tested binary interaction assays, can also be detected with LuTHy (Figure 1J). In general, LuTHy and N2H showed the highest recovery rates in detecting hsPRS-v2 interactions (Figure 1K).

- In the machine learning algorithm development, the authors should describe more clearly the process of selecting and validating the best-performing model. Additionally, the authors should consider including other machine learning algorithms to ensure that the support vector machine (SVM) approach is indeed the best choice for this task.

For this study, a supervised learning method was chosen since we have suitable positive and negative reference interactions. Among supervised learning approaches for classification problems, there are support vector machine (SVM) algorithms, but also linear regression, decision trees, random forest-based methods, k-nearest neighbors, artificial neural networks and others. We have chosen a SVM since it was shown to be quite effective with a low number of features and also a relatively low number of data points. We have not chosen e.g. naive Bayes since it assumes independence between features, which we do not expect when using e.g. the interaction readout and an expression readout (we assume they affect each other). We have not chosen e.g. linear regression since we do not necessarily assume a linear relationship between the features and the outcome. Also, linear regression can be sensitive to outliers.

Nevertheless, we believe that a thorough comparison of machine learning algorithms can be beneficial, wherefore we now include a comparison between random forest, SVM and maSVM machine learning algorithms (Appendix Figure S1). Furthermore, we included a better explanation of the training procedure in the text (page 6, line 148 to page 7, line 185).

We also would like to clarify that we did not select a best-performing model. In fact, during the training procedure, the training is performed using multiple randomly selected training subsets, thereby generating multiple models (with the same architecture). Finally, the average probability prediction is obtained by averaging each of the models' predictions. This was performed to reduce the risk of overfitting (page 6, lines 153-159).

Furthermore, we have now additionally performed extensive parameter evaluation and optimization for the maSVM-based PPI scoring approach. We have compared different ensemble sizes of input models, have analyzed how the number of iterations in the multi-adaptive sampling procedure impacts PPI classification and detection, and finally have evaluated how different values for the “C” regularization hyperparameter impact the models (Append Figure S1; Appendix Dataset 1; page 7, lines 191-194)

- The authors could discuss the potential applications of their machine learning algorithm to other types of protein interactions, such as protein-nucleic acid or protein-ligand interactions. This would help to highlight the broader applicability of the approach and its potential impact on the field of molecular systems biology.

We agree with Reviewer #1 and have added additional text discussing the potential applications of the maSVM algorithm for other methods/datasets including PPI, protein-DNA and -RNA as well as protein-ligand interactions in the discussion (page 21, line 566 to page 22, line 583).

- The authors should provide more information on the generalizability of the maSVM algorithm to other complex PPI datasets. It would be beneficial to know if the algorithm can be applied to other types of multiprotein complexes and under what conditions the algorithm performs best.

By including robust scaler normalization of the training features (Appendix Table 1), described in the methods (page 36, lines 885-888), and by performing hyperparameter optimizations (Append Dataset 1), we now trained the maSVM only on the hsPRS-v2 and hsRRS-v2 and tested its performance on the multiprotein complex dataset (Figure 3). We obtained good recovery rates and therefore concluded that the newly trained models can generally be applied to different types of PPI datasets. Therefore, we used the same models to detect interactions between the SARS-CoV-2 proteins (Figure 4), which resulted in the identification of novel, but also many previously identified SARS-CoV-2 interactions (Figure EV3A). Overall, this suggests that the models trained on the hsPRS-v2/hsRRS-v2 can be applied to complex PPI datasets.

- In the analysis of AlphaFold-Multimer's performance, the authors should discuss how the limitations of the current version of AlphaFold might impact the results, and whether these limitations might be addressed in future versions of the algorithm.

We have now included additional text discussing the limitations of the AlphaFold version used in this study (page 24, line 655-663). Some limitations (e.g. the maximum number of amino acids per protein complex) have already been overcome (when using large computer clusters with sufficient GPU memory). Other current limitations like e.g. prediction of low complexity domains (also due to their low representation in the AlphaFold training dataset of PDB), prediction of membrane protein structure and others are now discussed in the text (page 24, line 663 to page 25, line 666).

- The classification of binary interactions within multiprotein complexes is a complex task, and the authors should provide more information on how the maSVM algorithm was adapted for this specific application. The choice of training and test sets, as well as the evaluation of performance, could be discussed in more detail.

As already described above, we included a robust scaler normalization step of the training features and performed an extensive hyperparameter optimization (Appendix Dataset 1). This allowed us to train the maSVM only on the hsPRS-v2 and hsRRS-v2 PPI sets and applied the models to predict interactions in the multiprotein complex dataset (Figure 3) and in the SARS-CoV-2 dataset (Figure 4).

Importantly, we also evaluated the model performance using learning curves, which are found in Appendix Figure S1.

Reviewer #2:

In this interesting and complex manuscript, Trepte et al. describe a broad arch from benchmarking PPI detection to the development of a machine learning model integrating heterogeneous experimental PPI results and structural data, to virtual drug screening of a selected interaction and experimental testing of one of the top candidates. This is a very impressive work building on many different experimental and computational methods including several recent advances like AlphaFoldMultimer and VirtualFlow. The results are exciting, and the described pipeline appears applicable to other PPIs.

We would like to thank the Reviewer for the positive feedback and constructive criticism. In our view, especially the suggestions regarding the learning curves have significantly improved the manuscript and the maSVM algorithm.

This applicability though relies on the fact that all resources the pipeline builds on be made available to the community. It is currently labelled that code "will be made available upon request". Good practice is to make code available directly along with the publication (see e.g. <https://doi.org/10.1038/s41597-021-00981-0>). This will be essential for others to use this pipeline and build on this work.

We completely agree and now made the code available including a thorough description on GitHub:

- <https://github.com/philipptrepte/binary-PPI-classifier>
- <https://github.com/philipptrepte/AFM-Pisa-classifier>

Perhaps due to the complexity of the work, it is at times hard to find relevant details such as features and training of the machine learning model. Perhaps the graphical overview figure could be annotated to indicate which figures and segments deal with the respective part of the work, or have a table-of-contents like structure in supplementary, or other ways to structure the manuscript.

We agree that it was hard to find the relevant machine learning information easily, wherefore we have now generated a Table that contains the relevant information on training features, feature normalization, ensemble size and number of iterative relabeling in Appendix Table 1.

The 8-way tagging for interaction detection is interesting and will provide a good benchmark for the community. The low level of overlap between interactions detected - by what ultimately still is one assay - appears to be somewhat concerning though. Are there possible other explanations? Differences between biophysical and biologically relevant interactions are touched upon in the discussion for false positives, yet looking at Fig. 1J there also appears to be a lot of false negatives. What are the authors' thoughts on implications of this low overlap for interpretation of other studies on PPI detection?

We have now included new Appendix Figures S3 and S4, where the results of all tagging configurations are shown. We also described these results specifically on page 8, line 220 to page 9, line 234.

Importantly, these results show that we observe an overall very high overlap in the results between interactions tested in multiple configurations.

- in Fig. 1J it would be great to add a line with the maSVM prediction to allow the reader to visually assess its predictive power

The heatmap coloring indicates the probability prediction of the maSVM for an interaction to be true positive. To have an additional line with the probability would duplicate this presentation and in our mind probably not improve readability when drawn for each assay. However, we could make an additional figure in the Appendix that shows the predictions as a line plot.

- the example from the discussion (SLC6A1+TM4SF4) is not easily spotted from Fig. 1J, it may be helpful to highlight it or place a marker

We agree that it is essential to easily spot the respective interaction SLC6A1 + TM4SF4, where we have now included a black arrow highlighting it in Figure 1J.

The design and training procedures for the SVM need to be described in detail in the main text, or at the very least the reader should be pointed to the relevant section in the Methods.

- for reproducibility, it would be good to include the division into train, validation and test datasets

We have now included additional text in the main part of the manuscript to improve the description of the training procedures for the maSVM learning approach (page 6, line 148 to page 7, line 185). Also, we have included a step-by-step instruction on the training procedure in the manuscript (see MATERIALS AND METHODS, Multi-adaptive support vector machine learning algorithm, page 35, line 859 to page 38, line 938). Additionally, the code was made publicly available and a README was attached to guide users through the workflow (<https://github.com/philipptrepte/binary-PPI-classifier/>).

Regarding the division into train, validation and test dataset: We have used an ensemble training approach, in which multiple models are trained with different (randomly sampled) subsets of the total training data (hsPRS/hsRRS). In this approach we have implemented that a prediction of a protein pair in the test set, is never performed if it was already used for training the respective model. If for example the protein pair was used for training in the first ensemble due to random sampling into the training set, it will not be part of the test set and hence not predicted. However, in the second ensemble, it might not be sampled again into the training set, and will hence be present in the test set and thus its classification now predicted.

Furthermore, the 25, 50, or 100 training sets for all assays are now available as Appendix Dataset 2.

- what measures are taken to reduce the risk of overfitting?
- please include training/testing curves and a confusion matrix
- Including these details is crucial both for assessing the soundness of the work as well as for reproducibility.

To reduce the risk of overfitting, we now include Learning curves where Accuracy, Hinge Loss, and Binary-cross Entropy Loss are plotted against different fractions of the training sets (Appendix Figure S1; page 6, line 159-162). By including this analysis, we observed that the training described in the initial manuscript was for many assays suffering from overfitting due to the weighted sampling. Removing weighted sampling and performing hyperparameter optimization (page 7, lines 191-198) followed by plotting the learning curves allowed us to reduce the risk of overfitting and to find optimal training parameters. Furthermore, the results are now also found in a confusion matrix in Appendix Table 7.

Lastly, could the authors explain the rationale behind the k-means clustering of the PAE values of interface residues? Which scientific or technical properties are assessed via clustering of the alignment error?

The PAE (predicted alignment error) gives a distance error for every pair of residues. We rationalized that therefore, PAE values between the amino acids of the two proteins provide an estimated distance between them. We argued that interaction interfaces are formed by more than one amino acid pair, wherefore we decided to computationally search for amino acid clusters with a minimal distance using kmeans clustering. We provide additional text in the manuscript to clarify this topic (page 10, line 270 to page 11, line 285).

minor comments

- figure EV7G, it is misleading to provide results for 6 (six) complexes in percent, just provide the actual counts

We now show the actual number of recovered PPIs in Figures 4I and EV3I.

- p14 last line, reference to EV7H,I should be to EV7I,J

We changed the reference to this figure.

Reviewer #3:

Summary

In the manuscript "AI-guided pipeline for protein-protein interaction drug discovery identifies a SARS-CoV-2 inhibitor" Trepte et al developed a scoring algorithm for binary protein protein interaction based on a multi adaptive support vector machine (maSVM) learning algorithm.

Initially, the paper demonstrates the use of fixed cutoffs at a maximum specificity can result in highly variable recovery rates of protein-protein interactions (PPIs), so they developed the multi-adaptive support vector machine (maSVM) learning algorithm to classify PPIs, which showed lower variability of recovered interactions. To benchmark the maSVM algorithm to study interactions between subunits of protein complexes the authors selected 3 multi-subunit complexes, including LAMTOR, MIS12, and BRISC, and demonstrated that maSVM allows to successfully identify subunit interactions.

After training the maSVM on the two initial datasets, the authors applied the maSVM to identify high-confidence PPIs between SARS-CoV-2 proteins. From this dataset they selected an interaction between NSP10-NSP16 which has already been discovered previously and has also been structurally characterized to predict compounds targeting the NSP10-NSP16 interaction interface by virtual drug screening. Finally, one compound from the virtual drug screen (compound 459) was validated to moderately inhibit the NSP10-NSP16 interaction and reduce SARS-CoV-2 replication.

As my expertise is more on the experimental aspect of identifying protein-protein interactions and their biological validation, but less on the computational aspect, my comments focus more on the approach chosen, the conclusions from the data and the biological impact rather than on the computational approach chosen.

General remarks

In general, the manuscript is mostly focusing on a new algorithm for scoring binary protein-proteins interactions and showcasing it in multiple examples to demonstrate the recovery of known interactors. Therefore, I would categorize the nature of advance of the manuscript as technical, specifically on the scoring algorithm. Amongst these examples, the interactions between SARS-COV-2 proteins is one of the applications where the authors could confirm a number of known interactions between SARS-COV-2 proteins and a few novel interactions.

While the authors apply their algorithm to SARS-COV-2 PPIs, I don't think that the title "AI-guided pipeline for protein-protein interaction drug discovery identifies a SARS-CoV-2 inhibitor" is appropriate for this study. The main focus is the scoring algorithm for PPIs, but I do not see how the presented approach is specifically geared towards identifying interactions for drug discovery. If the authors want to keep this title, they should show how the high-scoring interactions from their binary PPI data can be further prioritized for drug discovery, which is however not shown in the manuscript. The combination with the structure prediction using AFM is interesting, but it would be more convincing if the authors would have chosen an interaction that was either not previously discovered or did not have a previous experimental structure available. Therefore, my suggestion would be to change the title to better represent the strength of the study.

However, I do believe that the study is of interest most likely for researchers using binary PPI mapping approaches to obtain insights in the scoring of their datasets and predicting protein interaction interfaces, but I doubt that it would be of interest for researchers interested in prioritizing PPIs for drug discovery or virologists with interest in SARS-COV-2.

We would like to thank the reviewer for the very thorough evaluation of our manuscript. We believe that many of the points raised by the reviewer, that we have now addressed, improved the quality of our study.

Major comments

The data for the first paragraphs is almost exclusively shown in the supplemental figures, which is confusing. Especially when developing a new scoring algorithm for the PPI data, I would want to see the performance comparison of the new and the old scoring algorithm in the main figure. Also, while there is a performance comparison of the maSVM to the fixed cutoff shown for the LuTHy assay for PPI mapping, the performance comparison for the other approaches SIMPL, NanoBiT, N2H etc is missing and should be added and critically discussed to provide a full characterization of the new algorithm for the reader. Especially since the authors claim that the maSVM algorithm is universal and unbiased, but it is not clear right now how universal it is, as the traditional scoring could outperform maSVM for some of the PPI mapping approaches.

We have now included the comparison of the performance between the old and new scoring algorithms in the main Figure 1K. We would also like to mention that the results of the first paragraph are presented as an “Extended View” Figure, which are directly visible similar to the main figures on the website without having to download Appendix Figures or similar. We also included additional text to better describe the comparison between the methods on page 9 lines 235-255.

How did the authors decide on splitting the data into 20 training sets? Would the performance characteristics comparing fixed cutoff vs maSVM look differently if the dataset would be split in a smaller number of training sets eg. 10 training sets?

We now compared the performance of the maSVM when splitting the data into 25, 50 or 100 training sets. We evaluated the results and performed additional hyperparameter optimizations that can all be found in Html markdown files in Appendix Data 1 (page 7, line 191-194).

In general, the recovery of the known protein interactions seem low with either of the scoring approaches. Why is this the case and how could this be improved? The authors should add a critical discussion to the manuscript addressing this point.

The recovery rate of PRS pairs was extensively studied in our previous work (Choi et al, 2019) where we reported that one PPI assay can only detect a fraction of positive reference PPIs (e.g. ~20% recovery of hsPRS-v2.0 pairs with one PPI assay at maximum specificity). Due to the inherent complexities of proteins such as heterogeneity of protein expression levels, folding efficiency, subcellular localization, and post translational modifications, PPI assays can never reach 100% sensitivity. We have previously shown that highly versatile PPI assays such as multi-environments (N2H) or multi-detection mechanisms (LuTHy) can increase PPI detection.

Indeed, LuTHy and N2H are the top two PPI assays at >95% probability of maSVM (Figure 1K). Further the combination of multiple high-performance PPI assays such as LuTHy and N2H could increase the recovery rate while maintaining high-precision. We have not extensively explored this aspect in this study.

Why is the probability for the maSVM probability depicted using a 4-color color code, which is highly confusing and not intuitive to read? White in the chosen color code means a probability somewhere between 50 and 75%, but I would actually intuitively think white is the lowest probability. And why is a yellow to red gradient used for lower probabilities and then suddenly the highest probability is turquoise. This is particularly confusing when looking at the heatmaps, which are really hard to understand with this color code. The authors should change the color code to less and more similar colors for the gradient. My sense is that the authors chose this color code only to make the data look better using this color code with high contrasts.

We chose this specific color schema since the reader can very easily distinguish between unlikely interactions (yellow to red) and likely interactions (turquoise), which are separated by the white color as the in-between probabilities. In general, we believe that color schemas are often highly subjective. Our personal preference would be to keep this and the other color schema.

I am struggling with the section "Benchmarking AlphaFold against established reference sets of protein pairs". It does not seem in line with the other sections and comes out of nowhere and after this section the manuscript changes back to benchmarking the maSVM algorithm for discovering interactions in known protein complexes. So, I think it would make it easier for the reader to leave this paragraph out. Especially, since all the figures referring to this section are anyways in the supplementary materials, it means to me that (1) either this section is not relevant for the overall message of the paper or/and (2) putting these figures in the main text would just interrupt the flow of the manuscript. Here, I think both are true. Furthermore, the text in this section and the associated figures are very hard to follow for a reader that is not familiar with structure/interface prediction (the flowchart in figure EV3A does not help the understanding). Finally, I am wondering if it scientifically makes sense to compare the performance of an experimental PPI mapping approach combined with a new scoring algorithm with a structure prediction approach. But given that I am not an expert in structural prediction, I might misunderstand the message/impact of this particular section.

We thank the reviewer for highlighting that the section of benchmarking AlphaFold was not well enough implemented in the initial outline of the manuscript. We therefore have put more emphasis on this part in the manuscript by making it a main figure (Figure 2) and describing it in greater detail in the text (page 10, line 256 to page 12, line 311), as we believe it is essential.

It is interesting that in the intra-complex PPI mapping section LuTHy-LuC shows worse performance than LuTHy-BRET and N2H, which is in contrast to the initial benchmarking using a selection of known binary interaction, for which LuTHy-LuC performed similar than the other approaches. Is there a scientific explanation for this? It would be also interesting to know if there is a complementarity in the different approaches (for both benchmarking dataset)? Are the high-scoring

interactions from the different approaches similar or are these different subsets? Also, how does the fixed cutoff approach perform on the intra-complex PPI scoring in comparison to the maSVM. For a rigorous benchmarking of their new algorithm it would be important to include this comparison.

By performing parameter optimizations, and performing a robust scaling normalization on the training features of all data sets, we could now use the models trained on the hsPRS-v2 and hsRRS-v2 to predict the classifier probabilities of the multiprotein complexes and the SARS-CoV-2 dataset. This led to a more generalizability of the maSVM models and resulted in a better performance especially for the LuTHy-LuC data (Figure 3J, Figure 4E).

In addition, in the original description of the LuTHy assay in Trepte et al. 2018 (MSB), we showed that the LuTHy-BRET assay detects interactions with low affinity better than the LuThy-LuC assay. One explanation could therefore be that the lower recovery observed for interactions in the multiprotein complexes and the SARS-CoV-2 datasets is due to the fact that they contain many lower affinity interactions, which cannot be detected with the LuTHy-LuC assay.

We also performed a comparison of the maSVM approach to the fixed cut-off approach on the multi-protein complex dataset for the LuTHy assay. The results from the maSVM are displayed in Figure 3. Since the results confirm that the maSVM results in higher recovery rates than the fixed cut-off scoring approach, we only provide those results in the rebuttal letter below (please compare to Figure 3).

Rebuttal Figure 1: Bar plots showing the fraction of protein pairs in the multi-protein complex dataset that scored above fixed cutoffs at maximum specificity, mean or median plus one standard deviation for LuTHy-BRET and LuTHy-LuC. Cutoffs were calculated using the hsPRS-v2 and hsRRS-v2 reference sets in Figure 1K.

In order to demonstrate that the new maSVM algorithm for scoring binary PPIs can identify novel PPIs in a dataset on which it has not been trained, it would be more convincing and meaningful to

validate an interaction not reported before, instead of the known NSP10-NSP16 interaction. The study detected 26 high-confidence SARS-Cov-2 PPIs with the maSVM algorithm (18 known & 8 un-known PPIs). The authors should experimentally validate one of the previously un-known PPIs using an orthogonal experimental approach.

Also the structure prediction followed by drug screening approach would be more convincing if it was validated on an interaction for which there was no previous structure available because it feels like the PPI mapping and the AFM prediction was not needed to do the drug screening and it could have been performed on the previously experimentally determined structure, if I understand correctly.

We agree with the reviewer that it would have been interesting to perform virtual screening in addition to the NSP10-NSP16 interaction also on a novel interaction. We chose NSP10-NSP16 at the time, since it was known to be critical for viral replication. We did not consider it “problematic” that an experimental structure was already available, since we hoped that it would maximize our chance of success in identifying an inhibitor of SARS-CoV-2. Additionally, we also believed that using an interaction with a known experimental structure would be a good control when setting up the pipeline.

Nevertheless, we still followed your advice and tested 8 of the novel interactions detected by LuTHy with the orthogonal experimental mN2H assay. We could validate 6 of these PPIs with a classifier probability of >95%, suggesting that our SARS-CoV-2 PPI screening resulted in a high quality data set (Figure EV3B-E; page 15, lines 404-409).

Is there an explanation why LuTHy-LuC also underperforms in the SARS-CoV-2 dataset compared to LuTHy-BRET? Why are so many previously reported interactions missed in this dataset?

See comment above.

The authors trained the maSVM on two different datasets to generate models using the hsPRS-v2/hsRRSv2 and the multiprotein complex protein pairs and then applied these to the SARS-COV-2 dataset. How would the results differ if the models were only trained on one of the datasets? In the future if researchers would like to use the maSVM algorithm, does it need to be retrained for each dataset?

We thank you for this comment. During the revision, we have performed extensive hyperparameter optimization on the maSVM scoring algorithm and compared the performance of the classifier after training with different datasets (Appendix Dataset 1). We found that with improved data normalization and hyperparameters, the maSVM classifier model that is trained on the hsPRS-v2/hsRRS-v2 reference sets can also be successfully applied to e.g. the multiprotein complex set (with even improved recovery). Since the hsPRS-v2/hsRRS-v2 reference sets are well established benchmarks that have been tested with a number of assays, we have chosen to only use this as a training dataset. This also enables other researchers to score their PPI data with the trained maSVM models without the need to (re-)screen a new reference set. Also, we have used the hsPRS-v2/hsRRS-v2 classifiers for scoring our SARS-CoV-2 LuTHy data. Hereby, we were able to also improve recovery e.g. of known interactions, when the LuTHy-LuC readout was applied.

There are various previous studies that have investigated SARS-CoV-2 protein interactions with host proteins and targeting some proteins using small molecules (PMID: 32353859, PMID: 32408336). How does this study compare to those? Can we learn complimentary information from this study that has not been provided previously?

We thank the reviewer for his suggestion. Gordon et al. Nature 2020 Jul;583(7816):459-468 (PMID: 32353859), Bjkova et al. Nature 2020 Jul;583(7816):469-472, as well as our previous study (Kim et al. Nature Biotechnol. 2023 Jan;41(1):140-149 are indeed focused on SARS-CoV-2-host protein interactions. Small molecules such as rotatifin (an eIF4A inhibitor), PB28 (a sigma 2 receptor inhibitor), NMS-873 (a P97 ATPase), or AZ1 (a USP25 protease inhibitor) identified in the above studies exhibit inhibition of viral replication by indirectly targeting a host protein. As shown in Figure 6H, the mode of action of compound 459 is likely due to its ability to disrupt an intra-virus protein-protein interaction (i.e. NSP10-NSP16 interaction), as opposed to the above studies or targeting catalytic subunits of a viral enzyme (e.g. Remdesivir and Ritonavir targeting the RdRp polymerase and MPro protease, respectively). As exemplified by current HIV treatment regimens, we envision that combining small molecules with different mechanisms of action will ultimately lead to potent antiviral therapeutics against SARS-CoV-2.

Besides Lysine 93 of NSP10, there is an amino acid of NSP16 showing very low ΔG comparable to Lys93 in Figure 4C. Does this amino acid of NSP16 also affect the NSP10-NSP16 interaction? Would it be possible to run a docking screen against this amino acid? Or is this residue not localized at the interaction interface?

We thank Reviewer #3 for this observation and suggestion. As suggested, we have now also performed site-directed mutagenesis on NSP16 substituting Asp106 with lysine inverting the charge at this position. We then performed BRET donor saturation assays with NSP10 and the NSP16 mutant. Strikingly, we observed a similar loss of binding with the NSP16 Asp106Lys mutant as with the NSP10 Lys93Glu mutant (Figure 4E), confirming that these residues are critical for the interaction (page 17, lines 447-459). Based on these results, one could also run a virtual screening campaign targeting Asp106 of NSP16. However, the NSP10 surface at this interaction hot spot seems slightly better druggable as compared to the corresponding NSP16 surface. Thus, finding suitable binders for the relevant NSP16 interface might be even more challenging.

For testing compound 459 in cells, was cell survival taken into consideration for the effect on virus replication? The authors should provide data on cell survival with increasing concentrations of compound 459 treatment.

Yes, cell survival was taken into consideration when testing compound 459 on viral replication. We have normalized the replication signal to cell viability obtained from the CellTiter-Glo assay (Promega). The cell viability data is now included in the manuscript (Appendix Figure S8; page 19, lines 504-505).

Minor comments

Figure 1 F/G: choosing blue and pink for the figure here is misleading, since I associated the colors immediately with PRS and RRS.

Thank you for this observation and very good suggestion. We have changed the color coding now matching the BRET and LuC coloring instead of the PRS and RRS colors.

Figure 1J: Indicating the interactions for which structures or homologous structures are available in bold is not easy to understand, also adding the AFM results in this heatmap does not make sense, as the AFM is only described in a later paragraph of the manuscript. I would advise to remove the information about structures in this figure.

We thank the reviewer for his/her comment. We removed the AFM results from Figure 1, and instead show the relevant AFM results in a new Figure 2 (previous Figure EV3).

Figure 2F: It says, "The number of protein pairs with classifier probabilities of >50%, >75% and >95% are indicated." in page 40, but nothing indicated in Figure 2F.

Thank you for this remark, it is now corrected.

Figure EV3:

(B) If this data is represented in the manuscript, use a simpler color code for the heatmaps. Why not use a two color gradient?

We have chosen a multi-color gradient for the heatmaps since it allows a better resolution.

(C) The boxes and arrows should be colored differently, as the green and yellow chosen is the same color as the rank 1 and 2 model from the predictions, which makes a complicated figure even more confusing.

Thank you for this remark, we have changed the colors. We also removed the teal boxes. They do not add additional information, but distracted from the main points.

(D) Unclear for a reader that is not familiar where these 8 clusters come from. Can this be visualized better with examples?

We have now improved the visualization of the clusters (see new Figure 2).

"Figure EV5F" in Page 9 is supposed to be "Figure EV3F".

We have corrected the reference.

Only because the inhibition effect of compound 459 on SARS-CoV-2 replication is enhanced upon combination with AZ1 in Figure 5J, it does not seem justified to say compound 459 and AZ1 work on distinct mechanisms as stated on Page 16.

By distinct mechanisms of action, we emphasize the fact that compound 459 acts by binding to the interface between two subunits (NSP10 and NSP16) of the replication/transcription complex (RTC) of SARS-CoV-2. In this case, inhibition of viral replication (Figure 6I) is likely a consequence of inhibition of the 2'-O-ribo methyltransferase as shown in Figure 6B. On the other hand, AZ1 has previously been identified as an inhibitor of deubiquitinating enzymes USP25/28 (<https://pubmed.ncbi.nlm.nih.gov/29131570/>). Subsequently, Kim et al; <https://pubmed.ncbi.nlm.nih.gov/36217029/> showed (i) an interaction between NSP16 and USP25, and (ii) inhibition of SARS-CoV-2 replication by AZ1. In later case, inhibition of viral replication could be a consequence of NSP16 degradation by the proteasome, as we discussed in Kim et al. (<https://pubmed.ncbi.nlm.nih.gov/36217029/>).

The authors should critically discuss the value of an inhibitor of the binding interface between two viral proteins vs a drug targeting a virus-host interaction.

A common and fundamental feature of all viruses is their ability to produce messenger RNA (mRNA) that can be read by the host translational machinery to produce proteins. RNA viruses, such as coronaviruses have a unique property of being the largest (26.4 – 31.7 kb) of currently known positive-strand RNA viruses (Brian and Baric, 2005: <https://pubmed.ncbi.nlm.nih.gov/15609507/>; Woo et al., 2010: <https://pubmed.ncbi.nlm.nih.gov/21994708/>). The precise strategy used by coronaviruses to effectively replicate this large genome is not well-understood. However, like all RNA viruses, coronaviruses must encode for a large replication/transcription complex (RTC) to efficiently synthesize mRNAs from their RNA templates. The RTC complex of SARS-CoV-2 is composed of the RNA-dependent RNA polymerase (RdRp) subunits (NSP7, 2xNSP8, and NSP12 catalytic subunits) (<https://pubmed.ncbi.nlm.nih.gov/32358203/>), the helicase (NSP13), the 3'-5' exonuclease (NSP14), the PolyU-specific endoribonuclease (NSP15) and the 2'-O-ribo methyltransferase (NSP16), and stabilizing subunits NSP9 and NSP10 (<https://pubmed.ncbi.nlm.nih.gov/26159422/>, <https://pubmed.ncbi.nlm.nih.gov/33232691/>). Proper functioning of each enzymatic activity of the above subunits is necessary for the replication of the large coronavirus genome, and an inhibitor of the binding interface between NSP16 and NSP10 could destabilize the entire RTC complex and specifically affect the 2'-O-ribo methyltransferase as shown on Figure 6B. We have previously described the organization of protein complexes into inner versus outer binary interactions (<https://www.biorxiv.org/content/10.1101/2021.03.16.435663v1>). By analogy to this model described for yeast complexomes, drug targeting inner binary interactions, such as interactions

between subunits of the RTC complex of SARS-CoV-2, would mainly affect the core enzymatic function of the complex (in this case viral genome replication), while drugs targeting a virus-host interaction (e.g. Interaction between NSP16 and USP25 <https://pubmed.ncbi.nlm.nih.gov/36217029/>) would affect a context-specific situation in the life cycle of the virus (e.g. viral proteins stability). To better reflect this point, we added the following sentences in the discussion of the manuscript:

Further we now discuss three major advantages of developing drugs targeting viral PPIs over virus-host interactions (see page 23, lines 627-630 and page 23, 633 to page 24 line 645) which involve 1) requiring far less resources (650 vs 454,272 pairwise search space for interaction mapping) for building PPI maps, 2) less off-target impacts in the host, and 3) plausibility of building resistant drugs against viral evolution by leveraging the evolutionary conserved vPPIs.

In the discussion the authors state that the NSP10-NSP16 interface has already been used for drug screening with Virtualflow in a previous study. Are the hits similar or different?

The previous study by Gorgulla et al. 2021 (PMID: 33426509) also targeted the NSP10 interaction interface with NSP16 by ultra-large virtual screening. There are two molecules presented from the top hits, of which one engages a similar binding mode predicted to also engage the critical lysine 93 residue in NSP10 (Gorgulla et al. 2021, Figure S36). However, this compound represents a different scaffold as the hit molecule (compound 459) from this study. We have added this information in the discussion (page 23, lines 612-628).

9th Jan 2024

Manuscript Number: MSB-2023-11595R

Title: AI-guided pipeline for protein-protein interaction drug discovery identifies an SARS-CoV-2 inhibitor

Dear Erich,

I hope you had a good start in 2024, all by best wishes for a Happy New Year! Thank you for sending us your revised manuscript. We have now heard back from the three reviewers who were asked to evaluate your revised study. As you will see below, the reviewers think that the performed revisions have addressed their major concerns and they support publication. Reviewer #3 lists a few remaining minor issues, which we would ask you to address in a final round of revision. We would also ask you to address some editorial issues listed below.

- Our data editors have noticed some missing information in the figure legends, that needs to be fixed:
 - Information related to n is missing in the legend of figures 1h-i, k; 2h; 3i-k; 5c, h; EV2b; EV3d, j-k.
 - The error bars are not defined in the legend of figures 2h; 6i, j; EV2b; EV3d, j-k.
- The funding information provided in the manuscript text should match the information entered in the online submission system. Currently the following funding information is missing from our submission system: PER-40003579, Wallonia-Brussels International (WBI)-World Excellence Fellowship.
- Please remove the 'Authors Contributions' from the manuscript. The 'Author Contributions' section is replaced by the CRediT contributor roles taxonomy to specify the contributions of each author in the journal submission system. Please use the free text box in the 'author information' section of the online submission system to provide more detailed descriptions if needed (e.g., 'X provided intracellular Ca⁺⁺ measurements in fig Y').
- Please include callouts to Figures 4F, 4G and Appendix Table S7 in the main text.
- Please fill all boxes of the checklist (currently D101 has not been answered).
- Appendix Datasets and Appendix Tables need to be renamed to Datasets EV1- and EV9. Please make sure that the callouts in the text are updated accordingly. A description of the EV Dataset should be provided in a separate sheet in each xls file.
- All Appendix figures should be provided in a PDF called Appendix. Please include a Table of Contents (with page numbers) in the beginning of the Appendix. Appendix figures should be labeled and called out as: "Appendix Figure S1, Appendix Figure S2..." etc. Each legend should be below the corresponding Figure in the Appendix. For detailed instructions regarding expanded view please refer to our Author Guidelines: . All information related to the Appendix should be removed from the main text file.
- Please provide a "standfirst text" summarizing the study in one or two sentences (approximately 250 characters), three to four "bullet points" highlighting the main findings.
- The manuscript sections should be in the following order: Title page - Abstract & Keywords - Introduction - Results - Discussion - Materials & Methods - Data Availability - Acknowledgments - Disclosure Statement & Competing Interests - References - Figure Legends - Tables with legends - Expanded View Figure Legends.

Please resubmit your revised manuscript online, with a cover letter listing amendments and responses to each point raised by the referees. Please resubmit the paper ****within one month**** and ideally as soon as possible. If we do not receive the revised manuscript within this time period, the file might be closed and any subsequent resubmission would be treated as a new manuscript. Please use the Manuscript Number (above) in all correspondence.

When you resubmit your manuscript, please download our CHECKLIST (<https://bit.ly/EMBOPressAuthorChecklist>) and include the completed form in your submission. ***Please note*** that the Author Checklist will be published alongside the paper as part of the transparent process (<https://www.embopress.org/page/journal/17444292/authorguide#transparentprocess>)

Best wishes,

Maria

Maria Polychronidou, PhD
Senior Editor
Molecular Systems Biology

If you do choose to resubmit, please click on the link below to submit the revision online before 8th Feb 2024.

IMPORTANT:

Please note that corresponding authors are required to supply an ORCID ID for their name upon submission of a revised manuscript (EMBO Press signed a joint statement to encourage ORCID adoption).
(<https://www.embopress.org/page/journal/17444292/authorguide#editorialprocess>)
Currently, our records indicate that the ORCID for your account is 0000-0001-8072-1630.

Please click the link below to modify this ORCID:
Link Not Available

***** PLEASE NOTE ***** As part of the EMBO Press transparent editorial process initiative (see our Editorial at <https://dx.doi.org/10.1038/msb.2010.72>), Molecular Systems Biology will publish online a Review Process File to accompany accepted manuscripts. When preparing your letter of response, please be aware that in the event of acceptance, your cover letter/point-by-point document will be included as part of this File, which will be available to the scientific community. More information about this initiative is available in our Instructions to Authors. If you have any questions about this initiative, please contact the editorial office (msb@embo.org).

Reviewer #1:

The authors have addressed the issues raised by myself as well as the other reviewers adequately and the manuscript clarity, quality and readability has improved significantly. I have no further critique.

Reviewer #2:

The authors have clearly put a lot of work into the revision and addressed all my major concerns. I have some minor comments still:

The code has been made available, there is no readme for <https://github.com/philipptrepte/AFM-Pisa-classifier> though, please add that.

In/around Fig 1: It would help to illustrate, e.g. in a flow chart, how the individual components are combined into maSVM.

WRT general applicability, It should be made clear to the reader that the universal applicability (bottom p.9) requires fine-tuning of the hyperparameters to the respective assay.

The manuscript has become long and it can be hard to find the relevant parts. For example, although the authors state in the response to reviewers that limitations are discussed now, searching for the term "limitations" does not bring up those sections, and there is no related headline. The current version is great for the reader who wants a lot of details, but those who are further from the work and would like a broader overview may have a harder time. Subheaders with relevant titles could help alleviate this, if compatible with the journal's guidelines.

Reviewer #3:

In the revised version of the manuscript "AI-guided pipeline for protein-protein interaction drug discovery identifies a SARS-CoV-2 inhibitor" Trepte et al. addressed the main concerns raised based on the initial submission sufficiently, so that the manuscript is in my opinion acceptable for publication.

All editorial and formatting issues were resolved by the authors.

23rd Jan 2024

Manuscript number: MSB-2023-11595RR

Title: AI-guided pipeline for protein-protein interaction drug discovery identifies an SARS-CoV-2 inhibitor

Dear Erich,

Thank you again for sending us your revised manuscript. We are now satisfied with the modifications made and I am pleased to inform you that your study has been accepted for publication.

Kind regards,

Maria

Maria Polychronidou, PhD
Senior Editor
Molecular Systems Biology
